# A global catalogue of $CO_2$ emissions and co-emitted species from power plants including high resolution vertical and temporal profiles

Marc Guevara[1], Santiago Enciso[1,a], Carles Tena[1], Oriol Jorba[1], Stijn Dellaert[2], Hugo Denier van der Gon[2], Carlos Pérez García-Pando[1,3]

[1] Barcelona Supercomputing Center, Barcelona, Spain
[2] TNO, Department of Climate, Air and Sustainability, Utrecht, the Netherland
[3] ICREA, Catalan Institution for Research and Advanced Studies, Barcelona, Spain
[a] now at: Applus IDIADA Group, Tarragona, Spain

*Correspondence to*: Marc Guevara (marc.guevara@bsc.es)

**Abstract.** We present a high-resolution global emission catalogue of $CO_2$ and co-emitted species ($NO_x$, $SO_2$, $CO$, $CH_4$) from thermal power plants for the year 2018. The construction of the database follows a bottom-up approach, which combines plant-specific information with national energy consumption statistics and fuel-dependent emission factors for CO2 and emission ratios for co-emitted species (e.g., amount of NOx emitted relative to $CO_2$; $NO_x/CO_2$). The resulting catalog contains annual emission information for more than 16000 individual facilities at their exact geographical location. Each facility is linked to a country- and fuel-dependent temporal profile (i.e., monthly, day-of-the-week and hourly) and plant-level vertical profile, which were derived from national electricity generation statistics and plume rise calculations that combine stack parameters with meteorological information, respectively. The combination of the aforementioned information allows to derive high-resolution spatial and temporal emissions for modelling purposes. Estimated annual emissions were compared against independent plant- and country-level inventories, including the Carbon Monitoring for Action (CARMA), the Global Infrastructure emission Database (GID) and the Emissions Database for Global Atmospheric Research (EDGAR) databases, as well as officially reported emission data. An overall good agreement is observed between datasets when comparing the $CO_2$ emissions. The main discrepancies are related to the non-inclusion of auto-producer or heat-only facilities in certain countries due to lack of data. Larger inconsistencies are obtained when comparing emissions from co-emitted species due to uncertainties in the fuel- and country/region-dependent emission ratios and gap-filling procedures. The temporal distribution of emissions obtained in this work was compared against traditional sector-dependent profiles that are widely used in modelling efforts. This highlighted important differences and the need to consider country dependencies when temporally distributing emissions. The resulting catalogue (https://doi.org/10.5281/zenodo.10002124, Guevara et al., 2023) is developed in the framework of the Prototype System for a Copernicus $CO_2$ service (CoCO2) EU-funded project to support the development of the Copernicus $CO_2$ Monitoring and Verification Support capacity (CO2MVS).

# 1    Introduction

Over 40% of fossil fuel related carbon dioxide ($CO_2$) emissions are caused by power plants that burn fuels to produce electricity and/or heat (Crippa et al., 2022). A correct representation of the spatial and temporal distribution of these point sources is important for verification of global $CO_2$ emissions through current and future satellite emission monitoring and inverse modelling efforts, like the envisioned European $CO_2$ Monitoring and Verification Support capacity (CO2MVS; Balsamo et al., 2021). The CO2MVS, which is planned to be fully operational by 2026, combines information from various observational data sets (i.e., satellite data from existing or new Copernicus Sentinel satellites and in situ data from various surface networks) and prior knowledge (i.e., mainly bottom-up emission estimates from inventories and reporting) with detailed Earth system modelling and data assimilation capabilities. The final goal of the CO2MVS capacity is to provide observation-based estimates of $CO_2$ emissions at multiple scales (i.e., from global to local industrial and urban hotspots) with a similar level of robustness that has proven critically important in other Copernicus Atmosphere Monitoring Service (CAMS) applications, such as air quality predictions (https://atmosphere.copernicus.eu/air-quality). To reduce the uncertainty in the inversion system and have higher accuracy in final predicted emission estimates, having high spatial and temporal resolution data for $CO_2$ emissions and co-emitted species (e.g., $NO_x$, CO), which are also used to derive observation-based $CO_2$ emissions as they can be detected more easily in satellite images (e.g., Kuhlmann et al., 2021), is a key element.

The spatial representation of large point sources in global state-of-the-art and/or widely used gridded emission inventories such as the Emissions Database for Global Atmospheric Research (EDGAR, Janssens-Maenhout et al., 2019) and the Open-Data Inventory for Anthropogenic Carbon dioxide (ODIAC, Oda et al., 2018) is primarily based on the Carbon Monitoring for Action (CARMA; Wheeler and Ummel, 2008), which was build using plant-level information from 2009 and is no longer maintained. Moreover, these inventories do not report the emissions from facilities at their exact geographical locations, but in the centroid of the respective inventory grid cells which typically have resolution of 0.1x0.1 degrees. Subsequently deviations from their exact locations can be up to a few kilometres. While this fact does not entail limitations for modelling applications working at the same or lower spatial resolutions (e.g., Agustí-Panadera et al., 2022), it may become critical for local and very high-resolution modelling applications (e.g., Brunner et al., 2023). The more recently developed Global Infrastructure emission Database (GID, Tong et al., 2018) overcomes this limitation by providing up-to-date information and high-resolution $CO_2$ emissions from global power plants at the facility-level. However, the latitude and longitude coordinates of each facility are not publicly available, instead georeferenced data is distributed in gridded format at a 0.1x0.1 degrees resolution). Moreover, no information is provided on how to distribute the emissions from each plant temporally and vertically, two parameters that are also essential for modelling purposes (e.g., Brunner et al. 2019; Guevara et al., 2021).

Here we present a global catalogue of $CO_2$ emissions and co-emitted species (i.e., $NO_x$, $SO_x$, CO, $CH_4$) from power plants at high spatial and temporal resolution for the year 2018. The dataset contains annual emission information for individual thermal power plants that burn coal, natural gas, oil, solid biomass and municipal/industrial solid waste (hereinafter referred to as waste) to produce electricity or combined heat and electricity at their exact geographical location. Moreover, each facility is

linked to a country- and fuel-dependent temporal profile (i.e., monthly, day-of-the-week and hourly) and facility-level vertical

distribution profile, which allows to derive spatial- and temporal-resolved emissions for modelling efforts.

Section 2 of the manuscript describes the methodology and databases considered for the construction of the global point source database, while Sect. 3 presents the main results and compares them against existing emission inventories at the plant-, grid- and country-level. Section 4 provides a description of the data availability, Sect. 5 lists the main limitations of the dataset and finally Sect. 6 presents the main conclusions and future perspective.

## 2    Methodology

The approach to construct the global point source database is divided in five phases: 1) Selection of facilities and definition of associated geographical location (i.e., latitude and longitude coordinates), 2) fuel allocation per facility, 3) estimation of annual emissions of $CO_2$ and co-emitted species (i.e., $NO_x$, $SO_x$, $CO$, $CH_4$) per facility, 4) construction of the monthly, weekly (day-of-the-week) and hourly (hour-of-the-day) temporal profiles associated to each facility and 5) construction of the vertical

distribution profiles associated to each facility.

The global point source database is a mosaic constructed using as a basis the European and global power plant databases described in Sect. 2.1. The temporal and vertical profiles associated to each plant are constructed following a common approach that uses as a basis information on measured electricity statistics and plume rise calculations, respectively (Sect. 2.4 and 2.5, respectively). The sources of information and approaches used to develop each dataset are described in the following sub-

sections.

### 2.1    Compilation of facilities and geographical locations

To compile information of each individual power plant including its name and exact geographical location, several public and commercial datasets were combined (Table 1). For the European database, the data sources used are the European Pollutant and Transfer Register database (E-PRTR_v18; EEA, 2020), the Large Combustion Plants database (LCP_v5.2; EEA, 2019),

the Platts World Electric Power Plant dataset (WEPP Europe, September 2015, Platts, 2015) and the integrated Industrial Reporting Database (IRD_v7; EEA, 2022). For the non-European database, the datasets considered included the Global Coal Plant Tracker (GCPTv2021_01; GEM, 2021a), the Global Gas Plant Tracker (GGPTv2021_02; GEM, 2021b), the Global Power Plant Database (GPPDv1.3.0; Global Energy Observatory et al., 2021), the IndustryAbout database (IndustryAbout, 2021), the Emissions and Generation Resource Integrated Database (eGRIDv2018; US EPA, 2020), the Chinese Ministry of

Ecology and Environment's domestic waste incineration power plant database (MIEE, 2022), the Tai biomass power plant database (DEDE; 2022), the Geocomunes Mexican power plant database (Geocomunes, 2020), the Taiwanese waste-to-energy plant database (Taiwan EPA, 2014), the electrical Japan power station database (Electrical Japan, 2022), the Argentinian renewable power plant database (MINEM, 2022) and the UNFCCC Clean Development Mechanism database (UNFCCC CMD, 2022). For both the European and non-European databases, substantial effort was put into identifying missing and

incorrect facility geographical locations. Coordinates were checked or searched manually using Google Maps or other websites and added to the dataset as follows:

- For Europe, the reported coordinates were consistently checked and corrected for the top-100 facilities (in terms of 2017 $CO_2$ emissions). Furthermore, all coordinates that did not fall within the correct country borders, or which were inconsistent between reported dataset versions, were manually checked and corrected. In addition, many other coordinates
(likely about 400) were checked during the process of linking up facilities between datasets, identifying fuel types, and by looking at the resulting emission maps. In total, all checks resulted in 360 plants with corrected coordinates, including about 75 of the top-100 plants.

- For the non-European dataset, the review process was performed for selected countries that are among the top 30 countries in terms of installed power generation capacity and that are representative of coal (i.e., South Africa, Japan, Taiwan,
Kazakhstan, Australia, Vietnam and Turkey), natural gas (i.e., Japan, Oman, Thailand, Bahrain, Algeria, Ukraine) and oil (i.e., Egypt, Iran, Iraq, Libya, Pakistan, Saudi Arabia) power plants. In both cases, some corrections improve the coordinates by only tens of meters or less, in other cases the original coordinates were further off. Multi-unit power plants were in most of the cases located at the same coordinates, since the distance between units is usually small (i.e., dozens of meters). However, in facilities where the distance between units was significant (i.e., few kilometres), original
coordinates were edited and assigned to individual units. Despite these efforts, there may be some errors still present in the dataset, especially in the case of small plants.

**Table 1 Main characteristics of the power plant databases considered in this work**

| dataset | information | fuels | countries | year | reference |
|---|---|---|---|---|---|
| E-PRTR_v18 | Name, geographical coordinates, annual emissions | Coal, natural gas, oil, biomass, waste | EU-27 plus United Kingdom, Norway, Switzerland and Serbia | 2007-2017 | EEA, 2020 |
| LCP v5.2 | Name, fuel input by type, geographical coordinates, annual emissions | Biomass, other solid fuels, liquid, natural gas, other gases | EU-27 plus United Kingdom | 2004-2017 | EEA, 2019 |
| IRD v.7 (combined EPRTR / LCP reporting) | Name, fuel input by type, geographical coordinates, annual emissions | Biomass, other solid fuels, liquid, natural gas, other gases | EU-27 plus United Kingdom, Norway, Switzerland and Serbia | 2007-2020 | EEA, 2022 |

| | | | | | |
|---|---|---|---|---|---|
| WEPP Europe v.2015 | Name, fuel type, capacity, city | Aggregated to: Biomass, solid fuels, liquid fuels, natural gas, waste | All European countries | 2015 | Platts, 2015 |
| eGRIDv2018 | Name, capacity, fuel type, geographical coordinates, annual emissions of $CO_2$, $NO_x$, $SO_2$ and $CH_4$ | Coal, natural gas, oil, biomass, waste | United States | 2018 | US EPA (2020) |
| GCPTv2021_01 | Name, capacity, fuel type, geographical coordinates, status, start/retire year | Coal | All except for United States, and EU-27 plus United Kingdom, Norway, Switzerland and Serbia | 2021 [(*)] | GEM (2021a) |
| GGPTv2021_02 | Name, capacity, fuel type, geographical coordinates, status, start/retire year | Natural gas | China, Japan, Republic of Korea, Indonesia, Thailand, Turkey, Philippines, Israel, Hong Kong, Oman, Bahrain, Myanmar | 2021 [(*)] | GEM (2021b) |
| GPPDv1.3.0 | Name, capacity, fuel type, geographical coordinates | Natural gas | Countries not covered by the other databases | 2021 [(**)] | Global Energy Observatory et al. (2021) |
| | | Oil | China, India, Russia, Brazil, Cuba, Lebanon, Guatemala, Nicaragua, Cameroon, Ethiopia | | |
| | | Waste | Countries not covered by other databases | | |
| | | Biomass | Countries not covered by other databases | | |
| IndustryAbout | Name, capacity, fuel type, geographical coordinates, status, start year | Natural gas | Iran, Egypt, South Africa, Canada, Ukraine, Argentina, Malaysia, Pakistan, Kazakhstan, Kuwait, Chile, Libya, Nigeria, Syria, Colombia, Puerto Rico, Turkmenistan, Dominican Republic, Angola, New Zealand, Ivory Coast, Tanzania, Brunei, Mozambique | 2021 [(**)] | IndustryAbout (2021) |
| | | Oil | Countries not covered by other databases | | |
| | | Waste | Japan, Republic of Korea, Egypt, Turkey, United Arab Emirates, Ukraine, Venezuela, Philippines | | |
| | | Biomass | Egypt, Turkey, United Arab Emirates, Ukraine, Venezuela, Philippines | | |
| MIEE | Name, capacity, fuel type, geographical coordinates | Waste | China | 2022 [(**)] | MIEE (2022) |
| Taiwan EPA | Name, capacity, fuel type, geographical coordinates | Waste | Taiwan | 2014 [(**)] | Taiwan EPA (2014) |
| DEDE | Name, capacity, fuel type, geographical coordinates | Biomass | Thailand | 2022 [(**)] | DEDE (2022) |

| Geocomunes | Name, capacity, fuel type, geographical coordinates | Natural gas, oil | Mexico | 2020 [**] | Geocomunes (2020) |
|---|---|---|---|---|---|
| Electrical Japan | Name, capacity, fuel type, geographical coordinates | Biomass | Japan | 2022 [**] | Electrical Japan (2022) |
| Argentinian renewable plant database | Name, capacity, fuel type, geographical coordinates | Biomass | Argentina | 2022 [**] | MINEM (2022) |
| UNFCCC CMD | Name, capacity, fuel type, address | Biomass | China, Indonesia, Malaysia | 2022 [**] | UNFCCC CMD (2022) |

[*] we only considered those facilities that were operating in 2018
[**] we assume all reported facilities were already/still operating in 2018

## 2.2 Fuel allocation

Each of the emission values in the European power plant dataset is allocated to one of five fuel types (i.e., biomass, coal, oil, natural gas or waste). Three methods were used to allocate the fuel type:

1. Link with LCP dataset: As LCP reporting includes the reporting of fuel input (but not for waste), this could be used to allocate emissions to different fuels when there was a link between an E-PRTR and LCP facility. Still, as only one emission value is reported, in case of a multi-fuel plant (e.g., co-combustion of biomass in a coal-fired power plant), a proxy emission value for each fuel type was estimated using country- and fuel-specific emission factors from the Greenhouse Gas–Air Pollution Interactions and Synergies (GAINS) model at the International Institute for Applied Systems Analysis (IIASA) (Amann et al., 2011; Klimont et al., 2017). The ratio between the proxy emission values was then used to allocate the actual emission values to specific fuel types.

2. Link with Platts WEPP dataset: If no LCP fuel data was available, for some plants the fuel type could be taken from a link with the Platts WEPP dataset. The Platts WEPP dataset contains a detailed fuel type for every electricity-producing unit and also lists the electric capacity for every unit. For those facilities that could not be successfully linked to an LCP plant, a link was made to electricity producing units in the Platts WEPP database. The listed power and fuel type of the units was used together with country- and fuel specific emission factors from the GAINS model to estimate a proxy emission value for each unit and attribute the emissions to different fuel types.

3. Manual search and allocation of fuel types for 133 remaining plants.

For non-European power plants, we used the plant-level fuel information provided by the databases listed in Sect. 2.1, which only report the main fuel even in the cases of multi-fuel plants. Therefore, for each power plant all emissions are linked to one single fuel, as we did not have information to split emissions between fuels in multi-fuel plants, as done for the European dataset. This limitation could have an impact on dual-fuel power plants that can use more than one fuel to operate (e.g., natural gas/diesel), as only emissions from its primary fuel will be allocated in them. To homogenise the results reported by the

European and non-European datasets, we assigned to each European power plant the fuel with the largest contribution to total $CO_2$ emissions.

## 2.3 Estimation of annual $CO_2$ emissions and co-emitted species

### 2.3.1 Europe

For European power plants, annual emissions were derived as a first step from the E-PRTR reporting in the E-PRTR_v18 and IRD v.7 databases. However, for many facilities, gaps in the E-PRTR emission reporting were identified and had to be corrected following a gap filling routine (see below). The gaps are mainly due to the E-PRTR emission reporting thresholds, which obliges companies to report emissions from individual pollutants only if they are above the values summarised in Table 2. Given the pollutant-specific reporting threshold for companies, many facilities report emissions for only a small number of pollutants. $NO_x$ and $CO_2$ are the pollutants that are on average reported most often. $CH_4$ reporting is almost non-existent for power plants, while CO and $SO_x$ are reported for a limited number of facilities, and more often in the earlier years (2004 – 2010) and less in recent years, when annual emission may lie more often below the reporting threshold due to emission reduction technologies. Reporting for large combustion plants (LCP) is not dependent on an emission threshold but is mandatory for all combustion plants from 50 MW or higher thermal input capacity, excluding ovens and certain types of chemical reactors. For each LCP, annual reporting emissions of $NO_x$, $SO_x$, PM and fuel input by fuel type is required.

**Table 2 Summary of the E-PRTR emission reporting thresholds per pollutant**

| Pollutant | E-PRTR threshold (ton/year) |
|---|---|
| $CH_4$ | 100 |
| CO | 500 |
| $CO_2$ | 100000 |
| $NO_X$ | 100 |
| $SO_X$ | 150 |

To complete the reporting for all five pollutants, a 5-step gap filling routine was designed that follows several steps to estimate missing emission values for each power plant (Fig. 1):

1. In gap filling step 1, the E-PRTR and LCP reported values are compared for those years that reporting exists in both datasets for a specific plant. As the scope of an E-PRTR facility can be broader than just the LCPs at that location, the E-PRTR reported emission are typically similar or higher than the LCP reported emissions. If the correlation between both series is >0.5, it is assumed that the trends in the LCP reported emissions are also representative for the trends in activity for the complete facility. In that case, the LCP value is used, multiplied with the average ratio between the E-PRTR and LCP reported emission values. This way, if the EPRTR facility typically encompasses several smaller units that are not in the LCP dataset (i.e., <50MWth), the gap filled emission value incorporates this

relatively fixed ratio between E-PRTR and LCP emissions. The gap filled emission value is capped at the highest reported emission value in the time series for this specific facility to limit the risk of gap filling unrealistic emission values. When the correlation is <0.5, but the aggregated ratio of the series total emissions is between 0.9 - 1.1, or if the median ratio between individual emission values for each year is between 0.9 - 1.1, the LCP value is used directly, as the two time series are considered to be sufficiently consistent, but no reliable adjustment ratio can be estimated.

2. In gap filling step 2, when no E-PRTR reporting for a specific pollutant is available for any year, or for none of the years where LCP reporting is available (which would allow a comparison), the LCP emission value is used directly when available, as this officially reported emission value is the most reliable estimate available and is assumed to be close to the total facility emissions in most cases.

3. After gap filling using LCP data, many gaps in the emission reporting remained. It was decided to fill the remaining gaps if emissions for at least one pollutant had been reported for the facility in a given year (implying the facility was active). Gap filling step 3 is performed by calculating average ratios between reported $CO_2$ emissions and the reported emissions of other pollutants for the specific facility, as the facility specific ratio between the fuel consumption and emission rates is assumed to be the most reliable relationship at this point. When specific pollutant emissions were missing, but $CO_2$ emissions were available, the plant-specific ratio between $CO_2$ and the missing pollutant is used to estimate the missing emission. When fuel use information was not available, the use of pollutant ratios is also deemed the most appropriate method to gap fill missing $CO_2$ emissions. However, $CO_2$ is only gap filled in this step when a $NO_x$ value is reported, as this ratio is typically more stable than for the other co-emitted pollutants. Assuming the downward trend (e.g., lowering of $SO_x/CO_2$ ratio over time due to increased implementation of abatement technologies) of country-, fuel- and year-specific emission factors from the GAINS model, the emission ratios based on co-reporting in earlier years are adjusted before using in later years to account for the effect of increasing use of abatement technologies.

4. In gap filling step 4, missing emission values were gap filled using the ratio between the IIASA GAINS model implied emission factors (IIASA, 2018) (e.g., $CO_2/CO$ ratio) for a specific country, year, fuel type and pollutant, applied to a $CO_2$ value established from E-PRTR reporting or gap filling steps 1 or 2.

5. In gap filling step 5, all emission values that are still missing are gap filled, by applying the ratio between GAINS emission factors on values gap filled in steps 3 or 4. For $CH_4$, a separate fuel-specific $CO_2/CH_4$ ratio is used to gap fill emission values based on the Tier 1 emission factors reported by the IPCC guidelines (Eggleston et al., 2006).

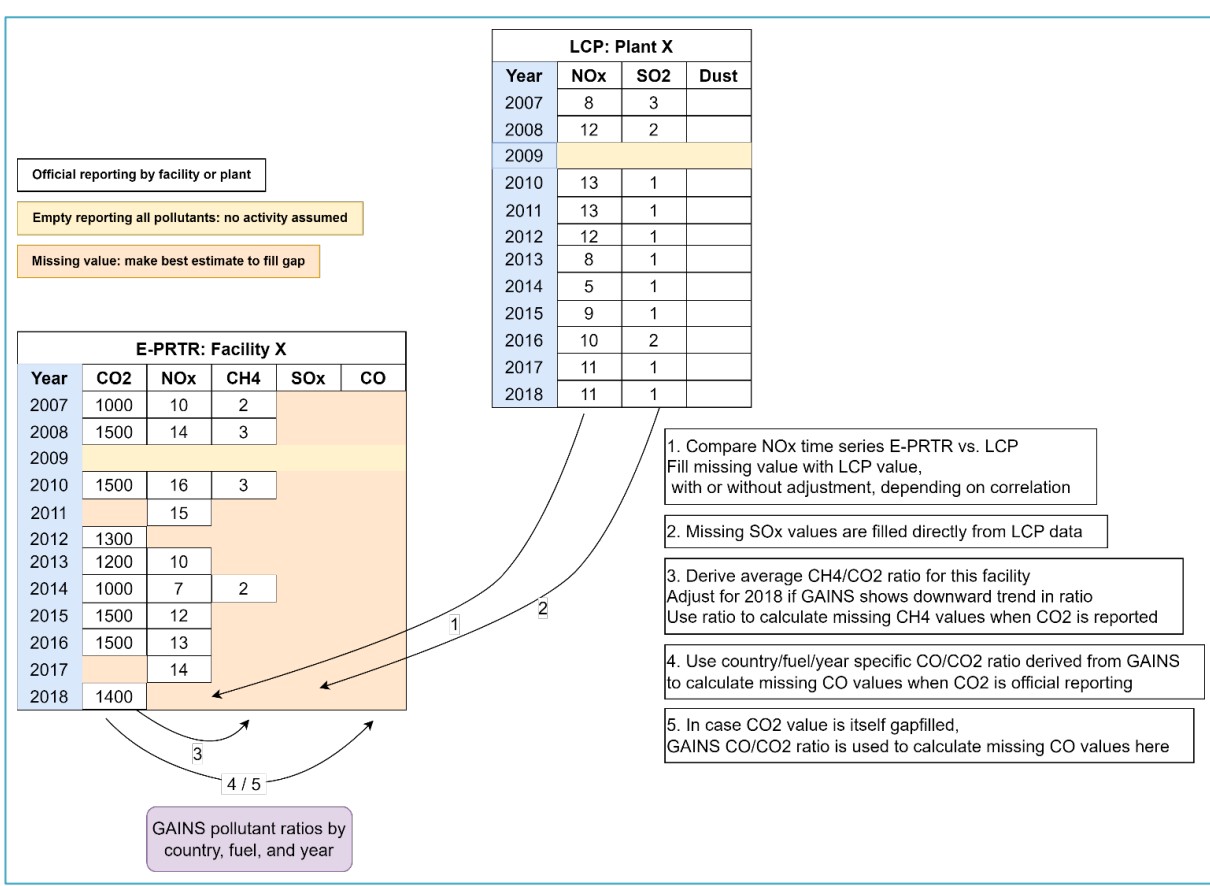

**Figure 1 Illustrated example of gap filling E-PRTR missing emission values for facility X**

As the gap filling steps progress, emission values filled by later steps are typically more uncertain. To limit outlier values, gap filled values derived from gap filling steps 3 to 5 for all pollutants except $CO_2$ were capped at the E-PRTR reporting threshold value (thus following the assumption that the value has not been originally reported due to being below the reporting threshold). Table 3 shows what share of the final emission has been derived from E-PRTR reporting and the subsequent gap filling steps (i.e., GF1 to GF5). The contribution of gap filled emissions is most substantial for CO and $CH_4$, with more than 60% of total emissions from gap filling. Table 4 shows for which percentage of power plant locations, the emission values have been gap filled. From this perspective, gap filling plays a more prominent role, as the highest emission values have typically been reported, and gap filling imputes mostly smaller emission values.

**Table 3: Contribution of reported and gap filled values to European power plant emissions, in terms of total emissions [kton/year**

| Source | $CO_2$ | NOx | SOx | CO | $CH_4$ |
|---|---|---|---|---|---|
| E-PRTR | 91% | 95% | 96% | 39% | 33% |
| GF1 | 3% | 3% | 2% | 0% | 0% |
| GF2 | 1% | 1% | 1% | 0% | 0% |
| GF3 | 3% | 1% | 0% | 11% | 7% |
| GF4 | 1% | 1% | 1% | 47% | 54% |
| GF5 | 0% | 0% | 0% | 4% | 6% |
| **Total emissions (kton)** | **1,069,862** | **843** | **1,018** | **465** | **66** |

**Table 4: Contribution of reported and gap filled values to European power plant emissions, in terms of the share of power plants**

| Source | $CO_2$ | NOx | SOx | CO | $CH_4$ |
|---|---|---|---|---|---|
| EPRTR | 56% | 52% | 21% | 6% | 2% |
| GF1 | 9% | 14% | 10% | 0% | 0% |
| GF2 | 29% | 28% | 50% | 0% | 0% |
| GF3 | 4% | 4% | 3% | 8% | 4% |
| GF4 | 3% | 3% | 12% | 80% | 88% |
| GF5 | 0% | 0% | 4% | 6% | 6% |
| **# power plants included** | **1,736** | **1,736** | **1,736** | **1,736** | **1,736** |

### 2.3.2 Non-European countries

Plant-specific $CO_2$, $NO_x$, $SO_2$ and $CH_4$ emissions for all US power plants were obtained from the eGRID database. Most
215 emissions of $CO_2$, $NO_x$ and $SO_2$ are taken from monitored data from the Clean Air Markets Division Power Sector Emission
Data. For all other units and for $CH_4$, the reported emissions are based on measured heat input multiplied by an emission factor,
as described in US EPA (2020). Emissions of CO, which are not reported by eGRID, were estimated using fuel-dependent
average ratios between $NO_x$ and CO emissions derived from the continuous emission monitoring system (CEMS) database
maintained by the Environmental Protection Agency (US EPA, 2021).
For the rest of the world, emissions per power plant were estimated following the steps below:

1. Estimation of $CO_2$ and $CH_4$ emissions per country, utility type (i.e., main or auto-producer plants) and fuel type
combining the national energy statistics provided by the IEA World Energy Balances (IEA, 2021a) with the Tier 1
fuel-dependent emission factors reported by the IPCC guidelines (Eggleston et al., 2006).
2. Estimation of NOx, $SO_2$ and CO emission from coal-, natural gas- and oil-fired power plants by combining the $CO_2$
annual emissions estimated in step 1 with fuel- and country/region-dependent average ratios between $CO_2$ emissions

and emissions of other pollutants (e.g., $SO_2/CO_2$ ratio) derived from the GAINS emission inventory (Amann et al., 2011; Klimont et al., 2017), which takes into account the heterogenous implementation of emission control restrictions in power plants across countries/regions. Emission ratios were constructed for a total of 23 non-EU countries/world regions, including China, India, South Africa, Japan and Australia, among others. The ratios were estimated as an average of the emissions reported by GAINS for the years 2015 and 2020, since they are the closest to the reference year of our catalogue (2018). Figure 2 shows a comparison between the $SO_2/CO_2$ and $NO_x/CO_2$ ratios for coal-fired power plants obtained for selected countries, indicating significant differences across them. The $SO_2/CO_2$ ratios estimated for Turkey and South Africa are approximately 17 and 10 times larger than the one estimated for China, the ratios reported for India and Australia being also considerably larger (i.e., between 5 and 7 times). The results are in line with differences across national emission legislations associated to the power generation industry. Emission standards for coal-fired power plants in China (200 mg·m$^{-3}$ for all existing plants and 35 mg·m$^{-3}$ for plants built after 2020) are much stricter than the ones established in Turkey (1000 mg·m$^{-3}$ in operation between 2004 and 2019), South Africa (680 mg·m$^{-3}$), India (600 mg·m$^{-3}$ for units commissioned before 2003, 200 mg·m$^{-3}$ for units commissioned between 2004 and 2016 and 100 mg·m$^{-3}$ for units after 2017) or Australia, where no national or state-wide limits exist. The estimated $NO_x/CO_2$ ratios also vary across countries, China being again the country reporting the lower value. As for $SO_2$, $NO_x$ emission limits for coal-fired power plants in China (100mg·m$^{-3}$ for plants built 2004-2011; 200mg·m$^{-3}$ for plants built before 2004) are stricter than the ones implemented in South Africa (1020mg·m$^{-3}$), Australia (856 mg·m$^{-3}$) and India (600 mg·m$^{-3}$ for units installed before 2003, 300 mg·m$^{-3}$ for units installed between 2004 and 2016 and 100 mg·m$^{-3}$ for units after 2017). The lower discrepancies among $NO_x/CO_2$ ratios when compared to $SO_2/CO_2$ ratios indicate that for $SO_2$ other elements than emission legislation may be also playing a role, such as the type and quality (e.g., sulphur content) of coal used in each country.

3. Estimation of NOx, $SO_2$ and CO emissions from biomass- and waste-fired power plants by combining the $CO_2$ annual emissions estimated in step 1 with fuel-dependent average ratios between $CO_2$ emissions and emissions of other pollutants (e.g., $SO_2/CO_2$ ratio) reported by the E-PRTR based European power plant database (see Sect. 2.3.1). Same ratios are assumed for all countries due to the lack of more detailed information. Despite introducing some uncertainty, it is important to note that the contribution of these two fuels to the total combustion-related electricity generation is rather residual (less than 5%; IEA, 2021a).

4. Estimation of CO emissions from biomass- and waste-fired power plants by combining the $NO_x$ annual emissions estimated in step 3 for these facilities with calculated fuel-dependent average ratios between $NO_x$ and CO emissions derived from the US EPA CEMS database.

5. Assignment of estimated country- and fuel-dependent emissions derived from step 1, 2, 3 and 4 to each facility as a function of the installed capacity and fuel information. The information on installed capacity per power plant is provided by the databases described in Sect. 2.1

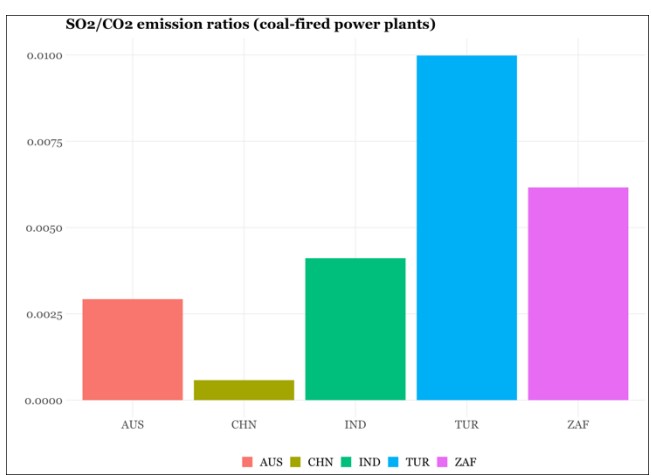
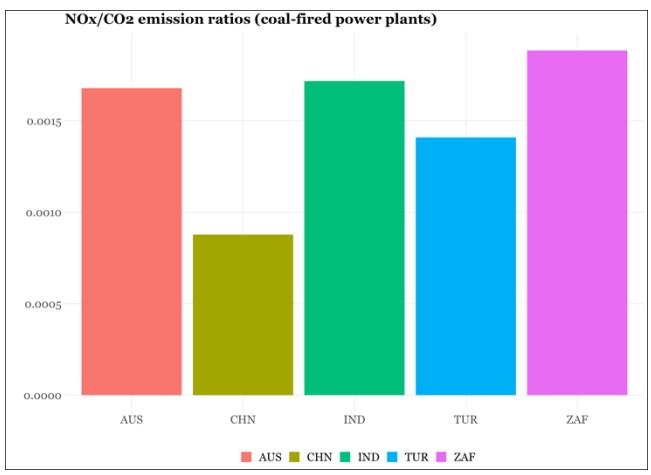

**Figure 2 SO₂/CO₂ and NOₓ/CO₂ emission ratios for coal-fired power plants estimated for selected countries using the GAINS inventory (Amann et al., 2011; Klimont et al., 2017)**

For coal-fired power plants we assumed that main and auto-producer facilities are correctly covered in all countries, as the GCPTv2021_01 database reports both public and industrial facilities. On the other hand, emissions from auto-producer plants

using oil, natural gas, biomass or waste were only considered in those countries where the difference between the total installed capacity (main plus auto-producers) reported by our database and UN (2021) was lower than 10%. For countries where this difference was larger than 10%, we assumed that our database is only covering main activity producer plants and therefore auto-producer emissions were excluded from the country-to-plant assignation process (step 4).

Figure 3 shows the relative differences between the total installed capacity reported by our database and the installed capacity

reported by UN (2021) for main producers (red rectangles) and main plus auto-producers (blue circles) for the top 50 non-European CO₂ emitting countries. For each country, the marker without the transparency effect indicates whether emissions from main producers plus auto-producers (e.g., China, USA, South Korea, Saudi Arabia) or only from main producers (e.g., India, Russia, Japan, Iran) were considered.

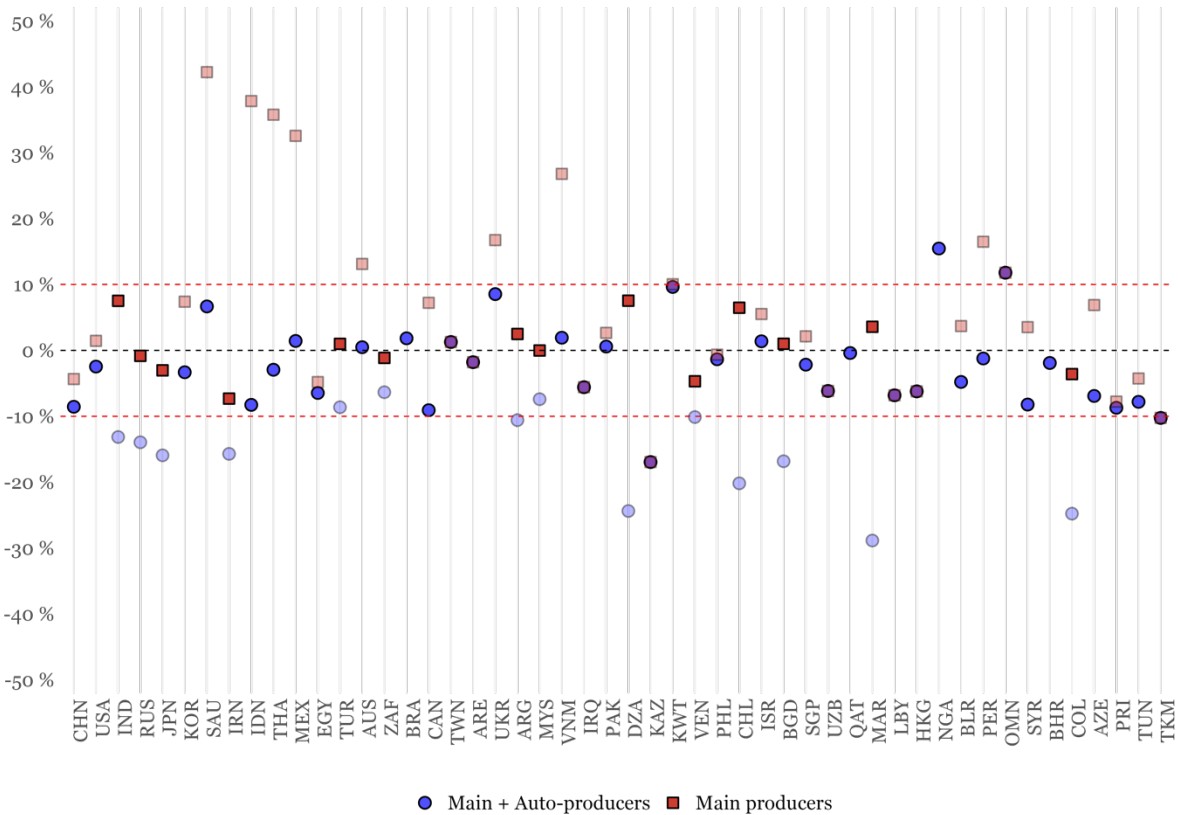

**Figure 3 Relative differences [%] in the total installed capacity reported by the global point source database and the installed capacity reported by UN (2021) for main producers (red rectangles) and main plus auto-producers (blue circles) for the top 50 non-European CO₂ emitting countries. For each country, the marker without the transparency effect indicates whether emissions from main producers plus auto-producers or only from main producers were considered.**

Overall, we could not include emissions from auto-producers in 35% of the countries considered. This translates into 4.1% of total estimated $CO_2$ emissions from the power sector that could not be allocated to the final non-European point source database due to the lack of information from auto-producers. Figure 4 represents the share of total national $CO_2$ emissions that could not be allocated per country. It is observed that most of the countries where information on auto-producers could not be found are in South America and Africa. Benin, El Salvador, Mali, Ecuador, Costa Rica and Madagascar are among the countries where the largest share of total $CO_2$ emissions remained unallocated (between 70% and 50%). Emissions from these countries are however not significant and therefore they have a very limited impact on the overall non-allocated emissions. In large emitting countries such as Russia, India or Japan, the share of national emissions that could not be assigned to individual facilities is much lower (i.e., 14% to 21%).

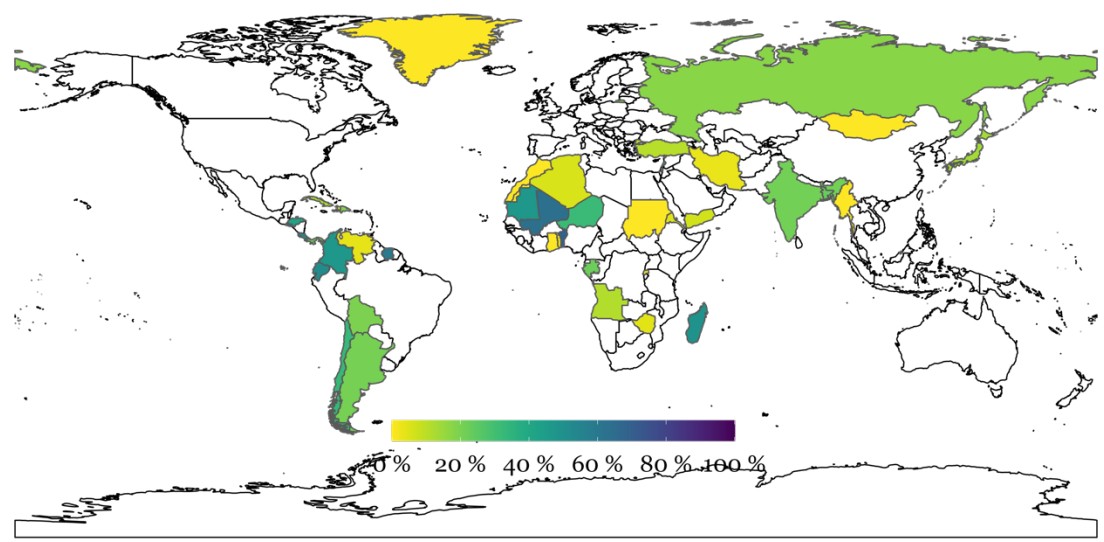

**Share of non-allocated CO2 emissions**

**Figure 4 Share of total national $CO_2$ emissions [%] from the power sector that could not be allocated due to the lack of information from auto-producers. Countries where emissions from main producers and auto-producers could be allocated are represented in white.**

## 2.4 Temporal profiles

Country- (state- for the US) and fuel-dependent monthly, weekly and hourly temporal profiles were constructed for all power plants (i.e., European and non-European datasets) using the electricity production statistics summarised in Table 5. For countries where electricity generation statistics are not disaggregated by fuel type, we assumed the same temporal distribution for all types of power plants. For countries with no information on electricity generation, or information only available at e.g., monthly scale but not at hourly scale, averaged profiles from countries belonging to the same world region were used. The definition of world regions was taken from the EDGARv5 emission inventory (Crippa et al., 2018). The resulting profiles were assigned to each facility as a function of the country and fuel type information.

**Table 5 Sources of electricity production statistics and corresponding characteristics**

| Country/Region | Source of information | Temporal resolution | Information per fuel |
|---|---|---|---|
| Uruguay | ADME (2021) | Hourly | yes |
| Australia | AEMO (2021) | Hourly | yes |
| Guatemala | AMM (2021) | Daily | yes |
| Indonesia | BPS (2021) | Monthly | no |
| Argentina | CAMMESA (2021) | Daily | yes |
| Mexico | CENACE (2021) | Hourly | yes |

| | | | |
|---|---|---|---|
| Algeria, Botswana, Lebanon, Malawi, Sri Lanka, Qatar | CEIC Data (2021) | Monthly | no |
| Chile | CNE (2021) | Hourly | yes |
| Peru | COES (2021) | Daily | thermal/renewable |
| United Arab Emirate | DEWA (2021) | Monthly | yes |
| EU27 + UK | ENTSO-E (2021) | Hourly | yes |
| Thailand | EPPO (2021) | Monthly | yes |
| South Africa | ESKOM (2022a) | Hourly | yes |
| Malaysia | GSO (2021) | Monthly | yes |
| China, Canada, Colombia, South Korea, New Zealand | IEA (2021) | Monthly | yes |
| Kazakhstan | KOREM (2021) | Monthly | thermal/renewable |
| Kuwait | MEW (2021) | Monthly | no |
| Moldova | MOLDELECTRICA (2021) | Hourly | no |
| Oman | NCSI (2021) | Monthly | yes |
| India | NPP (2021) | Daily | yes |
| Japan [*] | OCCTO (2021) | Hourly | thermal/biomass/renewable |
| Brazil | ONS (2021) | Hourly | yes |
| Bangladesh | PGCB (2021) | Hourly | yes |
| Russia | SO-UPS (2021) | Monthly | thermal/renewable |
| Switzerland [*] | SWISSGRID (2021) | Hourly | no |
| Turkey | TEIAS (2021) | Daily | yes |
| Ukraine | UNEC (2021) | Hourly | yes |
| USA | US EPA (2021) | Hourly | yes |
| [*] Monthly data derived from IEA as it is reported by fuel type | | | |

Figure 5 illustrates, on the one hand, the countries for which specific monthly, weekly and hourly profiles were constructed based on the statistics compiled and, on the other hand, the resulting share of total $CO_2$ emissions for which specific monthly, weekly and hourly profiles were available. For the monthly profiles, the database constructed is covering a total of 96 countries plus 42 USA states, which translates into more than 90% of total $CO_2$ emissions from the power sector. For weekly and hourly profiles, the coverage in terms of total $CO_2$ emissions is much lower (approx. 46% and 36%, respectively) partially because no information on electricity production and the daily and hourly level was available for China. For this country, we assumed that the weekly cycle of emissions follows the pattern obtained for India, which shows no significant difference between weekdays and weekends. This assumption is in line with the results found by Wu et al. (2022), in which weekly profiles for Chinese power plants were constructed using measured emissions derived from continuous emission monitoring systems.

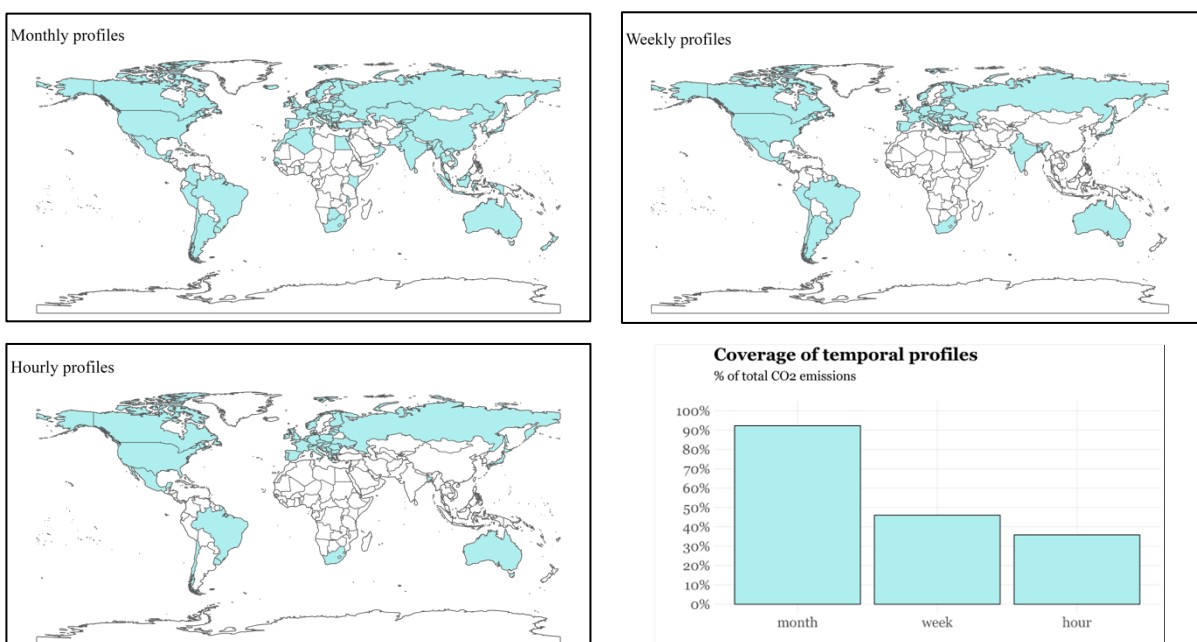

**Figure 5 Spatial coverage of the constructed monthly, weekly and hourly temporal profile databases. Share of total CO$_2$ emissions [%] from the power sector for which specific monthly, weekly and hourly profiles were developed.**

## 2.5    Vertical profiles

Hourly effective emission heights at the facility level were simulated by combining 2018 global hourly gridded meteorological information (i.e., air temperature at stack height, wind speed at stack height, surface temperature, boundary-layer height, friction velocity and Obukhov length) simulated by the MONARCH atmospheric chemistry model at 0.3x0.3 deg. (Badia et al., 2017) with facility-level stack parameter information (i.e., height, diameter, exit velocity and exit temperature). Information on stack parameters were obtained from the following sources:

- The point source database of electric generation units (PTEGU), obtained from the US EPA emission modelling platform (US EPA, 2021), which reports plant-level stack parameter information for USA power plants.
- The HERMES Spanish power plant database (Guevara et al., 2013)
- Atmospheric emission licences of South African power plants (CER, 2022)
- The list of tallest chimneys worldwide reported by Wikipedia (2022a)
- The list of tallest chimneys in Poland reported by Wikipedia (2022b)
- The list of tallest chimneys in Czech Republic reported by Wikipedia (2022c)
- The list of tallest structures in Germany reported by Wikiwand (2022)

The Indian Ministry of Environment, Forest and Climate Change (MoEFCC, 2015) requires all coal-fired power plants with generation capacity of 500 MW and above to build a stack of minimum 275m; those between 210 MW and 500 MW to build a stack of minimum 220 m; and those with less than 210 MW to build a stack based on the estimated SO$_2$ emissions rate (Q in

kg/hr) and a thumb rule of height = 14*(Q)0.3. Considering this information, we assumed that all coal-fired power plants in India with a generation capacity of 500 MW and above had a stack height of 275m, and those between 210 MW and 500 MW a stack height of 220m.

In some European coal -fired power plants built in recent years, which must be equipped with a flue gas cleaning system, the cooling tower also takes on the function of the chimney. Original chimneys were dismantled and now emissions are released

through the cooling towers, which have different stack conditions. For Germany, we identified the list of power plants with cooling towers used as chimneys and associated stack height through Wikipedia (2022d), and we completed the information with the stack diameter, exit temperature and exit velocity reported by Brunner et al., (2019). This level of detail is not considered in facilities from other countries due to lack of information.

Fuel-dependent and $CO_2$ emission-weighted average stack parameters were calculated using the PTEGU dataset and assigned

to all those facilities for which no specific information was found. For waste-to-energy power plants we considered the stack parameters reported by Pregger and Friedrich (2009) as the PTEGU dataset does not include this type of facility. Table 6 summarises the stack parameters proposed per fuel type and the associated number of units considered to calculate the values.

**Table 6 Fuel-dependent and $CO_2$ emission-weighted average stack parameters assigned to facilities with no specific information and number of sources considered to calculate them.**

| Fuel | Stack height [m] | Stack diameter [m] | Exit temperature [ºC] | Exit velocity [m/s] | N units |
|---|---|---|---|---|---|
| Coal | 182.6 | 7.7 | 91.8 | 21.0 | 675 |
| Natural gas | 53.0 | 5.6 | 143.5 | 20.0 | 1800 |
| Oil | 125.7 | 5.5 | 122.6 | 20.7 | 74 |
| Biomass | 72.6 | 2.8 | 147.6 | 28.5 | 33 |
| Waste | 103 | 2.5 | 118 | 8.5 | 230 |


Figure 6 illustrates, on the one hand, the facilities assigned with specific (red circles) or emission-weighted averaged (white circles) stack height information and, on the other hand, the share of total $CO_2$ emissions from the power sector assigned with specific stack parameter information. In terms of emission coverage, only 28% of total $CO_2$ emissions from the power sector are assigned with specific stack height values. This share significantly varies across world regions. In USA, South Africa and

India the share is between 75% and 90%, while in Central Europe is around 50%. In many Asian, African and South American regions the share is below 5%. The coverage of total $CO_2$ emissions for stack diameter, exit velocity and temperature is even lower than for the stack height parameter (i.e., approx. 15% globally in all cases), the differences between regions being equally heterogeneous. These results indicate the current lack of stack parameters information.

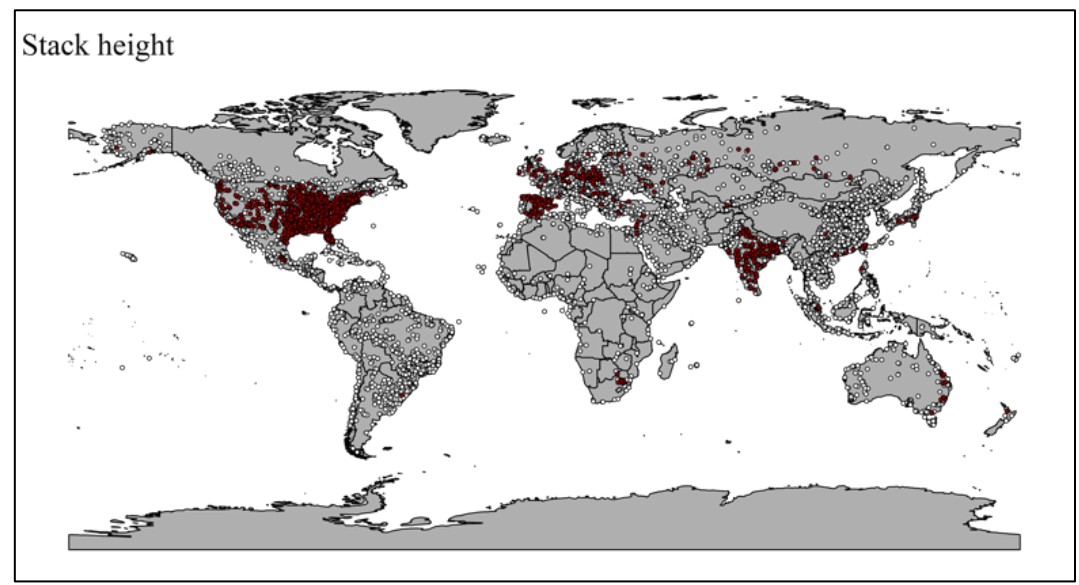

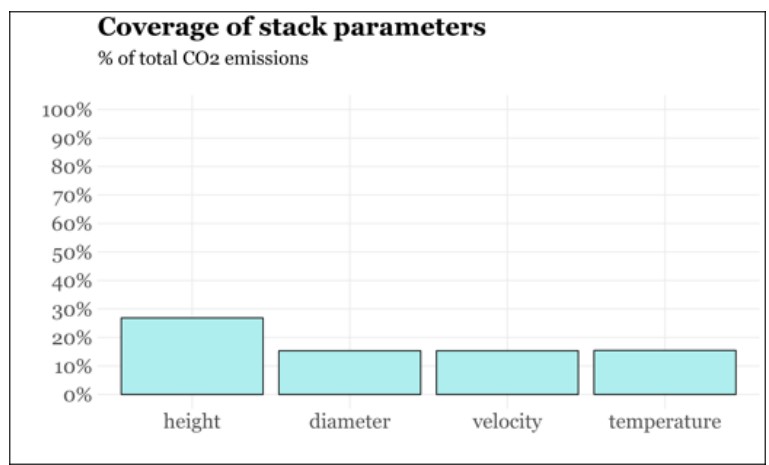

**Figure 6 Facilities assigned with specific (red circles) or emission-weighted averaged (white circles) stack height information (top) and share of total CO₂ emissions [%] from the power sector for which specific stack parameters (height, diameter, exit velocity and exit temperature) were assigned (bottom).**

The plume rise calculations at the hourly and facility level were performed using the High-Elective Resolution Modelling Emission System version 3 (HERMESv3) bottom-up emission system (Guevara et al., 2020), which includes plume rise formulas as described by Gordon et al. (2018). The HERMESv3 system was used to break down facility-level annual emissions into hourly resolution using of the temporal profiles described in Sect. 2.4, and to estimate hourly effective emission heights per plant considering the meteorological information provided by the nearest grid cell of MONARCH. Hourly plume top and plume bottom values per facility ($h_{top}(h, f)$, $h_{bot}(h, f)$) were derived from the estimated effective emission heights following the expressions reported by Bieser et al. (2011) (Eq. 1 and 2):

$$h_{top}(h, f) = h_s(f) + 1.5 * \Delta h(h, f) \tag{1}$$

$$h_{bot}(h, f) = h_s(f) + 0.5 * \Delta h(h, f) \tag{2}$$

where $h_s(f)$ is the stack height of the facility $f$ and $\Delta h(h, f)$ is the modelled effective emission height for the facility $f$ and

hour $h$.

# 3 Results

## 3.1 Annual emissions

Figure 7 and 8 show the plant-level $CO_2$ and $NO_x$ annual emissions as reported by the resulting global point source database. Results are distinguished by fuel type. It is observed that coal-fired power plants (red circles) are the main contributors to total $CO_2$ emissions, the top emitters being in China, India, US, Australia, South Africa, Central Europe and Indonesia. $CO_2$ emissions from natural gas power plants (blue circles) are dominant in Russia and some countries from the Middle East (e.g., Saudi Arabia and Iran). For $NO_x$, main contributors are also coal-fired power plants, but several oil-fired power plants (black circles) gain importance when compared to their contributions in the $CO_2$ emissions map, especially in the Middle East (i.e., Iran and Saudi Arabia), Indonesia, Venezuela and some countries in Northern Africa. In China, India, US, Australia, South Africa and Central Europe $NO_x$ emissions are mainly dominated by coal-fired power plants. For both pollutants it is observed that the number of large emitters in Africa and South America is rather scarce, expect for South Africa and some countries in North Africa as well as Venezuela. This is related to the fact that in both regions the electricity production is mainly dominated by renewable sources (e.g., hydro, solar) (IEA, 2021). Linked to this aspect, it is interesting to see the large amount of biomass power plants in Brazil (brown circles), as this fuel represents the second largest energy source in the country, just behind hydropower. A significant number of waste-to-energy plants (green circles) are reported in Japan and China, the two countries with the largest installed incineration capacity (Lu et al., 2017).

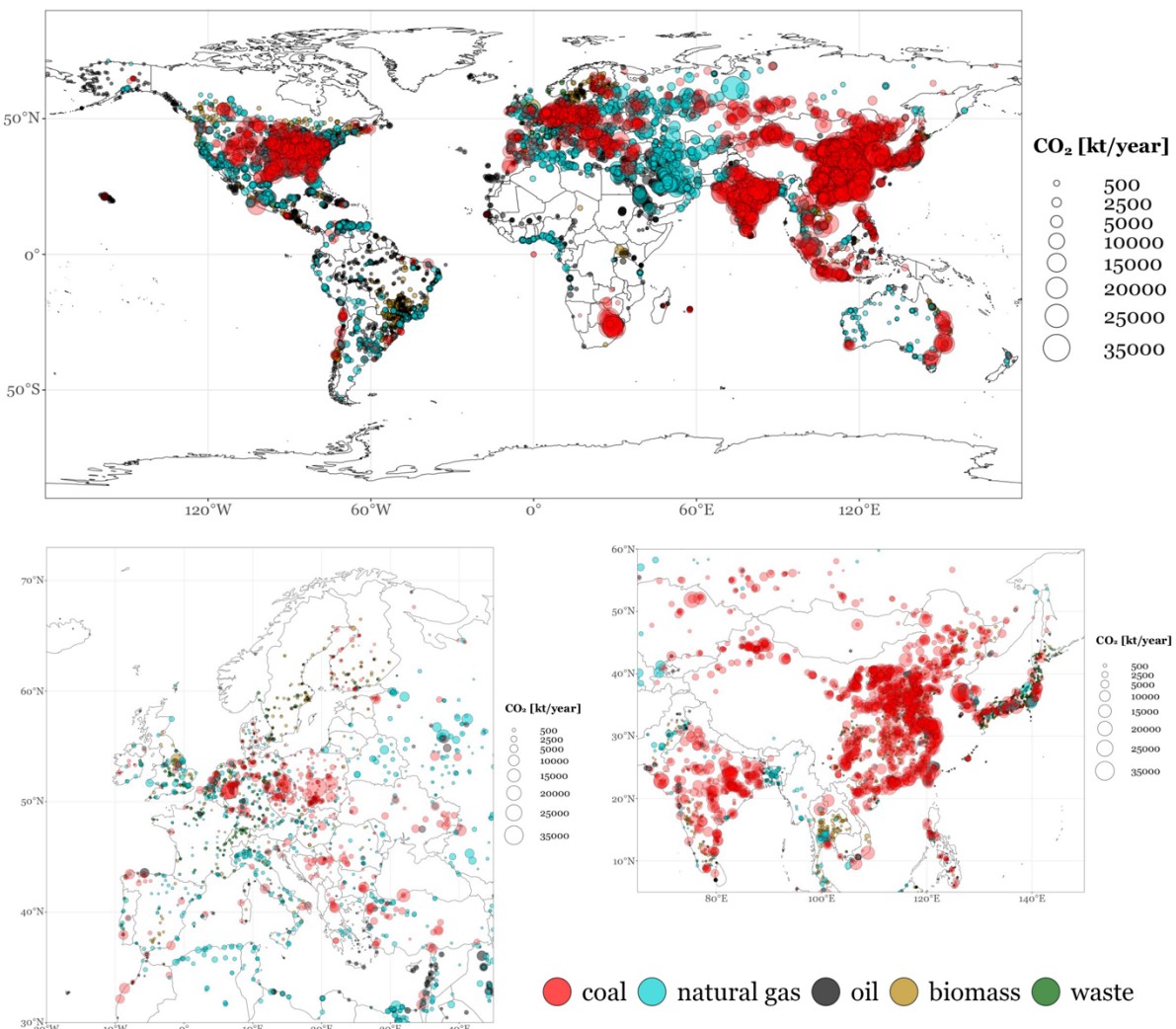


**Figure 7 Plant-level CO$_2$ annual emissions [kt/year] as reported by the resulting global point source database, including zooms over Europe and Asia. Emissions are colour-classified according to the main fuel used: coal (red), natural gas (blue), oil (black), waste (green) and biomass (brown).**

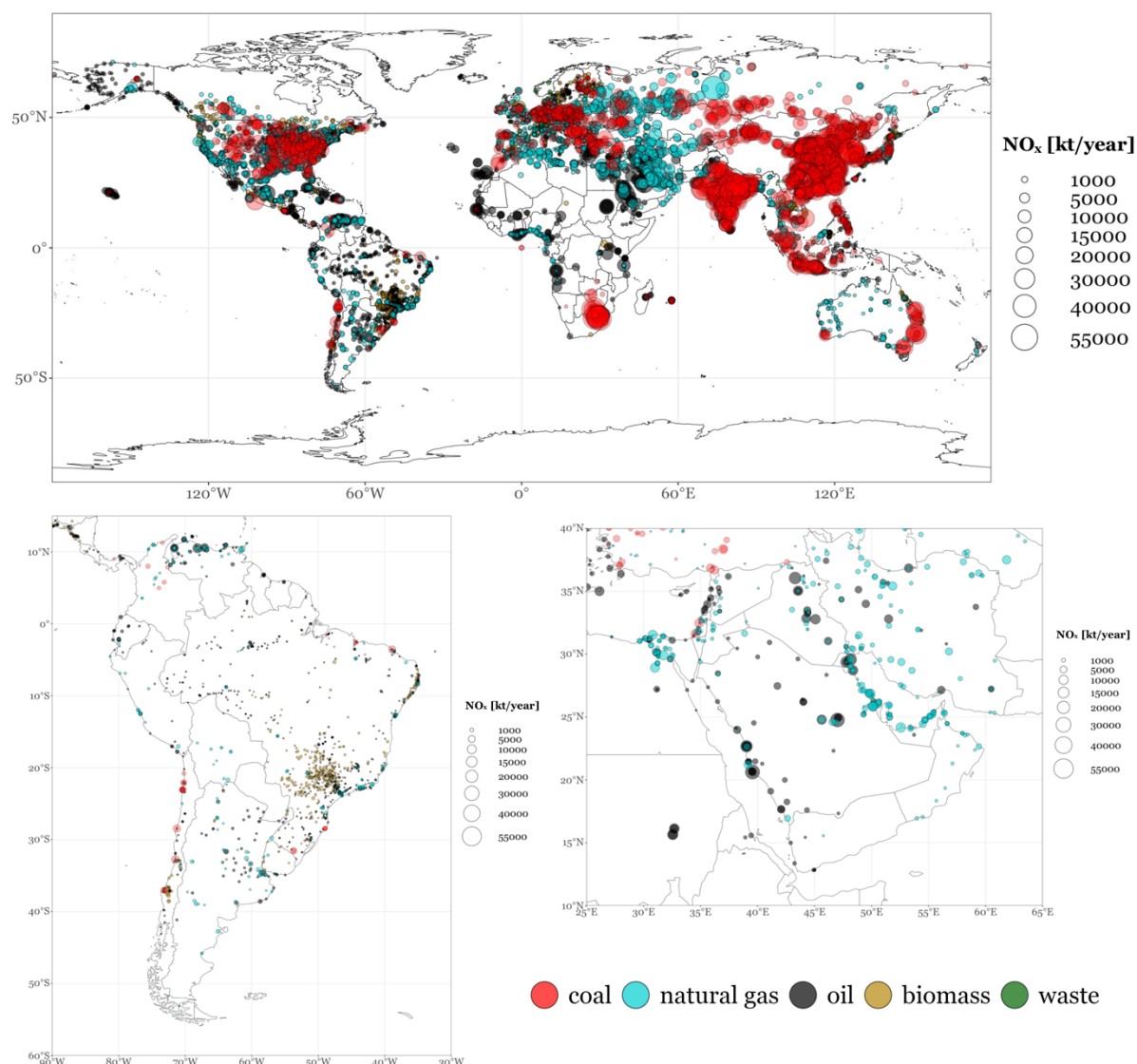


**Figure 8 Plant-level NO$_x$ annual emissions [kt/year] as reported by the resulting global point source database, including zooms over South America and the Middle East. Emissions are colour-classified according to the main fuel used: coal (red), natural gas (blue), oil (black), waste (green) and biomass (brown).**

Table 7 and Table 8 list the top 15 CO$_2$ and NO$_x$ emitting power plants worldwide and in EU27+UK.

At the global level, the Belchatów (Poland), Taean (South Korea), Taichung (Taiwan), Dangjin (South Korea) and Datang Tuoketuo (China) power plants are the top 5 CO$_2$ emitters. These five facilities are also the five largest coal-fired power stations in the world (with installed capacities between 6700MW and 5300 MW). All top 15 CO$_2$ emitters are coal-fired power plants, except for Surgutskaya GRES-2 (Russia), which is the largest combined-cycle natural gas-fired power station of Russia

(8865MW) and supplies energy to nearly 40% of the population. Most of the top 15 $CO_2$ emitters are in Asian countries, including: South Korea (3), China (2), Taiwan (2), Malaysia (2), India (1) and Kazakhstan (1), while the rest are in Europe: Germany (2), Poland (1) and Russia (1). Seven out of the 10 top emitters identified in this work are also listed in the 2018 top ten $CO_2$ polluting power plants reported by Grant et al., (2021). At the EU27+UK, it is observed that most of the 15 top $CO_2$ emitters are in Germany (6) and Poland (3). Similarly to what is observed at the global scale, 14 out of 15 facilities are coal-

fired power plants, the remaining worst polluter being the Drax biomass power station, the largest power plant in the UK (3906MW) that is also capable of co-firing petroleum coke. The largest emitter in EU27+UK (Belchatów, Poland) reports almost 5 times more $CO_2$ emissions than the fifteenth facility (As Pontes, Spain).

For NOx, the list of top emitters mainly consists of coal-fired power plants (14 out of 15). Seven of these plants appear in both the $CO_2$ and $NO_x$ top 15 emitters lists, including Surgutskaya GRES-2 (Russia), Taean (South Korea), Dangjin (South Korea),

Manjung (Malaysia), Yeongheun (South Korea), Ekibastuz-1 (Kazakhstan), Vindhyachal (India). Concerning the other top 15 emitters, six of them are located in South Africa and two in India. At the EU27+UK level, Belchatów is again the largest emitter. Four out of the top five emitters are in Germany, all of them being coal-fired power plants. There are also four Spanish facilities, three of them being oil-fired internal combustion engines located in the Canary Islands. The other non-coal facilities that complete the European top 15 list are Drax (UK) and Atherinolakkos (Greece), the later also being operated with diesel

engine units. Additional information on the total emissions obtained at the country level is provided in Sect. 3.2.

**Table 7 List of top 15 $CO_2$ [kt/year] and $NO_x$ [t/year] emitting power plants worldwide.**

| Plant | Fuel | Country | $CO_2$ [kt/year] | Plant | Fuel | Country | $NO_x$ [t/year] |
|---|---|---|---|---|---|---|---|
| Belchatów | coal | POL | 38400 | Taean | coal | KOR | 58256 |
| Taean | coal | KOR | 35877 | Ekibastuz-1 | coal | KAZ | 55122 |
| Taichung | coal | TWN | 34499 | Dangjin | coal | KOR | 54979 |
| Dangjin | coal | KOR | 33859 | Vindhyachal | coal | IND | 47126 |
| Datang Tuoketuo | coal | CHN | 31435 | Majuba | coal | ZAF | 46682 |
| Manjung | coal | MYS | 30418 | Kendal | coal | ZAF | 46377 |
| Neurath | coal | DEU | 29900 | Yeongheung | coal | KOR | 46241 |
| Yeongheung | coal | KOR | 28477 | Mundra (Adani) | coal | IND | 45740 |
| Niederaussem | coal | DEU | 27200 | Matimba | coal | ZAF | 44958 |
| Surgutskaya GRES-2 | natural gas | RUS | 25640 | Surgutskaya GRES-2 | natural gas | RUS | 44548 |
| Ekibastuz-1 | coal | KAZ | 25522 | Lethabo | coal | ZAF | 41780 |
| Vindhyachal | coal | IND | 24733 | Tutuka | coal | ZAF | 41172 |
| Waigaoqiao | coal | CHN | 24512 | Manjung | coal | MYS | 40622 |
| Mailiao | coal | TWN | 24463 | Matla | coal | ZAF | 40563 |
| Tanjung Bin | coal | MYS | 24068 | Tata Mundra | coal | IND | 39602 |

**Table 8 List of top 15 CO₂ [kt/year] and NOₓ [t/year] emitting power plants in EU27 + UK.**

| Plant | Fuel | Country | CO₂ [kt/year] | Plant | Fuel | Country | NOₓ [t/year] |
|---|---|---|---|---|---|---|---|
| Belchatów | coal | POL | 38400 | Belchatów | coal | POL | 30100 |
| Neurath | coal | DEU | 29900 | Neurath | coal | DEU | 20200 |
| Niederaussem | coal | DEU | 27200 | Jänschwalde | coal | DEU | 19000 |
| Jänschwalde | coal | DEU | 24000 | Niederaussem | coal | DEU | 18000 |
| Eschweiler | coal | DEU | 19100 | Kraftwerk Boxberg | coal | DEU | 13500 |
| Kraftwerk Boxberg | coal | DEU | 19100 | Eschweiler | coal | DEU | 13000 |
| Drax | biomass | GBR | 16600 | Drax | biomass | GBR | 12200 |
| Kozienice | coal | POL | 14100 | Punta Grande | oil | ESP | 11200 |
| Lippendorf | coal | DEU | 11400 | Atherinolakkos | oil | GRC | 10700 |
| Maritsa East 2 | coal | BGR | 9574 | Kozienice | coal | POL | 9650 |
| Agioy Dhmhtrioy | coal | GRC | 9230 | Las Salinas | oil | ESP | 8220 |
| Enea Połaniec | coal | POL | 8220 | Enea Połaniec | coal | POL | 7760 |
| Eemshaven | coal | NLD | 8210 | Agioy Dhmhtrioy | coal | GRC | 7100 |
| Torrevaldaliga Nord | coal | ITA | 8081 | Granadilla | oil | ESP | 7030 |
| As Pontes | coal | ESP | 7940 | As Pontes | coal | ESP | 6360 |

## 3.2 Comparison with independent inventories

The estimated annual emissions were compared against other independent plant- and country-level inventories. The following subsections present and discuss the results.

### 3.2.1 Plant level

Estimated plant level emissions were compared against information reported by the CARMAv3 global database. As mentioned in Sect. 1, and despite not being longer maintained, the CARMAv3 database is still used as a proxy for the spatial representation of power plant emissions in several state-of-the-art inventories and modelling systems, like the EDGAR inventory (Janssens-Maenhout et al., 2019) and the Carbon Cycle Fossil Fuel Data Assimilation System (CCFFDAS) (Asefi-Najafabady et al., 2014).

Table 9 summarises the comparison between total number of power plants and associated CO₂ emissions reported by CARMAv3 and this work for selected countries, including China, the United States, India, Germany, South Korea, South Africa, Australia, Taiwan and Poland. For China the present work reports 79% more facilities and 92% more emissions than CARMAv3. This result is in line with the fact that CARMAv3 was build using information from 2009, and during the last decade the number of power plants in China and associated emissions has significantly increased (IEA, 2023). For USA the number of plants reported by each database is almost the same (-1%) but emissions are lower in this work when compared to CARMAv3 (-17%). This difference is mainly related to the transition from coal to natural gas and renewables that occurred during the last decade (EIA, 2021). Greenhouse gas emissions for electricity generation from natural gas are generally lower than those from oil and coal due to a more beneficial heat per carbon density and higher combustion efficiencies (e.g., IPCC, 2011). For Germany, South Africa, Poland and Australia it is observed that despite including less facilities (differences between -47% and -63%), total CO₂ emissions reported by this work are generally in line with CARMAv3 values (differences between

440 -12% and 0%). This is because CARMAv3 is mostly based on Platts WEPP (Platts, 2015), which contains many small size auto-producer units (e.g., boilers located in commercial and institutional buildings such as hospitals or airports) with very low emission levels associated to them that are not considered in the present work. Moreover, and as shown below, CARMAv3 includes power plants that are not currently operating as they were shut down during the last decade. For India the present catalogue reports 81% more emissions than CARMAv3 despite including -30% less facilities, which indicates that the

445 additional plants considered in CARMAv3 are low-level emission small plants. This hypothesis is confirmed when comparing the median $CO_2$ annual emission values of each dataset, the one reported by the present catalogue (154669 kt $CO_2 \cdot$ year$^{-1}$) being almost 18 times larger than CARMAv3 (8839 kt $CO_2 \cdot$ year$^{-1}$). Differences between the present database and CARMAv3 are also linked to the fact that emissions reported by CARMAv3 exclude $CO_2$ from biofuels, while the present catalogue includes solid biomass-fired power plants.

**Table 9 Comparison between total number of facilities and associated CO$_2$ emissions [kt·year$^{-1}$] reported by CARMAv3 and this work for selected countries (China, the United States, India, Germany, South Korea, South Africa, Australia, Taiwan and Poland.**

| ISO3 | Number of plants | | | CO$_2$ [kt/year] | | |
|------|-----------|---------|------|-----------|---------|------|
| | this work | CARMAv3 | diff | this work | CARMAv3 | diff |
| CHN | 1744 | 977 | 79% | 4732145.0 | 2469937.5 | 92% |
| USA | 2847 | 2866 | -1% | 1928603.6 | 2315648.5 | -17% |
| IND | 450 | 641 | -30% | 1185786.9 | 653460.9 | 81% |
| DEU | 365 | 997 | -63% | 298746.1 | 297996.3 | 0% |
| KOR | 113 | 107 | 6% | 294318.4 | 213915.2 | 38% |
| ZAF | 22 | 43 | -49% | 224744.1 | 224515.0 | 0% |
| AUS | 190 | 368 | -48% | 188235.9 | 215089.9 | -12% |
| TWN | 58 | 92 | -37% | 161218.5 | 111306.1 | 45% |
| POL | 150 | 282 | -47% | 146717.6 | 148787.1 | -1% |

Figure 9 shows a plant-to-plant comparison between the top 20 emitters reported by this work and CARMAv3 for selected countries (i.e., United States, Taiwan, South Africa and Poland). In all of them it is observed that CARMAv3 reports emissions for plants that are not included in the present catalogue as they are currently retired or not operating (e.g., Jenwu power station in Taiwan, Adamow power station in Poland). Except for the case of United States (i.e., the Monticello Steam Electric Station), most of these plants were already reporting low emissions in 2009, which could indicate that they were already in the process of being disconnected from the grid. The good agreement in South Africa, Poland and Taiwan (R$^2$ between 0.86 and 0.97) indicates that the level of emissions from the top emitters in these countries remained stable between 2009 and 2018 (e.g., Kendal power plant in South Africa, Belchatów power station in Poland). On the contrary, significant discrepancies are observed in the United States (R$^2$ = 0.41), the results reported by this work being consistently lower than CARMAv3 for all top emitters. As mentioned before, these differences are mainly driven by the reduction of the rate of utilization of coal-fired power plants and the conversion of coal-fired plants to natural gas during the last decade. Note that for most of the emissions in the USA (99%), European Union (63%), Canada (96%), India (78%) and South Africa (91%), the plant-level CO$_2$ reported by CARMAv3 was directly obtained from official disclosure databases such as E-PRTR or the US EPA Clean Air Markets, which are also considered in the present work, while for the rest of countries plant-level emissions were computed considering estimated key variables (i.e., capacity factor, heat rate and CO$_2$ emission factors) using statistical models fitted to a detailed dataset of USA facilities (Wheeler and Ummel, 2008). This emission estimation approach is substantially different from the one considered in the present work (see Sect. 2.3.2) and could also contribute to the discrepancies obtained between datasets, beside the differences in the year of reference mentioned above.

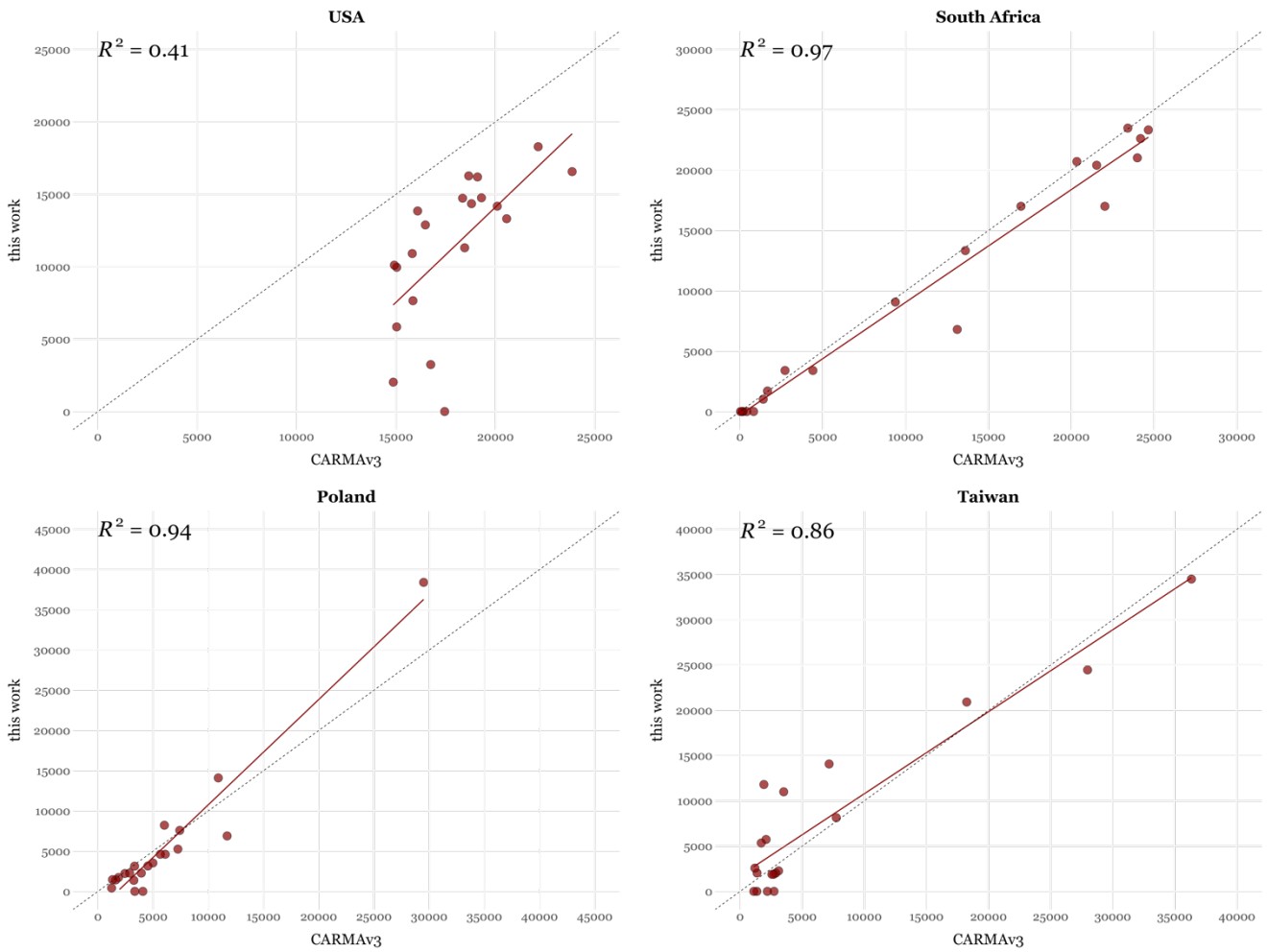

**Figure 9 Plant-to-plant CO$_2$ annual emission comparison between top 20 emitters reported by this work and CARMAv3 for United States, Taiwan, South Africa and Poland (dashed line represents the 1:1 line).**

Besides comparing total annual emissions, we also compared the geographical location reported by the present catalogue and CARMAv3 for each one of the top 20 emitters in the nine countries listed in Table 9. We found 6 facilities in which the location reported by CARMAv3 was off by hundreds of kilometres (between 120km and 337km), while in 24 cases the locations provided by CARMAv3 were displaced from the right coordinates by tens of kilometres (between 11km and 79km). Most of these cases (22 out of 30) correspond to units in Asia (i.e., China, Taiwan and India) where the wrongly allocated CARMAv3 power plants tend to be assigned to nearby city centres (Fig. S1). The differences between locations reported in this work and CARMAv3 are much lower when looking at European (Poland and Germany) and USA facilities, where the average distance between geographical coordinates reported by each dataset is of approximately 600m, and the maximum discrepancy is of 5.7km. These findings are consistent with the methods used in CARMA to add geographic data to the power plants. For about

70% of the CARMAv3 power plants, the geocoding is performed using an algorithm that derive city-center latitude and longitude coordinates from the geopolitical data (i.e., country, state/province, and city names) provided by WEPP. On the other hand, for the facilities located in Europe, USA and Canada (approximately 6000), exact geographical coordinates were obtained from high resolution disclosure databases and manual geocoding. The geographical dislocation of CARMAv3 facilities described here for Asia is consistent with other recent investigations (e.g., Zhang et al., 2022).


Figure 10 shows a comparison between plant-level $CO_2$, annual emissions [kt/year] estimated by this work and reported by the GIDv1.1 database. Results are shown for the top 50 emitters reported by each inventory (a total of 76 facilities). Overall, total emissions are almost equal (differences of -0.3%). However, important discrepancies are observed at the plant level. In 41 out of the 76 facilities the differences between reported annual $CO_2$ emissions is between +-25%, with larger discrepancies being

observed for the rest of the power plants. The $CO_2$ emissions estimated for the Bełchatów coal-fired power plant (Poland) in this work (38400 kt/year) are 2.75 larger than the results reported by GIDv1.1 (14051 kt/year). The emissions estimated in this work are in line with the results reported by Grant et al. (2021) for the same facility (i.e., 37600 kt/year), both studies suggesting that Bełchatów is the top one $CO_2$ emitter worldwide. Important discrepancies are also observed in several German coal-fired power plants (i.e., Niederaußem, Neurath, Weisweiler, Kraftwerk Boxberg and Jänschwalde), in which emissions reported by

this work are between 2 and 2.6 larger than GIDv1.1 results. Part of these discrepancies are probably related to the fact that the GIDv1.1 inventory is based on 2019 activity data, while the present work considers 2018 as a reference year. Despite differing only by one year, quick decarbonization efforts may be playing an important role in the resulting emissions. Following with the example of Germany, the total amount of coal used in this country to produce electricity decreased -24% between 2018 and 2019 (IEA, 2023). This fact is in line with the results reported by the integrated Industrial Reporting Database v.7

(EEA, 2022), which indicates that $CO_2$ emissions in Niederaußem, Neurath and Jänschwalde coal-fired power plants where 1.4 larger in 2018 when compared to 2019. The $CO_2$ emissions reported for Niederaußem by this work coincide with the value estimated by Grant et al. (2021) (27200 kt/year).

The result of this comparison also shows power plants for which the emissions estimated by this work are much lower than

the results reported by GIDv1.1. This is the case, for instance, of the Rajiv Gandhi coal-fired (India) and Shin Sakaiko liquefied natural gas-fired (Japan) power plants, where emissions are 0.3 and 0.15 times the ones reported by GIDv1.1, respectively. For the Rajiv Gandhi coal-fired power plant, the results reported by this work (6235 kt/year) are larger than the facility-level emissions reported by the Central Electricity Authority (3557 kt/year; CEA, 2022), indicating that GIDv1.1 (19979 kt/year) may be overestimating the emissions in this facility. Concerning the Shin Sakaiko power plant, no independent values could

be found to compare against the results estimated by this work and GIDv1.1. However, and based on the information of installed capacity, we hypothesise that GIDv1.1 emissions reported for this plant are also overestimated. According to GIDv1.1, the Shin Sakaiko power plant, is the 14[th] top $CO_2$ emitter worldwide despite having an installed capacity of 2000MW,

which is more than two times lower than the capacity of the Futtsu power plant (5040MW), the first (fourth) largest gas-fired power station in Japan (the world).

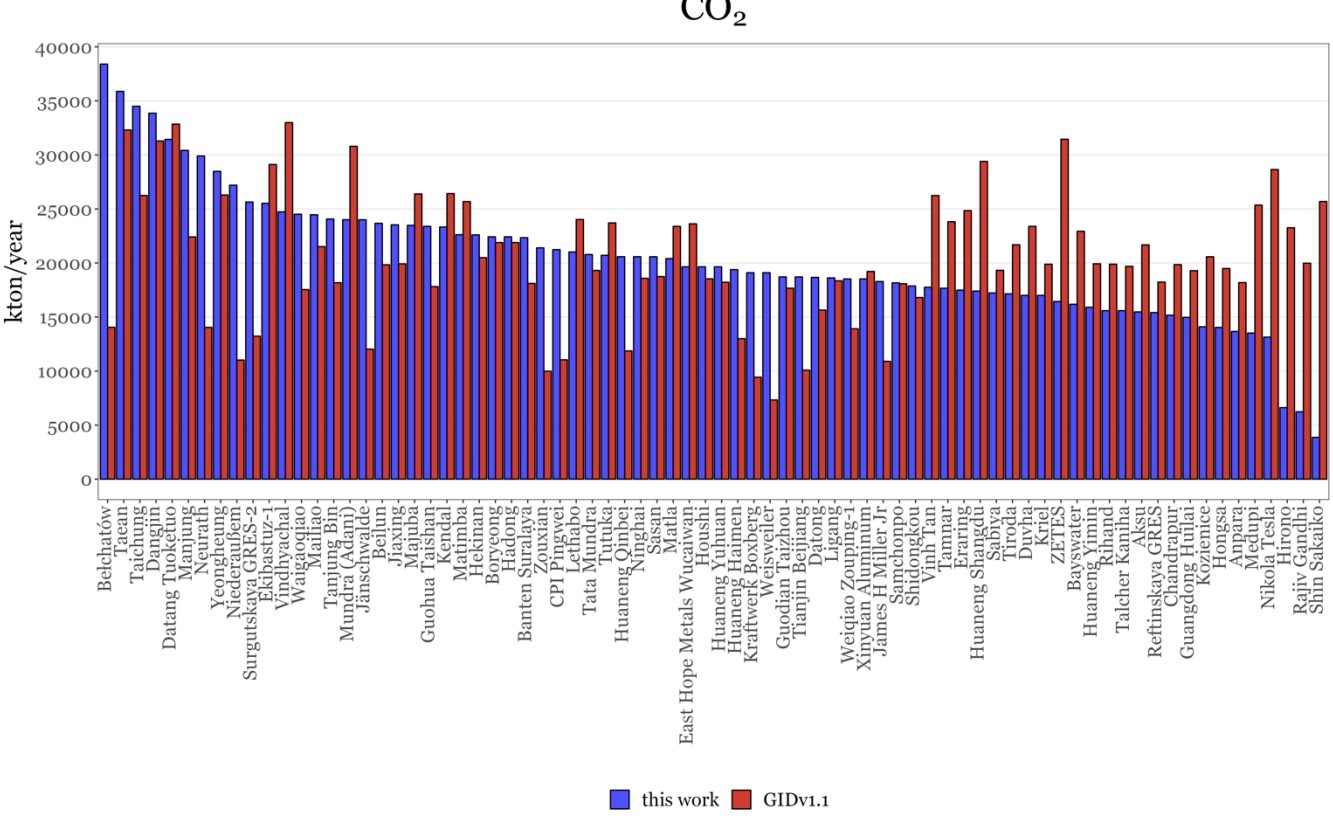


**Figure 10 Comparison between plant-level CO₂, annual emissions [kt/year] estimated by this work and reported by the GIDv1.1 database. Results are shown for the top 50 emitters of each inventory (a total of 76 facilities)**

Additionally, we also performed a plant-to-plant comparison between the top $CO_2$ 100 emitters reported by this work and the
Central Electricity Authority (CEA, 2022) for India and the National Pollutant Release Inventory (NPRI, 2022) for Canada for the year 2018, finding overall a good agreement between the datasets ($R^2$ between 0.79 and 0.81, Fig. S2).

### 3.2.2    Grid cell level

We added the $CO_2$ annual emissions of our point source catalogue onto the same 0.1x0.1 degree grid as the EDGARv7 $CO_2$ inventory and evaluated the spatial correlations between the two gridded datasets. Figure 11 shows the resulting spatial
correlation obtained per country, as well as comparisons of the $CO_2$ gridded distributions obtained with each dataset for selected countries/regions. South Africa, Poland, Australia and the United States are the top 15 emitting countries showing the largest spatial correlation, with values ranging between 0.77 and 0.88. Despite the high correlation obtained for South Africa,

it is important to note that EDGARv7 is not reporting emissions in the grid cells where the Matimba and Medupi coal-fired stations are located. This discrepancy is relevant considering that the Matimba power plant is a typical case study in many top-

down emission studies as it is well isolated and easy to identify using satellite observations (e.g., Hakkarainen et al., 2021). Spatial correlations are in general larger in European and North American countries than in other regions such as Asia, Middle East or Africa. This is in line with the fact that the spatial distribution of EDGAR emissions mostly relies on CARMAv3, which considers exact geographical coordinates for European, US and Canadian facilities but mainly city-center latitude and longitude coordinates for the rest, as explained in Sect. 3.2.1. The reasons for the low correlations observed outside of these

three countries/regions are many, including the aforementioned misallocation of CARMAv3 facilities, the non-inclusion of facilities that were built between 2009 (CARMAv3 reference year) and 2018 (reference year of the present work) or inclusion of facilities that were retired in between these years, and the inclusion of heat only power plant emissions in EDGARv7, which are distributed according to population density. For instance, in Saudi Arabia the low correlation (0.05) is mainly linked to the different spatial patterns observed in the north-east coast, where this work presents a much larger number of grid cells with

high emissions when compared to EDGARv7.

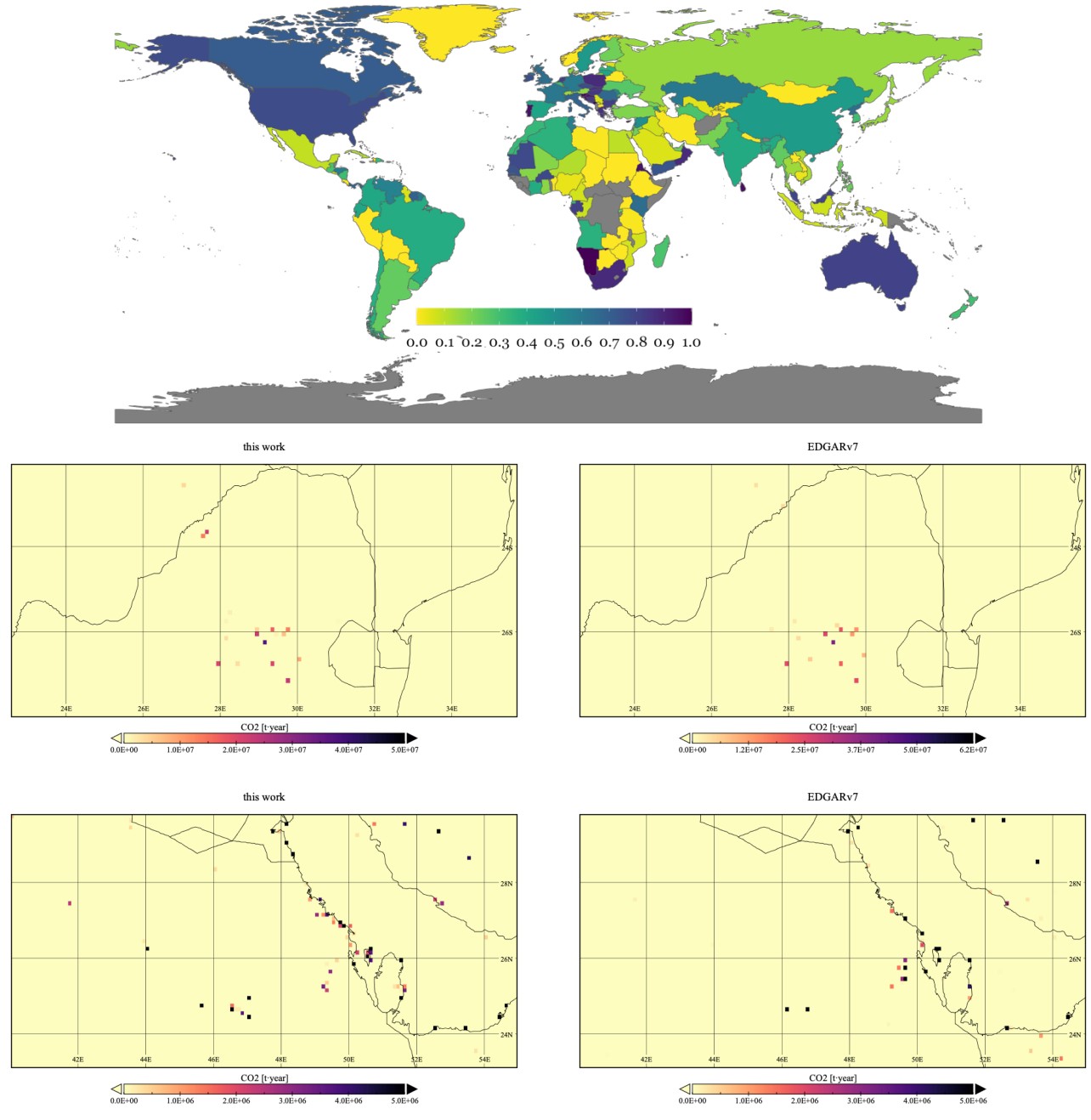

**Figure 11 Map showing the spatial correlation between power plant CO₂ emissions from this work and EDGARv7 at 0.1x0.1 deg resolution per country. The maps below show comparison between 0.1x0.1deg gridded CO₂ annual emissions reported by this work and EDGARv7 in North region of South Africa and North-East coast of Saudi Arabia.**

### 3.2.3 Country level

Estimated country-level emissions were compared against information reported by the EDGARv7 greenhouse gases ($CO_2$ and $CH_4$) and EDGARv6.1 air pollutant ($NO_x$ and $SO_2$) global inventories, as well as national estimates reported by the EMEP Centre on Emission Inventories and Projections (CEIP, 2022), the national inventory submissions to the UNFCCC (UNFCCC, 2022) and other national databases including the USA National Emission Inventory (EPA, 2021), the Chinese Multi-resolution Emission Inventory model for Climate and air pollution research (MEICv1.3 for air pollutants and MEICv2.0 for $CO_2$; Li et al., 2017; Zheng et al., 2018), Cropper et al., (2021) and the GHG Platform-India (2022) for India, the South African Atmospheric Emission License reports (AEL; ESKOM, 2022b), the Mexican National Emission Inventory (INEM; SEMARNAT, 2021), the Australian National Pollution Inventory (NPI; DCCEEW, 2022), the South Korean Clean Air Policy Support System (CAPSS; Choi et al., 2020) and the Taiwan Air Pollutant Discharge Inventory (TEDS; EPA Taiwan, 2021). For all cases the reference year is 2018 except for the national estimates reported for the USA (2017), China (2017), Mexico (2016) and Taiwan (2019). Figure 12 shows the comparison for $CO_2$, $CH_4$, $NO_x$ and $SO_2$ emissions in the top 20 emitting countries.

A general good agreement is observed between the $CO_2$ emissions reported by this work and EDGARv7. The largest differences are observed in Russia, Japan and China, where the present catalogue reports lower emissions (-34%, -15% and -9%, respectively) as it does not include auto-producers (Japan and India) and heat-only plants (Russia). The UNFCCC and independent national estimates are also generally in line with our work, the differences in China, USA and India being of 10%, 9% and -8%, respectively. The discrepancies observed in USA could be related to the fact that the national estimates reported by UNFCCC do not include emissions from auto-producers (IPCC, 2019).

For $CH_4$, EDGARv7 tends to report larger emissions than this work, especially in Russia (-59%), India (-52%) and USA (-29%). In the case of Russia, national estimates are more aligned with EDGARv7 (21%) than the present catalogue (-50%). Oppositely, in India and the USA national emissions are more in line with estimates from this work than EDGARv7, the former presenting substantially higher values (32% in the USA and 122% in India). In Europe sometimes the $CH_4$ emissions from this work matches with UNFCCC reported values (e.g., Italy, Germany), but under and overestimations also occur (e.g., Poland, UK). Generally speaking, the share of the power sector to total national $CH_4$ emissions is small, often around or below 1% (e.g., 0.24% for Italy, 0.19% for Poland, 1.1% for Sweden; UNFCCC, 2022). Hence, the deviations have a negligible influence on national total $CH_4$ emissions and are not further investigated. Moreover, $CH_4$ emissions in power plants are scarcely measured and that corresponding emission factors are associated to very large uncertainties (IPCC, 2019).

EDGARv6.1 reports larger $NO_x$ emissions than this work in all top 20 countries (up to 3 times in Saudi Arabia). The emissions reported by our catalogue are more in line than EDGARv6.1 with the national estimates in most of the countries: China (differences of 26% with this work and of 46% with EDGARv6.1), India (-26% versus 49%), USA (16% versus 222%), Australia (- 15% versus 46%) and Mexico (-15% versus 108%). Emissions reported by EDGARv6.1 in South Africa, Kazakhstan and Turkey are closer to the national values than our estimates, which are between 1.5 and 2.5 times lower. The

discrepancies found for $NO_x$ are much larger than the ones reported for $CO_2$. This is in line with the fact that the estimation of $NO_x$ emissions is typically much more complex, as there are more elements that influence the emission rates, such as

combustion conditions, combustion technologies or air pollution control levels implemented in the facilities, among others. As described in Sect. 2.3.2, the $NO_x/CO_2$ emission ratios considered in this work to estimate $NO_x$ emissions in non-European countries are country/region- and fuel-dependent, but they do not capture differences across power plants within the same country linked to e.g., different technological implementations.

For $SO_2$, comparison results are very similar to what is observed for $NO_x$. Emissions reported by this work are in general much

lower than the EDGARv6.1 values (up to almost 4 times in USA). When compared to national estimates, this study presents lower discrepancies than EDGARv6.1 in India, China, USA, South Africa and Canada. Oppositely, the comparisons performed for Turkey and Mexico indicate that EDGARv6.1 is closer to the national estimates, the present catalogue reporting 1.8 and 3.2 times lower values, respectively. Both EDGARv6.1 and this work present overestimations of similar magnitude in Russia, Kazakhstan and Ukraine when compared to the independent national estimates (between 4.6 and 8.8 times). As mentioned for

$NO_x$, we believe that the discrepancies observed between this work and national inventories are mainly related to the $SO_2/CO_2$ emission ratios considered, which cannot capture differences in emission abatement technologies or type of coals (e.g., sulphur content) used across facilities from individual countries.

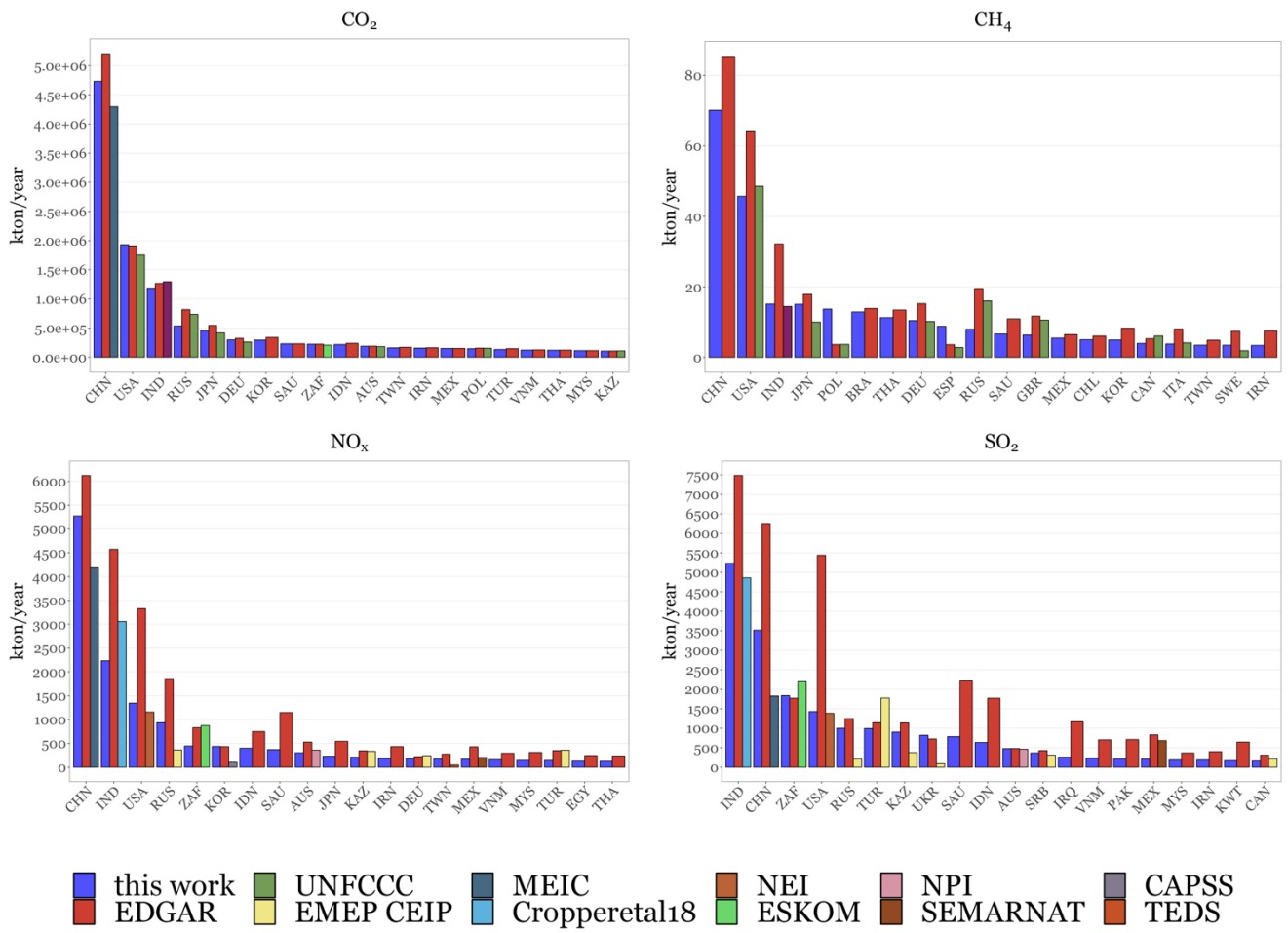

**Figure 12 Comparison between country-level CO₂, CH₄, NOₓ and SO₂ annual emissions kt/year] estimated for the power sector by this work and reported by independent inventories. Results are shown for the top 20 emitters.**

Figure 13 shows the comparison of national $CO_2$ and $NO_x$ emissions reported by the present catalogue and officially estimates (UNFCCC and EMEP CEIP, respectively) for EU27+UK. Results from this work are distinguished between emissions directly obtained from the E-PRTR_v18 database and derived following the gap filling routine described in Sect. 2.3.1.

For most countries there is a good agreement with the nationally reported $CO_2$ total for the energy sector, which is not surprising since most countries will include the emission reporting by facilities in their national inventory. There are several countries, however, where the current catalogue sums up to less than 60% of the reported $CO_2$ national total: France, Denmark, Luxembourg, Austria, Lithuania, Latvia, Malta and Norway. When looking into more detail at the $CO_2$ emissions by fuel type (Fig. 14), the discrepancies for these countries appear to be caused by a significantly lower contribution of biomass $CO_2$

emissions. For example, for France, Austria and Denmark, but also for Germany, the national inventory has much higher $CO_2$ emissions included from biomass combustion. The biomass/biogas power plants responsible for this contribution, however, mostly cannot be found in the EPRTR/LCP or the other plant specific databases that were consulted for this work (see Sect. 2.1). This suggests these are mostly small size plants that fall below the reporting thresholds.

For $NO_x$, the differences are a bit larger than for $CO_2$, with the current catalogue covering on average about 88% of the national

total $NO_x$ emissions. For Spain, the combined reporting of EPRTR facilities already substantially exceeds (by 73%) the national reporting of energy sector emissions, which is related to the fact that national estimates reported to EMEP CEIP do not include emissions from the Canary Islands, as they are located outside of the geographical scope of EMEP. As shown in Sect. 3.1, $NO_x$ emissions from power plants located in the Canary Island are substantial as they are operated by internal combustion diesel engines. For both $CO_2$ and $NO_x$, the influence of the gap filling of emissions appears limited on the national level.


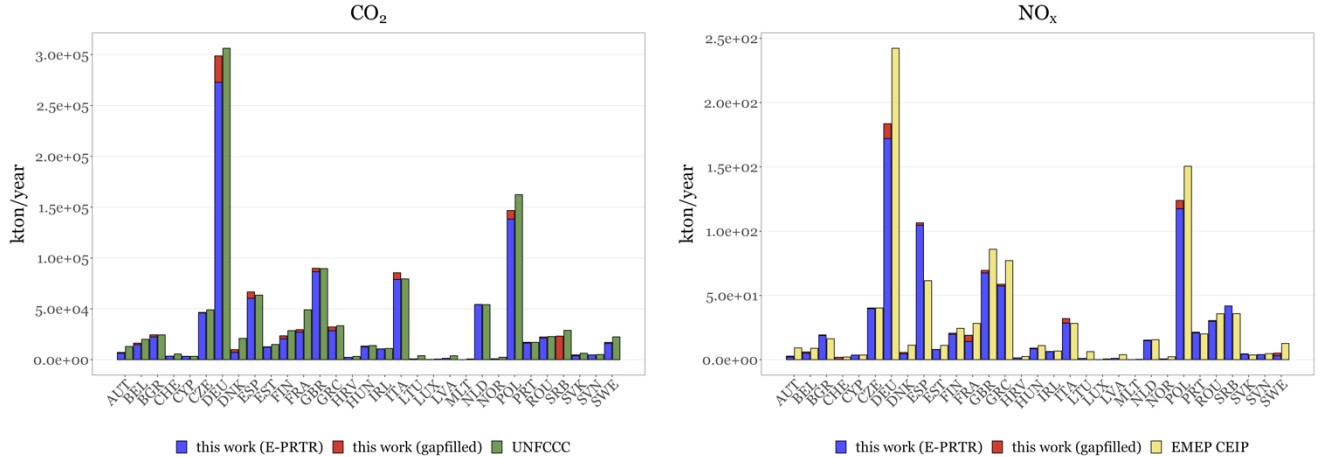

**Figure 13 Comparison between country-level CO₂ and NOₓ annual emissions kt/year] estimated for the power sector by this work and reported by EMEP CEIP and UNFCCC for EU27+UK. Results from this work are distinguished between emissions directly obtained from the E-PRTR_v18 database and derived following the gap filling routine.**

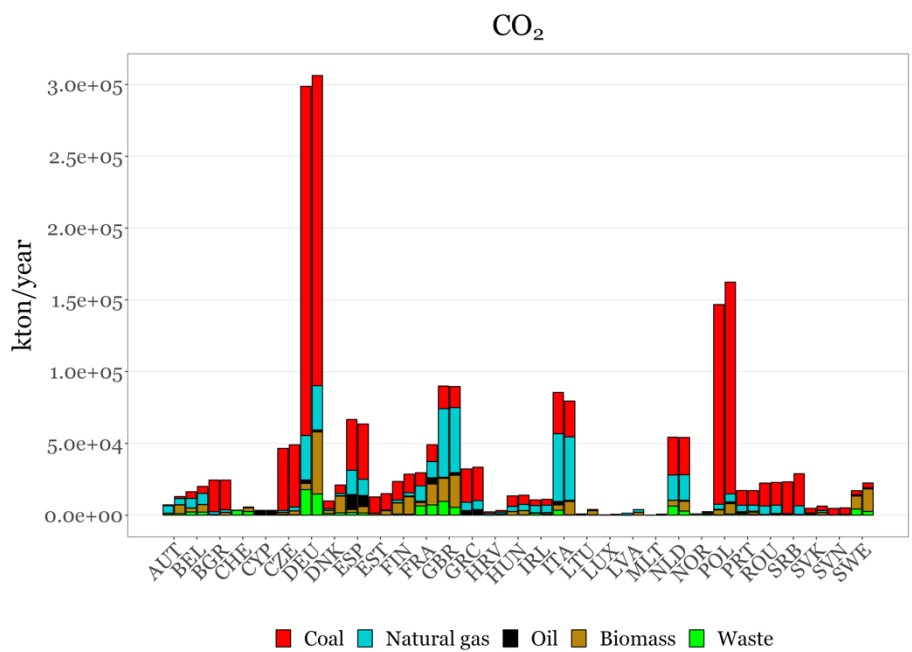

**Figure 14 Comparison between country-level CO₂ annual emissions kt/year] estimated for the power sector by this work (left columns) and reported by UNFCCC (right columns) for EU27+UK by fuel type (coal, natural gas, oil, biomass, waste).**

## 3.3    Temporal distribution

Figure 15 illustrates with an example the plant-level hourly $CO_2$ emission estimates (kg·h$^{-1}$) obtained when combining the information on total annual emissions with the temporal profiles reported in the resulting catalogue. We used the HERMESv3 emission system (Guevara et al., 2020) to combine the total annual emissions per facility (Sect. 2.3) with the corresponding country- and fuel-dependent profiles (Sect. 2.4) and derive hourly emissions for the year 2018. To distribute the annual emissions to hourly emissions per facility, the following relationship is used in HERMESv3 (Eq. 1):


$$E_{p,t} = E_p \cdot \frac{M_{p,m}}{12} \cdot \frac{W_{p,d} \cdot n_{d,m}}{\sum_{d=1}^{7} W_{p,d} \cdot n_{d,m}} \cdot \frac{H_{p,h}}{24}$$    Eq. 1

Where $E_{p,t}$ are the hourly emissions [kg/h] for power plant $p$ and date $t$ (e.g., 2018-11-27 17:00h); $E_p$ are the original annual emissions [kg/h] for power plant $p$; $M_{p,m}$ is the monthly weight factor [0-12] for power plant $p$ and month-of-the-year $m$; $W_{p,d}$ is the weekly weight factor [0-7] for power plant $p$ and day-of-the-week $d$ [1 Monday – 7 Sunday]; $n_{d,m}$ are the number of

day-of-the-week $d$ in month $m$ and $H_{p,h}$ is the hourly weight factor [0-24] for power plant $p$ and hour-of-the-day $h$.

The results shown in Fig. 15 correspond to four coal-fired power plants: As Pontes (Spain), Belchatów (Poland), Jänschwalde (Germany) and Matimba (South Africa). The Matimba power plant is the facility that presents the flattest distribution, the results indicating that it is a base load power source. On the other hand, emissions from Belchatów, Jänschwalde and As Pontes present a clear seasonality, with emissions peaking during February, coinciding with a European cold spell that caused below

average temperatures in most European countries (C3S, 2018) and, in the case of As Pontes, also during summer, when energy demand increases due to the use of air conditioning systems. A weekend effect is also clearly observed for all facilities, with emissions significantly dropping during Saturday and Sunday when compared to the weekdays.

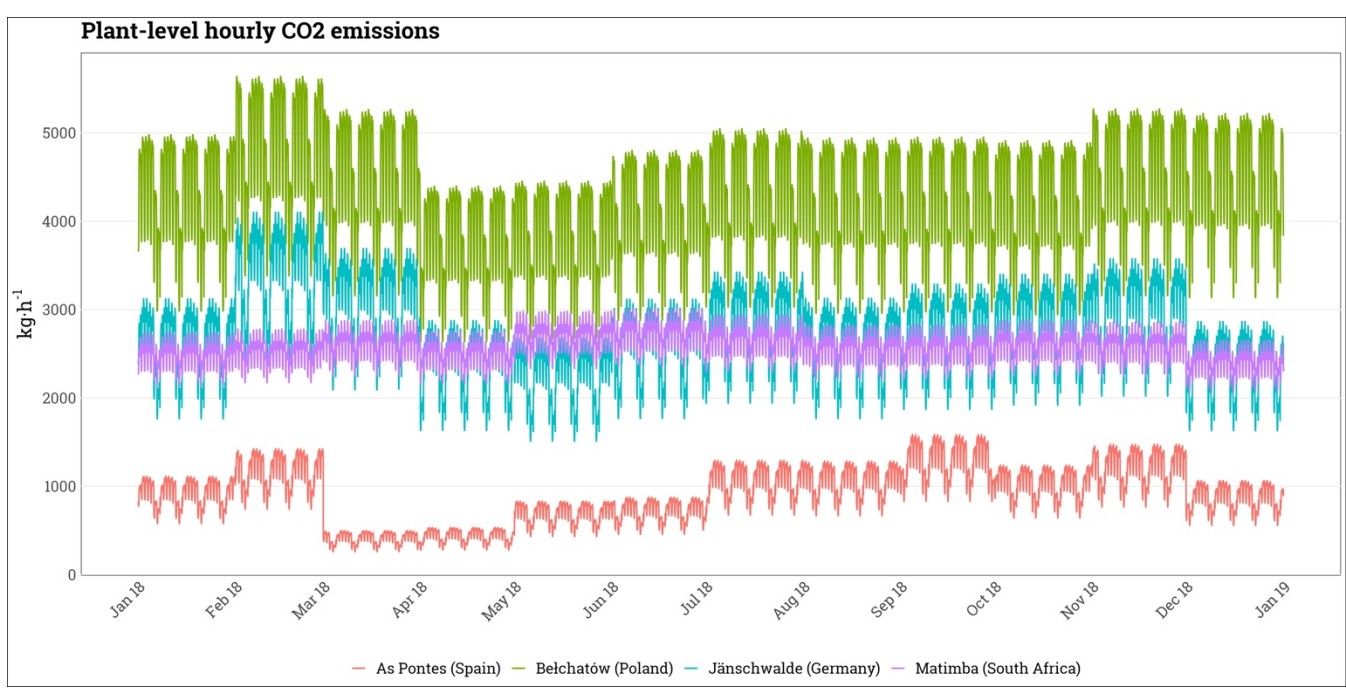

Figure 15 Estimated hourly CO$_2$ emissions [kg/h] for the As Pontes (Spain), Belchatów (Poland), Jänschwalde (Germany) and Matimba (South Africa) coal-fired power plants.

Country-dependent monthly, weekly and hourly profiles were constructed using as a basis the estimated plant-level hourly emissions. The resulting emissions were aggregated at the country level and normalised to derive the corresponding temporal profiles. Results were compared against the temporal profiles reported by Denier van der Gon et al. (2011) for the power industry sector (hereinafter referred to as the TNO profiles), which are widely used in the modelling community to quantify observation-based emission estimates (e.g., Kuhlmann et al., 2021). Figure 16 shows an example of monthly, weekly and hourly profiles constructed for the power sector for selected countries and comparison against the TNO profiles.

At the monthly level, large variations are observed between countries. Profiles for United Arab Emirates (ARE) and Kuwait (KWT) present a clear peak during summer, coinciding with the intensive use of air conditioning systems. In the case of USA Pennsylvania (USA-PA), we identify two types of peaks, one related to space cooling needs during July and August, and another one linked to space heating needs during January and December. In Germany (DEU) and Poland (POL) we also distinguish the peaks during wintertime, while the increase of emissions during summer is much lower than the previous cases as these countries are in higher latitudes where the summers are not too hot. The seasonality in India (IND), China (CHN), South Africa (ZAF) and Australia (AUS) are much flatter. The TNO profiles were designed for Europe and the mismatch for countries with different climatic regimes such as United Arab Emirates and Kuwait is to be expected. Nevertheless, we can see that all the profiles differ significantly with the TNO profile, which reports a V-shape seasonality, with emissions peaking

during wintertime and presenting their lowest value during summer, and therefore not capturing the peak related to space cooling needs, which is also relevant in Europe.

Concerning the weekly variability, profiles constructed for European countries (i.e., Germany and Poland) are in line with the TNO profile, showing a strong weekend effect, with emissions being reduced more than 20% between weekdays and Sundays. On the other hand, profiles estimated for USA Pennsylvania, South Africa and Australia are much flatter (5 to 10% differences between weekdays and weekends), while India shows almost no differences between weekdays and weekends.

Finally, constructed hourly profiles are quite consistent between countries, all of them showing a rather flat variation, with

emissions being slightly larger (10-15%) during daytime (between 07:00h and 20:00h). Similarly to what we see for the monthly profiles, large inconsistencies are observed between the constructed profiles and the TNO profiles, the latter showing a much larger variation between emission levels during night- and daytime and not reproducing the afternoon peak reported by the constructed profiles in most of the countries.

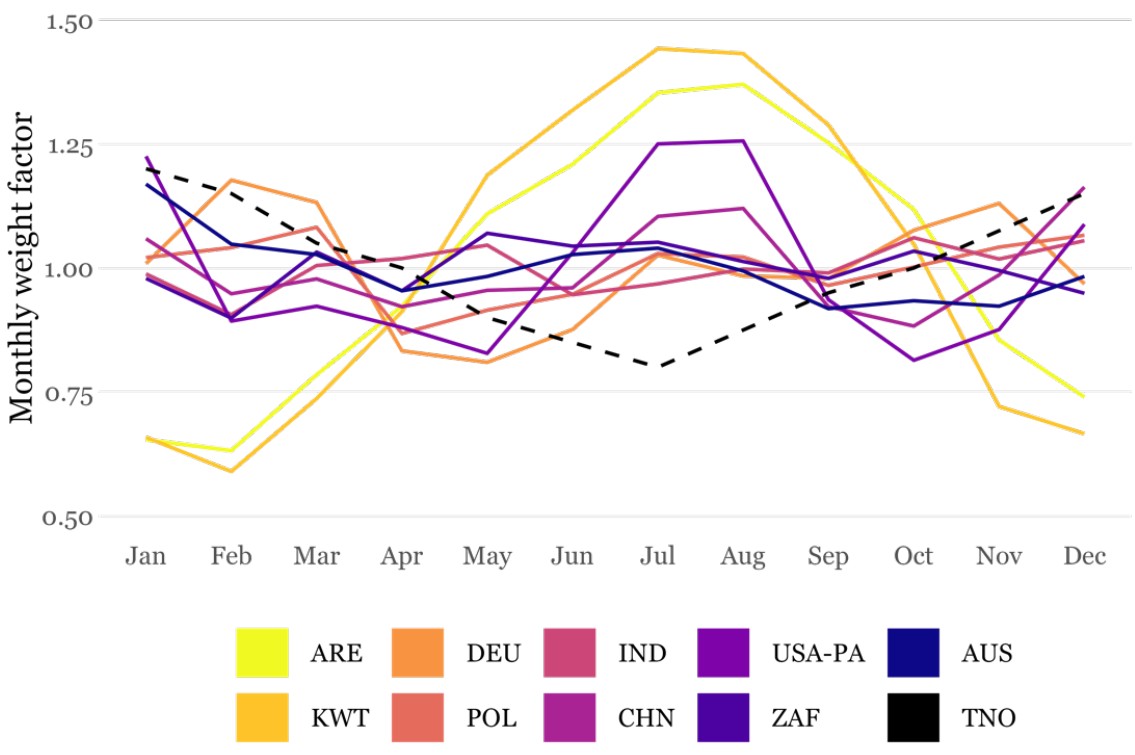

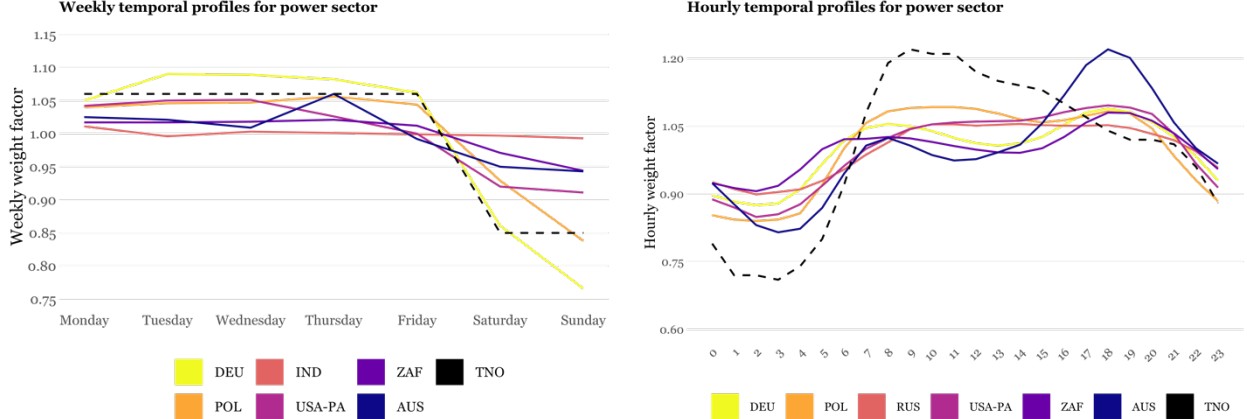

**Figure 16 Power sector monthly, weekly and hourly profiles constructed for selected countries, including Arab Emirates (ARE), Kuwait (KWT), Germany (DEU), Poland (POL), India (IND), China (CHN), USA Pennsylvania (USA-PA), South Africa (ZAF) and Australia (AUS). For each temporal resolution, estimated profiles are compared against the profiles reported by Denier van der Gon et al. (2011) (TNO).**

The country-level monthly profiles derived in this work were also compared against the EDGAR temporal profiles (Crippa et al., 2020). We normalised the 2018 EDGARv7 $CO_2$ monthly emissions reported per country for the energy sector, which are calculated using the temporal distribution profiles described in Crippa et al. (2020), and then compared them against our profiles. Figure 17 shows the correlation values obtained between monthly profiles per country as well as the comparisons between monthly profiles for selected countries. Correlations are large in several of the top 20 emitting countries (e.g., China, 0.75; Australia, 0.95; Russia, 0.92; Japan, 0.78; Mexico, 0.87; South Korea, 0.80; Taiwan, 0.89; Turkey, 0.95; Kazakhstan, 0.89). Low or even negative correlations are observed in some of the top emitters, including India (r = -0.19), South Africa (r = 0.16) and the United States (r = 0.54). Nevertheless, when looking at the comparisons between monthly profiles reported in these three countries, we observe that both EDGARv7 and the present work suggest a very similar seasonality, with India and South Africa presenting a rather flat distribution, and United States showing two peaks in winter and summer, coinciding with the increase of electricity demand for space heating and cooling purposes, respectively. Brazil, Peru and Kuwait are among the countries presenting the largest negative correlation values (between -0.77 and -0.45). In the three cases, the seasonality reported by EDGARv7 and this work are completely opposite. While we suggest that emissions from the energy sector peak between June and September, coinciding with summer in Kuwait and southern winter in Brazil and Peru, EDGARv7 indicates that the largest emission level occur between December and February (southern summer in Brazil and Peru). For these three countries, our profiles were constructed from national electricity generation statistics (see Table 5), while in the case of EDGARv7 they were indirectly estimated from regional averages computed using country specific profiles belonging to the same world region. Most of the other countries for which a negative correlation exist are in Africa and Southeast Asia, where the information on electricity statistics to derive monthly profiles is rather scarce.

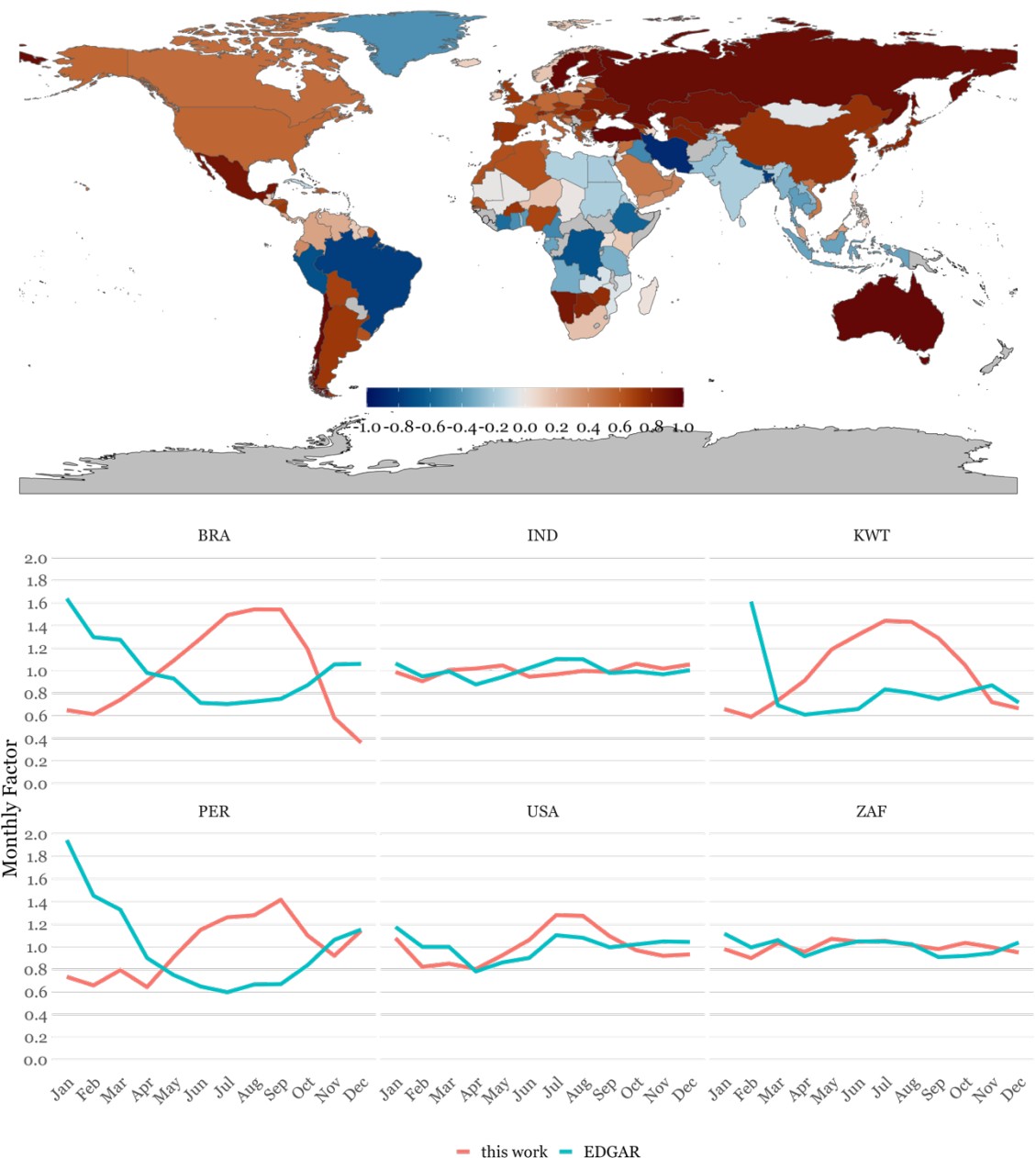

**Figure 17 Map showing the correlation between monthly profiles constructed in this work and reported by Crippa et al., (2020) (EDGAR) for the energy sector per country. Countries in grey indicate that no correlation could be computed as no profiles are reported. The line plots below the map show the comparison between monthly profiles for selected countries.**

The temporal profiles constructed in the present work are country and fuel-dependent, but not facility-dependent. Large differences between the emission temporal distribution of plants belonging to the same country may occur, e.g., base load versus peak load, or if they are used for electricity only or electricity and heat. Figure 18 illustrates these differences by comparing the monthly, weekly and hourly profiles constructed in the present work for German coal-fired power plants (black solid lines) against profiles estimated for individual facilities making use of plant-level electricity generation statistics provided by ENTSO-E (2021). The comparison includes the top 20 producing coal-fired German power plants, which together supplied more than 75% of the national electricity from burning coal in 2018. Profiles from power plants are represented with different colours and sizes that indicate their annual electricity production (the thicker and darker the line is, the more electricity supplies). Results indicate a significant heterogeneity between profiles across plants. As expected, the more electricity produces a power plant, the more continuously it supplies electrical energy throughout the year and, subsequently, the flatter are its associated monthly, weekly and hourly profiles. On the contrary, power plants producing less energy tend to show large variations between e.g., weekdays and weekends or daytime and night-time hours, as their behaviour is more linked to demand response. The discrepancies between our profiles and the plant-level profiles tends to be lower when looking at the top generating facilities. Consequently, our profiles better represent the temporal behaviour of those facilities emitting more emissions and that subsequently are easier to be detected by satellite observations. However, important differences are still observed when comparing the profiles from the catalogue with the ones derived from the largest generating plant (i.e., Neurath), with differences up to 18%, 10% and 8% for the monthly, weekly and hourly weight factors, respectively. The largest discrepancies occur with the monthly distributions as they are influenced by many factors besides the typical changes in electricity consumption (e.g., more demand during weekdays than weekends) including economic variables, meteorological conditions and electricity trade, among others.

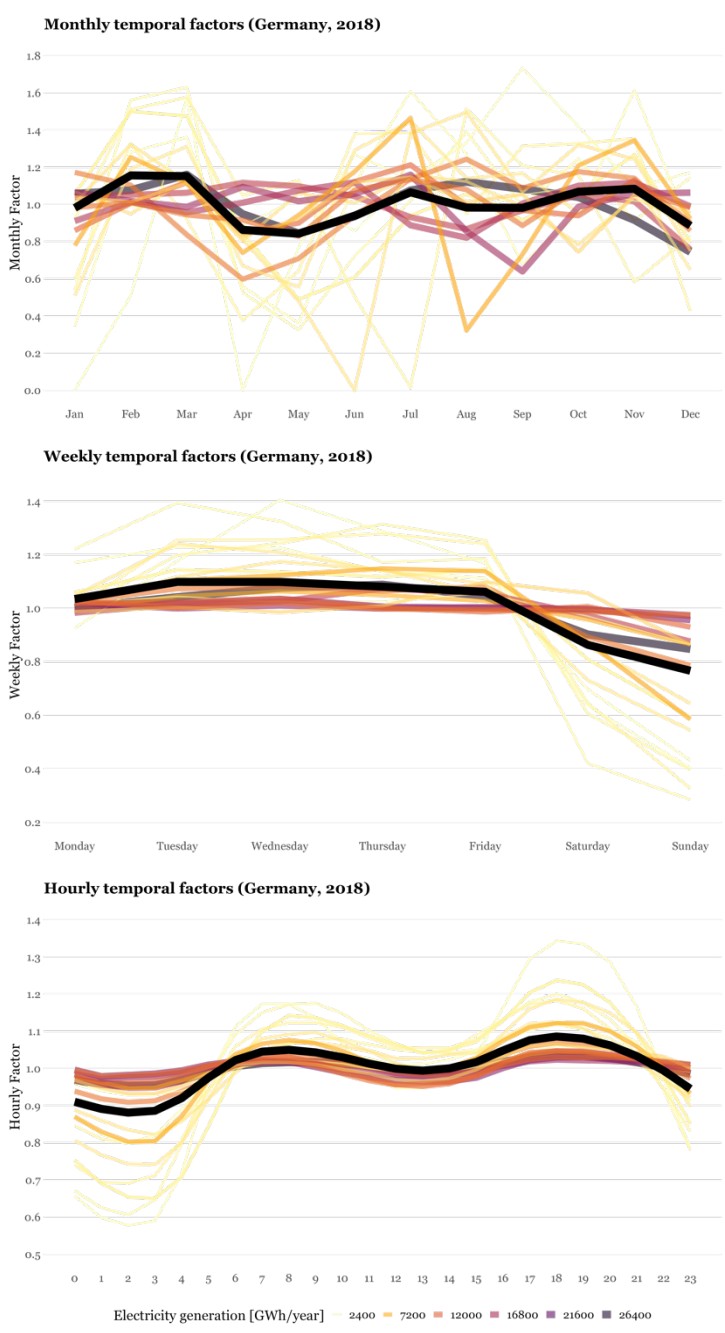

**Figure 18 Comparison between monthly, weekly and hourly temporal profiles constructed in the present work for German coal-fired power plants (black solid lines) against profiles estimated for the top 20 producing coal-fired German facilities making use of plant-level electricity generation statistics (ENTSO-E, 2021). Profiles from power plants are represented with different colours and sizes that indicate their annual electricity production [GWh/year]**

 **3.4    Vertical allocation**

Figure 19 shows an example of the daily bottom (blue) and top (red) plume values [m] at the Matimba (South Africa) and Bełchatów (Poland) coal-fired power plants estimated by the HERMESv3 model for the year 2018. Dashed lines indicate the stack height of each facility (i.e., 250m for Matimba and 300m for Bełchatów). Large month-to-month and day-to-day variations are observed for both the bottom and top plume heights at the two facilities, which are related to changes in the
meteorological parameters and atmospheric stability driving the plume rise calculations, mainly the air temperature at the stack height and the boundary-layer height (Guevara et al., 2020). The bottom plume heights are, on average, 41% (Bełchatów) and 70% (Matimba) higher than the corresponding physical stack heights, while top plume height values are on average 124% (Bełchatów) and 206% (Matimba) higher.

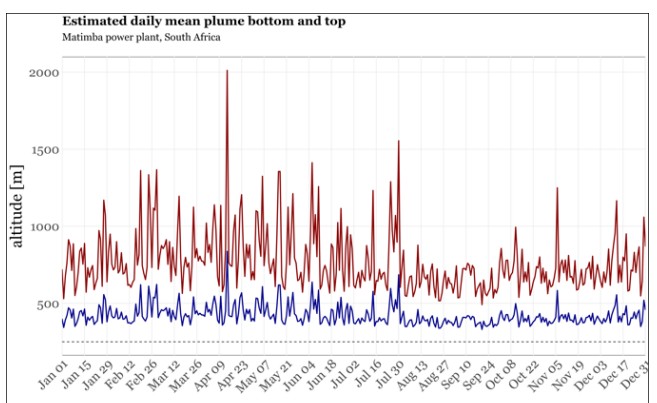 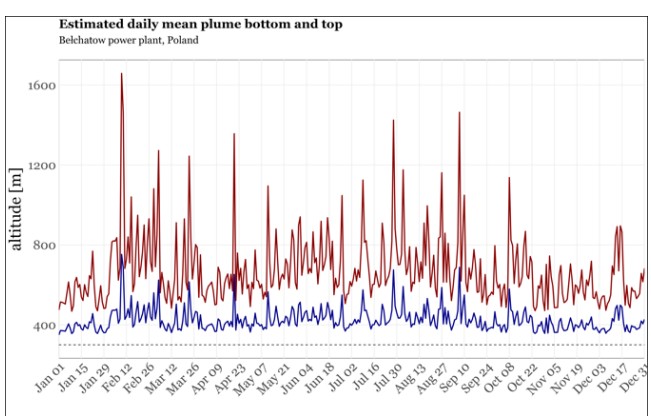

**Figure 19 Estimated daily bottom (blue) and top (red) plume values [m] at the Matimba (South Africa) and Bełchatów (Poland) coal-fired power plants for the year 2018. Dashed grey lines indicate the stack height of each facility.**

For each facility, the estimated $CO_2$ hourly emissions were first uniformly allocated across 16 vertical layers (from 0m up to 1500m with breaks every 100m, and above 1500m) considering the modelled hourly plume top and bottom values, then summarised to the annual level and finally normalised to 1 to derive annual and emission-weighted vertical profiles. Figure 20
shows the emission-weighted average annual vertical profiles computed for the As Pontes (Spain), Belchatów (Poland), Jänschwalde (Germany) and Matimba (South Africa) coal-fired power plants. The estimated profiles are compared against the vertical distribution proposed by TNO in the Copernicus CAMS-REG inventory for the public electricity and heat production sector (Kuenen et al., 2022). Jänschwalde is the power plant with the largest share of emissions occurring in lower layers (i.e., 78% of total emissions allocated between 100 and 300m). This is due to the fact that emissions from this facility are released
through the cooling towers, which have a height of only 120m. On the other hand, As Pontes is the facility with the largest share of emissions allocated between 400m and 600m (76%), as it is the power plant with the highest chimney in Europe (365.5m). Belchatów and Matimba present rather similar vertical distribution profiles, partially because both facilities have

stacks of similar height (300m and 250m, respectively). Matimba is the power plant allocating the largest share of emissions across the top layers (8% of total emissions above 1000m). This is related to the larger exit velocity of the gases when compared to e.g. As Pontes (i.e., 26m/s versus 21m/s, almost 25% larger) as well as to differences in the local climatological conditions. The TNO profile distributes most of the emissions (85%) between 200m and 500m, the shares reported in higher altitudes (i.e., between 500m and 800m) being considerably lower (8.5%) than the ones computed for As Pontes, Belchatów and Matimba (between 32% and 51%). This discrepancy is in line with the fact that the TNO profile is derived from the work by Bieser et al., (2011), which assumed an average stack height of 159m for calculating the plume rise in large power plants (> 50MWth), a value significantly lower than the stack heights of the three power plants included in the comparison (365.5m, 250m and 300m). The results suggest that the TNO vertical profile may not be representative of power plants with high chimneys.

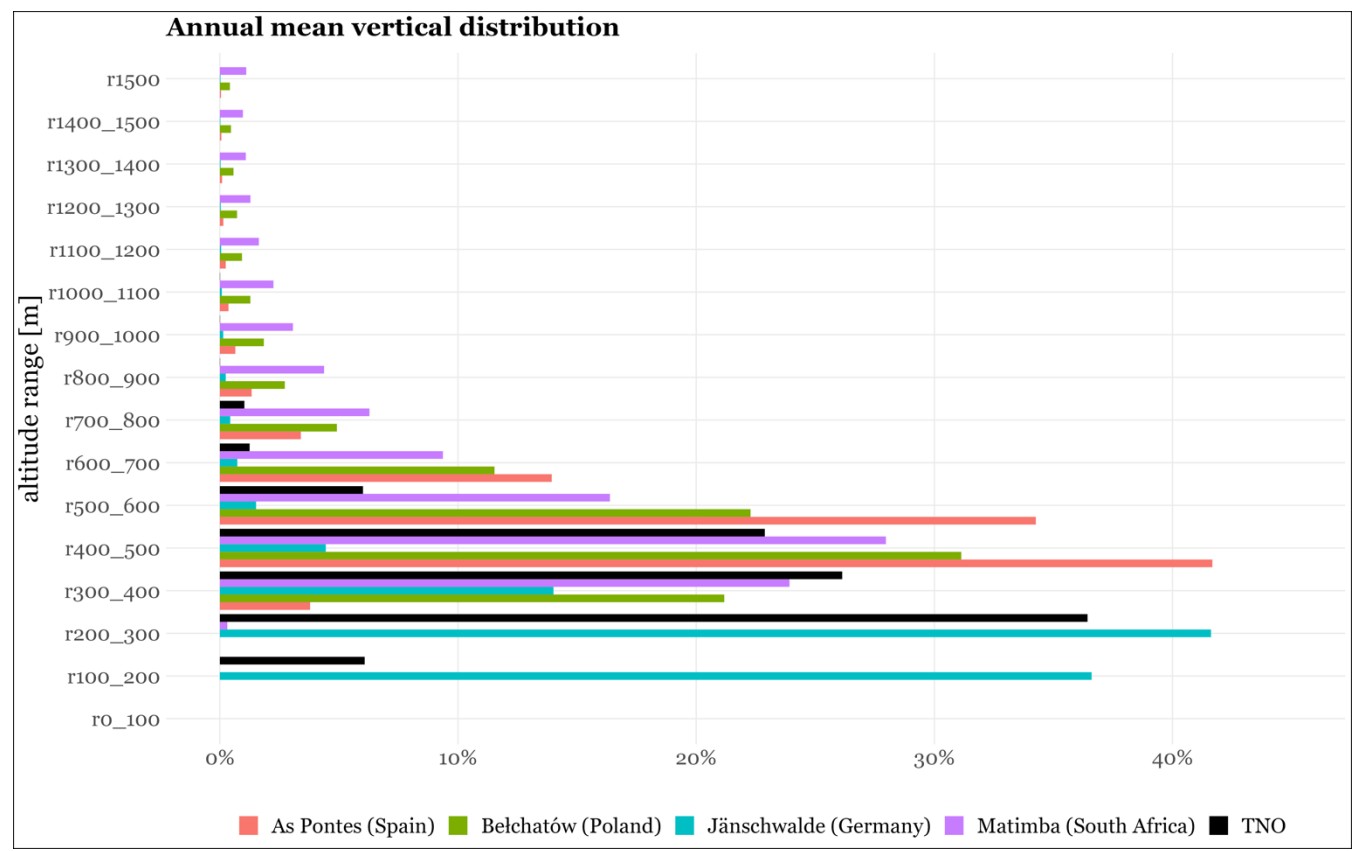

**Figure 20 CO$_2$ emission-weighted average annual vertical profiles estimated for the As Pontes (Spain), Belchatów (Poland), Jänschwalde (Germany) and Matimba (South Africa) coal-fired power plants. For each facility we represent the associated vertical weight factors [%] across 16 vertical layers (from 0m up to 1500m with breaks every 100m, and above 1500m). Estimated profiles are compared against the one provided by TNO in the Copernicus CAMS-REG inventory (Kuenen et al., 2022; TNO).**

## 4    Data availability

The global catalogue of $CO_2$ emissions and co-emitted species from power plants, including plant-level total annual $CO_2$, $NO_x$, $SO_2$, CO and $CH_4$ emissions (in t·year), information on fuel type (i.e., coal, natural gas, oil, biomass and waste), geographical location (latitude and longitude coordinates in decimal degrees) and associated temporal profiles (monthly, weekly, hourly unitless weight factors that sum up to 12, 7 and 24, respectively) and vertical profiles (normalised relative contribution of total emissions across 16 vertical layers, from 0m up to 1500m, with breaks every 100m, and above 1500m, that sum up to 1), is provided in a collection of five CSV files through the CAMS document repository (https://doi.org/10.5281/zenodo.10002124, Guevara et al., 2023). The CSV files included in the catalogue are:

- coco2_ps_catalogue_v2.0.csv: list of power plants with associated total annual $CO_2$, $NO_x$, $SO_2$, CO and $CH_4$ emissions [t·year], information on fuel type (i.e., coal, natural gas, oil, biomass and waste), geographical location (latitude and longitude coordinates in decimal degrees), monthly, weekly and hourly temporal profile unique identifiers and vertical profile unique identifiers.
- coco2_ps_monthly_profiles_v2.0.csv: list of monthly temporal profile unique identifiers with monthly weight factors (unitless) associated to each month [between 0-12]
- coco2_ps_weekly_profiles_v2.0.csv: list of weekly temporal profile unique identifiers with weekly weight factors (unitless) associated to each month [between 0-7]
- coco2_ps_hourly_profiles_v2.0.csv: list of hourly temporal profile unique identifiers with weekly weight factors (unitless) associated to each month [between 0-24]
- coco2_ps_vertical_profiles_v2.0.csv: List of vertical profile unique identifiers with weight factor associated to each vertical layer [between 0-1]. Distribution is defined across 16 vertical layers (from 0m up to 1500m with breaks every 100m, and above 1500m)

The catalogue is provided together with a README file in .docx format that contains a description of each file of the global catalogue listed above and associated fields of information. The catalogue is maintained as a Gitlab repository hosted at https://earth.bsc.es/gitlab/mguevara/global_catalogue_power_plant_emissions/ (last access: September 2023). Bug reports and other issues should be posted to the issue tracker at https://earth.bsc.es/gitlab/mguevara/global_catalogue_power_plant_emissions/-/issues (last access: September 2023). The catalogue is licensed under Creative Commons Attribution 4.0 International (CC-BY-4.0: https://creativecommons.org/licenses/by/4.0, last access: August 2023).

# 5 Limitations of the dataset

The current catalogue provides an updated and high-resolution global picture of the spatial (horizontal and vertical) and temporal characterization of emissions from power plants. Despite all the efforts, there are, however, some limitations associated with the current version of the dataset that potential users should consider:

- Emissions from non-European auto-producer facilities are not consistently included across countries due to the lack of information. Overall, we could not include emissions from auto-producers in 35% of the non-EU countries considered, which translates into 4.1% of total estimated $CO_2$ emissions that could not be allocated to specific point sources. The most relevant countries affected by this limitation are Russia, India and Japan, the share of national emissions that could not be assigned to individual facilities for these countries is between 14% and 21%.

- For the non-European dataset, heat-only facilities are not included due to the lack of information. This gap may be relevant in countries where the share of fossil fuels used to produce heat only is significant, mainly Ukraine (25%), Russia (20%), Belarus (20%), Kyrgyzstan (18%) and Uzbekistan (10%).

- We identified a list of countries for which we found the location of their power plants but that we could not include in the final catalogue since their energy balances are not reported by the IEA World Energy Balances database, and subsequently corresponding emissions could not be estimated. It's important to note that most of these missing countries are small island countries (e.g., Aruba, Anguilla, Samoa Nord-americana, Antigua and Barbuda, Bahamas, Fiji, Cabo Verde, Cayman Islands), which have a very limited contribution to total $CO_2$ emissions from the power sector (i.e., 0.03% according to Tong et al., 2018).

- For the European dataset, a substantial number of emission values was gap filled using a tiered routine, using facility-specific-, or more generic pollutant ratios to estimate emissions. In total, gap filled emission values contribute less than 10% to total emissions for $CO_2$, NOx and $SO_2$, but more than 60% for CO and $CH_4$. In terms of emission values for individual locations, close to 50% of power plants have been gap filled for $CO_2$ and $NO_x$, 80% for $SO_2$, and above 90% for CO and $CH_4$. These results indicate that gap filling plays a more prominent role in small and medium-sized combustion facilities, and that emission values from large power plants are typically reported. The approach implemented could however lead to under- or overestimations of emissions for individual plants, especially for $NO_x$ and $SO_2$ and the important role of air pollution control levels on this species, which can vary across facilities.

- For the non-European coal-, natural gas- and oil-fired power plants emissions from co-emitted species ($NO_x$, $SO_2$, CO) were estimated using fuel- and country/region-dependent emission ratios derived from the GAINS inventory, which reflect national emission standard aspects. However, pollution abatement controls do not only differ by country or regions, but also across power plants within the same country. The use of not only fuel- and country-dependent, but also technology-dependent emission ratios could potentially help in reducing this uncertainty.

- For the non-European dataset, plant-level emissions were estimated by distributing fuel-dependent national emissions among facilities as a function of their installed capacity, which in some cases may not be representative of their actual activity (i.e., capacity factor) and may lead to over- or underestimations.

- For the European dataset, there was mostly good agreement with the national inventory totals for the energy sector in case of $CO_2$ and $NO_x$. The main source of discrepancies appeared to be the missing biomass power plant capacity in the dataset. These plants are likely too small to be included in official reporting and most public power plant datasets, for example because they fall below reporting thresholds. A more in-depth look into these biomass plants is needed to improve coverage and completeness.

- The final catalogue of power plants covers the main fuels used to produce energy and heat, including coal, natural gas, oil, solid biomass and solid waste. However, we are still missing some fuels that are relevant in specific countries and for emissions from certain species (e.g., $CH_4$) such as biogas (e.g., Thailand, India, Turkey, Australia) and liquid biofuel (e.g., South Korea).

- The comparison between geographical locations reported by the present catalogue and the CARMAv3 database indicate that the location of current top emitters is better represented in this work, especially in Asian countries such as China, Taiwan and India, where the majority of the CARMAv3 facilities are assigned with city-center latitude and longitude coordinates. Despite putting substantial efforts in correcting the location of facilities that are originally reported with wrong coordinates, there may be some error still present in the dataset, especially in the case of small and medium sized plants.

- Concerning the representativeness and stability of the temporal profiles constructed for 2018 over the years, we refer to the analysis performed by Crippa et al. (2020), in which monthly temporal profiles for the power generation sector for the 35 Organisation for Economic Co-operation and Development (OECD) countries over the time period 2000–2017 was analyzed. According to their results, large standard deviations mainly occur in countries where the use of fossil fuels to generate electricity is scarce such as Finland, Iceland, Norway or Sweden, where more than 90% of total electricity comes from renewable sources, or where the number of fossil fuel power plants that supply energy to the grid is very low (e.g., Latvia, with five natural gas power plants). These situations can cause large relative year to year changes in the monthly profiles, as they are more sensible to changes in meteorological conditions or the dynamics of specific facilities, among others. On the other hand, the year-to-year variations of the monthly profiles obtained for top emitting countries (e.g., China, Japan, USA, Australia) is in general much lower.

- The temporal profiles assigned to the power plants are country and fuel-dependent, but not facility-dependent. Large differences between the emission temporal distribution of plants belonging to the same country may occur, e.g., base load versus peak load power plants or electricity and heat versus heat only power plants. A comparison between the monthly, weekly and hourly profiles constructed in the present work for German coal-fired power plants (black solid lines) against profiles estimated for individual facilities supports this hypothesis, but also indicates that our profiles are sufficiently capable of representing the temporal behaviour of the top emitting facilities, which are easily detected

by satellites. However, important differences are still observed when comparing the profiles from the catalogue with the ones derived from the larger generating plants, specially at the monthly scale, for which an important heterogeneity between plants exist due to the influence of not only changes in the demand, but also in the meteorological conditions, economic variables and electricity trade. The development of plant-level profiles is nevertheless limited by the lag of information on plant-level electricity generation statistics, which is currently limited only to certain regions (e.g., EU27).

- The final database provides plant-level annual mean vertical profiles that consider meteorology and stack parameters information. However, large variations in the vertical distribution of emissions may occur between seasons, days of the year and hours of the day due to changes in the meteorological parameters that influence the atmospheric stability and the corresponding vertical dispersion of the emissions.

- Despite identifying several power plants in which emissions are released through the cooling towers instead of the traditional chimneys (mainly in Germany), there may still be multiple facilities in the catalogue that are not correctly flagged. Moreover, for power plants using the cooling towers to release the emissions, we considered the same plume rise formulas as the ones used for traditional stack chimneys. According to Brunner et al. (2019), this assumption may entail an underestimation of the resulting effective emissions height of 20% to 100% due to the combination of several factors, including the additional release of latent heat from cooling towers or the interaction of plumes from cooling towers located next to each other.

- The stack parameters information used to perform the plume rise calculations has a limited coverage (e.g., only 28% of total $CO_2$ emissions have specific stack height information, and only 15% specific exit velocity data), which may bring an additional uncertainty to the estimated vertical profiles. According to the sensitivity runs performed by Bieser et al. (2011), changes in estimated emission heights are almost linear with changes in stack height and exit velocity, indicating a large influence of these parameters on the result.

- Caution should be taken when combining the global point source dataset with other existing gridded emission inventories (e.g., EDGAR) to avoid issues of double counting or incompleteness. Avoiding these problems can be challenging if, for instance, the sector classification of the gridded inventory is broad (e.g., emissions from power plants are included together with emissions from refineries and other energy industries under the same sector). A reclassification of the gridded emissions may be needed in these cases to ensure an appropriate combination of datasets.

## 6    Conclusions

We present a high-resolution catalogue of $CO_2$ emissions and co-emitted species ($NO_x$, $SO_2$, CO, $CH_4$) from thermal power plants for the year 2018. The construction of the database follows a bottom-up approach, which combines plant-specific information with national energy consumption statistics and fuel-dependent emission factors and emission ratios. Annual emissions are provided for each plant at their exact geographical locations. Each facility is linked to a country- and fuel-dependent temporal profile (i.e., monthly, day-of-the-week and hourly) and plant-specific vertical distribution profile, which allows to derive spatial- and temporal-resolved emissions for modelling purposes. The resulting catalogue has been developed in the framework of the Prototype System for a Copernicus $CO_2$ service (CoCO2) EU-funded project to support the development of the CO2MVS capacity. Results from the catalogue were compared to widely used and state-of-the-art emission inventories like the Carbon Monitoring for Action (CARMA), the Global Infrastructure emission Database (GID) and the Emissions Database for Global Atmospheric Research (EDGAR) databases, as well as officially reported emission data.

### 6.1    Future perspective

The current point source catalogue represents an effort to improve the spatial (horizontal and vertical) and temporal characterization of emissions from $CO_2$ and co-emitted species derived from power plants to be used for modelling efforts. Future work should focus on overcoming the limitations currently identified (see Sect. 5) and extending the temporal coverage to more recent years in order to capture, on the one hand, the impact of the decarbonisation efforts that are occurring in several countries and regions such as EU27, UK or USA and, on the other hand, the large uptick in commissioning of new coal power plants that is happening in China (https://www.carbonbrief.org/mapped-worlds-coal-power-plants/). In parallel, other large $CO_2$ emitting industries that are detected by satellite instruments, including cement and steel and iron plants, should be added in future versions of the global point source database.

The current catalogue does not report prediction intervals or standard errors of the estimated emissions for each plant. Hence, uncertainty information is unknown. The comparison against independent inventories indicates good agreement for $CO_2$ but also important discrepancies for $NO_x$ and $SO_x$, highlighting that the co-emitted species estimates and their uncertainty deserve more attention in future research. This is in line with the fact that the estimation of $NO_x$ and $SO_x$ emissions is typically much complex than for $CO_2$, as there are more elements that influence the emission rates, such as combustion conditions, combustion technologies or air pollution control levels implemented in the facilities, among others. Including information on the emission abatement technologies implemented in each plant could help defining more detailed (i.e., technology-level) emission ratios for the estimation of co-emitted species. Unfortunately, none of the other plant-level emission inventories used for comparison include information on co-emitted species, and therefore plant-level emission comparisons were limited to $CO_2$. In this sense, performing intercomparisons against existing satellite-derived point source catalogues (e.g., Beirle et al., 2023; Fioletov et al., 2023) could also help to better constraint bottom-up emissions from co-emitted species.

The present work revealed that information on stack parameters is currently limited not only in developing countries but also in developed regions such as EU27. Efforts should be put to compile this information from individual national environmental permits and centralised it in a European database, at least for the large point sources considered under the European Industrial Emissions Directive (2010/75/EU). Furthermore, flagging the power plants that channel emissions through cooling towers should be assessed to better represents the vertical distribution of these emissions.

Finally, future works will include performing study applications that show the impact of using this emission catalogue on modelling results.

# 7 Author contribution

MG conceived and coordinated the development of the global point source emission database. SD and HDvdG constructed the European point source database and contributed to the analysis and discussion of the results. SE contributed to the construction of the non-European point source database and the construction of the temporal profiles. SE and CT contributed to the construction of the vertical profiles. OJ and CPGP helped conceive the dataset and supervised the work. MG prepared the paper with contributions from all co-authors.

# 8 Competing interests

The authors declare that they have no conflict of interest.

# 9 Acknowledgements

The present work was funded through the CoCO2 project. The CoCO2 project has received funding from the European Union's Horizon 2020 research and innovation programme under grant agreement No 958927. The research has also been supported by the Copernicus Atmosphere Monitoring Service (CAMS), which is implemented by the European Centre for Medium-Range Weather Forecasts (ECMWF) on behalf of the European Commission. We acknowledge support from the VITALISE project (PID2019-108086RA-I00) funded by MCIN/AEI/10.13039/501100011033 from the Agencia Estatal de Investigación (AEI); from the BROWNING project (RTI2018099894-BI00) from the Ministerio de Ciencia, Innovación y Universidades; from the AXA Research Fund; and from the European Research Council (grant no. 773051, FRAGMENT). We would also like to thank the three reviewers for their positive and constructive feedback, which helped improve the quality of the paper.

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
