# Peer review of "A global catalogue of CO2 emissions and co-emitted species from power plants including high resolution vertical and temporal profiles"

_Earth System Science Data, 2023_

## Referee Comment (RC3)

Dear authors of the manuscript, and the handling editor,

This study developed a global high-resolution (spatial and temporal) emission dataset for $CO_2$ and co-emitted species emissions from power plants. The authors have collected and utilized a wide variety of data, including reported data, publicly available data, and commercial data to gather relevant underlying information, such as location, facility profiles, etc. The authors gap filled where needed in order to construct a global database. The dataset was compared to existing datasets that are somewhat old and outdated and thus the assessment did not give us validation.

I have no doubt that this type of data would be useful for improving the representation of emissions in spatially explicit inventories as well as analyses of point source emissions. However, the usefulness is not objectively highlighted in a scientific/objective way, in my opinion. The dataset does, as the authors acknowledged in the manuscript, have a list of limitations. What are the subsequent impacts on the modeling? Given the limitations, users need to assess and decide if they should use this database (or not) for their own research objectives. I do feel this manuscript in the current form would not be helping users. That's the significance of ESSD papers and that distinguishes random reports from high-standard ESSD papers. Some of the underlying data are shared by the outdated datasets that the authors attempted to use as a reference to measure their improvement/advancement they made. The emission calculation is reasonable. Thus, my comments would be just for presentation and uncertainty analysis. Ensuring the following of proper guidelines should be done prior to the review stage.

As an author for ESSD, I assume you've familiarized yourself with the paper preparation guidelines including things we discussed in the ESSD guidelines published as Carlson and Oda (2018). What I discuss in this review aligns with the ESSD guidelines, not based on my own personal opinion. If you ever wonder and need clarification, the guidelines should be able to help you. Thus, I try to keep my review short by not repeating the guidelines. The purpose/hope for the guidelines was not to repeat this type of instruction at the review stage. What I've discussed so fare are something taken care of before the manuscript appeared on ESSD discussion. Ensuring the following of journal guidelines should be done prior to the review stage. As we all know, it is VERY important for ESSD.

Ultimately, I will let the handling editor decide, but should data journals like ESSD publish the datasets just because reviewers think they are useful? What is the real point/value of publishing a paper in ESSD? Why should we/experts in the community review a manuscript for ESSD? The utility and advancement needs to be shown. Otherwise, should we publish all the useful data like a free pass? I hope ESSD continues tackling and addressing this type of challenge and remain as more than a collection of "useful" data. I thank the handling editor for the opportunity to review this manuscript. Reviewing a manuscript for ESSD is always a challenge for us/reviewers because of ESSD's unique nature and the quality it needs to strive for. I would appreciate if you could provide a little more guidance and support to ESSD authors before sending their manuscript for review.

Sincerely,
Tomohiro Oda (toda@usra.edu)

Manuscript format/contents for ESSD

This should have been suggested in an earlier stage of the submission process before sending this to reviewers.  Several things need to be fixed and/or improved, including:

- Data table - the data format table is often a key component of ESSD papers.  You should be able to construct a useful informative table based on the description in Section 2.
- Limitation sections should be there - This might be a preference, but the limitation should stand by itself.  It is great to see the list of limitations that the authors have recognized.  But some of them seem to deserve a little bit of investigation
- Data section - Any update description of data?  Unit? The title is not the same as presented.  Future update plan, and how do you do version control, etc.

I hesitate to pick previously published studies as a good example (which we did not do in the 2018 guidelines), but you should be able to find one.

Title

Database?  High-resolution?  Where is the spatial resolution?  What is the resolution for spatial scale?  This is a point source catalog as I understand.  A bit strange to me.

Evaluation of the dataset

This is often a difficult part for ESSD manuscripts, especially for a study like this.  The authors essentially mentioned that there is no suitable perfect database to compare.  I agreed.  But it does not mean no evaluation is justified.  As ESSD is a science journal, we have to give it a try.  Especially, the dataset's skill/utility and the authors' claims need to be supported objectively.  Here are several points to be discussed and improved with potential suggested approaches:

- Location error (geolocation error) - Location uncertainty, Oda et al. (2019).  Uncertainty is a function of spatial resolution.  Given the potential use of this dataset, this is so important to highlight the unique value of this dataset.  If this is incorporated into modeling, what would be the impact?
- Location error (relevant to emission data and/or atmospheric modeling) - The modeling is not happening at point source level.  Does the CAMS model run at a 1km spatial resolution globally by any chance?  What are the impacts then in concentration?  If the model spatial resolutions are > 0.1 degree, the comparison to any 0.1 degree resolution dataset does provide meaningful practical insights.  The changes the users would expect can be shown.  The paper needs to self-stand and help users to decide if they should use that paper for the analysis or not.  This paper is currently not helping such decisions by providing guidance.  Depending on the focus of evaluation, the seemingly not usable datasets can be used for meaningful evaluation.  Further, some existing datasets could also be useful as the authors provided a list of reasons not to compare to them. Other power plant databases, such as this (https://www.nature.com/articles/s41893-017-0003-y)?
- Impact of the methodological/data differences - CARMA?  Methodological differences.  How is that?  The CARMA did have text and we are aware of how they built the database.  Even quantitative.  Differences cause differences.  If so, what are driving the differences then?  At least characterize them.  Users need to understand the difference to decide if they use this data w/ adequate understanding.
- Injection height - Especially, the vertical injection height section is something that could be more quantitative.  How about the sensitivity of the height to the modeling results?  What are the

potential impacts?  Regional difference?  How can we determine if we should use your data?  Would the inclusion of the injection height be a worth a try given the improvement we could potential get?  Of course, there won't be a single answer given there are many study applications.  However, even one single example of the potential use cases should help the potential users, and more importantly, should support the "usefulness (and degree/level of the usefulness)" that the authors claim objectively.

- Temporal error – Is Crippa et al. (2022) high temporal?  How are those two different (or similar) and what are the potential impacts we see in modeling applications?  It is for 2015, but we could easily imagine that many users would have to use the temporal profiles for different years.  In inventory research and applications, it is easy to imagine a situation to use the 2015 seasonality to other non-COVID times.  Thus, regardless of the 2015 vs. 2018 base year difference, the comparison should provide practical information to guide the future users of this database.

- Errors across different species – This is not just $CO_2$, but other species.  This study should evaluate location in space and time (as mentioned earlier), and also the different errors over different species.  It is hard to imagine that we can use the same error/uncertainty estimates for non-$CO_2$ other than location errors.  Are they the same?

Potential conflict of interest

The main author and the handling editor are working in the same working group under the Global Emission Initiative (GEIA).  It is totally understandable that finding a right handling editor is a challenge for ESSD.  However, the potential conflict of interest should've been clearly indicated, even if it is very minor, in order to protect the scientific integrity that the authors and the handling editor have maintained.

Line by line comments

P2, L45: inadequate – this depends on your purpose/application.  Need elaboration.  If the authors claim that this dataset is more "adequate," you need to objectively demonstrate the skill and utility.  You also have limitations.  Where are the impacts of the limitations in terms of potential errors and uncertainties?  These could be discussed in the database space and also in an applications space (e.g. composition simulations)

P2, L49: 0.1 deg – this is a statement for EDGAR?  Relating to the comment above.  The uncertainty should depend on the spatial resolution.

P2, L50: if so, you could use the 0.1 degree database to compare to the database presented in this manuscript.

P2, L60: not perfectly – as far as I understood.

P3, L71: where are those datasets available?

2.1 Create a table?  Base year 2018 reasonable?  What are the uncertainties associated with that?

L5, L126: Another assumption here. What is the potential impact of this?

2.2 how many of them were done by manual search?

P13, Figure 4 , maybe by region/country?
You should try to examine the utility of the gap filling of any approach by using the existent data,

3.2 spatial errors

P20, L405, not always. See Oda et al. 2019

P21, L414, R2 0.8 is concerning if at plant level.  Is it adequate?
P21, L428, if CO2 is this bad, other compounds could be way worse.
P21, L442, CH4 needs to be evaluated.

P22, L443, why?

L26, L504. ESSD paper should stand by itself.  Describe what the system does for the temporal modeling.

Figure 11, What are the uncertainties around the temporal estimates?

4 data Name is different. What is the readme? That's what we need here, too.

5.1 limitation should be before the conclusion and discussed in more detail.  Some of them could be examined in this study.

P34, L675 not just for inverse modeling. This is a huge limitation for the potential users of this product.

Supplemental information

Figure 2. The correlation should be high.  The high correlation does not really tell us any practical information regarding the performance of the dataset.  How about using RMSE for location and emission intensity as a metric for potential improvements brought by this dataset?

References:
- Carlson and Oda (2018) https://essd.copernicus.org/articles/10/2275/2018/
- Oda et al. (2019) https://link.springer.com/article/10.1007/s11027-019-09877-2
- Crippa et al. (2022) https://www.nature.com/articles/s41597-020-0462-2

---

## Author Comment (AC1)

We would like to thank the three reviewers for their positive and constructive feedback, which helped improve the quality of the paper The review comments have been helpful in pointing out parts that required further improvements. Below we address specific issues mentioned by the reviewers point by point. The manuscript has been updated accordingly.

**Anonymous Referee #1**

This work presents a global emission dataset of CO2 and co-emitted species including NOx, SO2, CO and CH4 for power plants in 2018. The database assigned more than 16000 individual facilities with geographical locations. Temporal proxies and vertical spatial distribution factors were also provided. The data is publicly available to the community and can be used to support atmospheric modeling activities. Overall, the manuscript is written in a logical way. But lots of details regarding how the raw data is reported, how the gap filling is done, and how to harmonize various information are not clarified. On the other hand, authors should point out the assumptions clearly especially when "lending" information from one power plant to another. I suggest re-visiting the methodology section carefully with clearer statement. The figures and tables also need to be updated.

The detailed comments are listed below.

**Line 10: can you explain why using "catalogue" instead of "inventory" or just "dataset"?**

While doing a literature review, we found several papers in which similar products to the one described in this work – collection of point sources with associated geographical location and emissions - were presented as a "catalogue" (e.g., Beirle et al., 2021 and 2023; Fioletov et al., 2023) and subsequently we thought it would be appropriate to refer to this work in the same way. We also thought the concept "catalogue" is adequate for this product as we do not only provide location and emissions per point source, but also associated temporal and vertical profiles.

Beirle, S., Borger, C., Dörner, S., Eskes, H., Kumar, V., De Laat, A. and Wagner, T.: Catalog of NOx emissions from point sources as derived from the divergence of the NO2 flux for TROPOMI, Earth Syst. Sci. Data, 13(6), 2995–3012, 510 doi:10.5194/essd-13-2995-2021, 2021.

Beirle, S., Borger, C., Jost, A., and Wagner, T.: Improved catalog of NOx point source emissions (version 2), Earth Syst. Sci. Data, 15, 3051–3073, https://doi.org/10.5194/essd-15-3051-2023, 2023

Fioletov, V. E., McLinden, C. A., Griffin, D., Abboud, I., Krotkov, N., Leonard, P. J. T., Li, C., Joiner, J., Theys, N., and Carn, S.: Version 2 of the global catalogue of large anthropogenic and volcanic SO2 sources and emissions derived from satellite measurements, Earth Syst. Sci. Data, 15, 75–93, https://doi.org/10.5194/essd-15-75-2023, 2023.

**Line 13: what emission ratios? It'll bring confusion to readers when you mention "ratios" in an abstract but not specify what it is.**

We have specified what do we mean by emission ratios in the abstract as follows:

"*The construction of the database follows a bottom-up approach, which combines plant-specific information with national energy consumption statistics and fuel-dependent emission factors for CO₂ and emission ratios for co-emitted species (e.g., amount of NOx emitted relative to CO2; NOx/CO2).* " (lines 12 to 13 of the revised manuscript)

**Line 14: as my understanding, the temporal and vertical distribution profiles are NOT facility-specific. It's shared within a country. Please revise this.**

The vertical distribution profiles are facility-specific, as they are estimated making use of gridded meteorological information and stack parameters that can vary per facility. However, temporal profiles are indeed country and fuel-dependent, so not facility dependent. We have revised the text to clarify this aspect:

"Each facility is linked to a country- and fuel-dependent temporal profile (i.e., monthly, day-of-the-week and hourly) and plant-level vertical profile" (lines 14 to 15 of the revised manuscript)

A similar clarification was introduced in the conclusions section (lines 1365 to 1366 of the revised manuscript)

**Line 51: can you add comparisons with GPED since they are bottom-up emission estimates at a facility-level.**

We have added a comparison between our work and the GPED database at the facility-level for the top 50 emitters reported by each database (a total of 76 plants). The discussion of the results has been included in Sect 3.2.1 of the revised manuscript (lines 812 to 845 of the revised manuscript) together with a bar plot showing the plant-to-plant CO2 annual emission comparison (Figure 10 of the revised manuscript).

**Section 2.1: I'm confused by the statement of "selection of facilities". Where are the raw data reported to and what's your criteria of selecting the data? Please specify.**

The word "selection" is not appropriate. We have changed for "compilation", as what we did was to compile information of power plants from different available databases.

**Line 80 – 90: Lots of information but not clear. Can you draw a global map with the data sources labeled? Or prepare a table?**

Authors agree with the reviewer. We have included a summary table (Table 1 of the revised manuscript) describing the main characteristics of the power plant datasets considered to construct the inventory and specifying for which countries and fuels we used each one of them.

**Line 80 – 90: Which year of these data are available?**

The reference year of each database considered has been added to the summary table mentioned in the previous comment. For E-PRTR_v18, LCP_v5.2, IRD_v7 and eGRIDv2018 the information reported is for the year 2018 (other versions of the same databases will report the data for other years). For the GCPTv2021_01 and GGPTv2021_02 the base year is 2021, but we only selected those power plants that were operating in 2018 considering the information on status and start/retired year. For the other datasets, which base years are 2021/2022 for most of the cases, we assumed all power plants included were already operating in 2018, as no information on the status or start/retired year is reported. This information has been also added in a footnote in the new Table 1.

**Line 95: what's the criterion when there are conflicts between different dataset? For example, how do you know two power plants are the same between two dataset? What if different capacities are reported for the "same" power plants from different dataset?**

For the non-EU catalogue, our first priority was to use national datasets, as they are the ones that are more complete (e.g., inclusion of auto-producers and small-to-medium facilities, more precise geographical locations). We managed to obtain this information for selected countries and/or fuels, including eGRIDv2018 for all types of power plants in the USA, MIEE for waste-to-energy plants in China and DEDE for biomass power plants in Thailand. For the rest of the countries/fuels, we tried to consider a unique dataset per fuel, to keep consistency. For instance, for coal-fired power plants we considered the GCPTv2021_01 dataset for all non-EU countries except USA (eGRIDv2018) as it is the most updated and complete coal-fired power plant dataset (in terms of geographic location of the plants, number of plants reported per country and completeness of installed capacity). This conclusion was obtained after manually comparing the information provided by the different coal-fired power plant databases (e.g., GCPTv2021_01, GPPDv1.3.0, IndustryAbout) in selected countries that are among the top 30 countries in terms of installed power generation capacity and that are representative of coal (i.e., South Africa, Japan, Taiwan, Kazakhstan, Australia, Vietnam and Turkey).

For natural gas, oil, biomass and waste we could not follow this strategy, as the datasets that best represent these fuels were not covering all countries worldwide. For instance, in the case of natural gas, and after manual comparisons in countries that are representative of this type of power plants (i.e., Japan, Oman, Thailand, Bahrain), we concluded that GGPTv2021_02 was the most accurate dataset. Nevertheless, this version of the database misses several countries, and therefore we had to combine it with GPPDv1.3.0 and IndustryAbout. We compared the number of facilities and installed capacity reported by each of the two datasets in the remaining countries (focussing always on the top 30 countries in terms of installed capacity), and if results were the same of very similar, we selected GPPDv1.3.0. For countries in which IndustryAbout reported a much larger number of facilities or value of installed capacity, we then manually checked and compared the information with third independent sources (e.g., internet searches to check installed capacity or location of a specific power plant) and if the results where satisfactory we considered IndustryAbout for that country. Similar strategies were followed for oil, biomass and waste.

As mentioned in a previous comment, Table 1 of the revised manuscript summarises the main characteristics of the power plant datasets considered to construct the inventory and specifies for which countries and fuels we used each one of them.

**Line 135 – 140: emission controls of NOx and SOx on power plants are widely applied over developed countries like the U.S. and developing countries like China. How do you take the emission control into consideration in your emission estimation?**

We have reviewed our approach to estimate emissions of NOx, SOx and CO by replacing the current fuel-dependent emissions ratios by a new set of fuel- and region/country-dependent emission ratios derived from the GAINS emission inventory (Amann et al., 2011; Klimont et al., 2017), which takes into account the heterogenous implementation of emission control restrictions in power plants across countries/regions. The new ratios were computed for 23 world regions/countries and the three fuel categories reported by GAINS for the power plants sector: coal, natural gas and oil. The emission ratios were estimated as an average of the emissions reported by GAINS for the years 2015 and 2020, since they are the closest to the reference year of our catalogue (2018). We included a figure (Figure 2 of the revised manuscript) showing the differences between coal-fired plants ratios computed for selected countries to illustrate the capability of GAINS in reflecting differences between national emission legislations associated to the power generation industry. For the other two fuel categories considered in the present catalogue (biomass and waste), we estimated a new set of fuel-dependent emission ratios considering the European power plant database based on E-PRTR. In the previous version of the manuscript, the emission ratios considered for these two power plant categories were derived from the US eGrid database. However, we decided to switch to the European database since the total number of plants considered to derive the emission ratios is much larger (i.e., 153 versus 319 for biomass power plants and 62 versus 210 for waste-to-energy plants). Regardless of this change, for these two fuels we still assume that all countries share the same ratios, which may introduce some uncertainty in the results. However, it is important to note that the contribution of these two fuels to the total combustion related electricity generation in non-European countries is rather residual (less than 5 %; IEA, 2021a). A description of the methodology and datasets considered to derive the new set of fuel- and country/region-dependent emission ratios has been included in the revised version of the manuscript (lines 275 to 315 of the revised manuscript).

The country-level NOx and SOx emissions estimated using the updated set of emission ratios were compared against independent national emission inventories, finding in general a better agreement than with our previous estimates. For instance, when using the previous emission ratios, our SOx emission estimates for India and South Africa were 5.6 and 14.4 times lower than the official inventories, while with the new emission ratios the level of disagreement has significantly decreased (i.e., our SOx estimates are now 1.1 times higher in India and 1.2 times lower in South Africa). A similar improvement is observed with regards to NOx emissions. The section of the manuscript describing the comparison between country-level emission estimates (Section 3.2.3 of the revised manuscript) and

corresponding figures (Figures 12 of the revised manuscript) have been updated according to the new results.

Despite the improvements observed in the estimation of co-emitted species, we still find some discrepancies between our estimates and national independent inventories. For instance, our NOx emissions in South Africa and Turkey are 2 (4 with the previous version of the emission ratios) and 2.5 (4 with the previous version of the emission ratios) times lower than the corresponding national estimates, respectively. We attribute these discrepancies to the fact that pollution abatement controls do not only differ by country or regions, but also across power plants within the same country. The use of not only fuel- and country-dependent, but also technology-dependent emission ratios could potentially help in reducing these discrepancies. However, information on the technology implemented in each power plant is not currently included in our catalogue and should be added in future versions. We have introduced this point in the "limitations" and "future perspectives" sections of our revised manuscript.

*IEA: International Energy Agency. World Energy Balances 2021 Edition. Available at: https://www.iea.org/data-and-statistics/data-product/world-energy-balances (last access: November 2022), 2021a.*

*Amann, M., Bertok, I., Borken-Kleefeld, J., Cofala, J., Heyes, C., Höglund-Isaksson, L., Klimont, Z., Nguyen, B., Posch, M., Rafaj, P., Sander, R., Schöpp, W., Wagner, F., Winiwarter, W.: Cost-effective control of air quality and greenhouse gases in Europe: modeling and policy applications. Environmental Modelling and Software 26, 1489–1501. doi:10.1016/j.envsoft.2011.07.012, 2011.*

*Klimont, Z., Kupiainen, K., Heyes, C., Purohit, P., Cofala, J., Rafaj, P., Borken-Kleefeld, J. and Schöpp, W.: Global anthropogenic emissions of particulate matter including black carbon. Atmos. Chem. Phys., 17, 8681-8723, https://doi.org/10.5194/acp-17-8681-2017, 2017.*

**Line 150 – 180: please specify the assumption when gap-filling the emission values step-by-step.**

We added additional explanation and a figure to the EU data gap filling description to make our assumptions more explicit (figure 1 and section 2.3.1 of the revised manuscript). Moreover, we have included information on the shares of the final EU emissions that are directly derived from E-PRTR reporting and from the five gap filling steps (Tables 3 and 4 of the revised manuscript).

**Line 192: The ratios can vary significantly by countries and fuel type due to pollution abatement strategies. It can introduce large uncertainties when assuming each country can share the same SOx/CO2 ratios for all power plant units.**

We agree with the reviewer. As mentioned in a previous comment, we have reviewed our approach to estimate emissions of NOx, SOx and CO by replacing the current fuel-dependent emissions ratios by a new set of fuel- and region/country-dependent emission ratios derived from the GAINS emission inventory (Amann et al., 2011; Klimont et al., 2017), which takes into account the heterogenous implementation of emission control restrictions in power plants across countries/regions. The description of the methodology to derive the new emission ratios has been included in the revised version of the manuscript (lines 275 to 315 of the revised manuscript). We have also updated the results concerning the comparison between country-level NOx and SOx emissions (Section 3.2.3 of the revised manuscript), finding a better agreement between our new estimates and independent national inventories than with the previous results.

Amann, M., Bertok, I., Borken-Kleefeld, J., Cofala, J., Heyes, C., Höglund-Isaksson, L., Klimont, Z., Nguyen, B., Posch, M., Rafaj, P., Sander, R., Schöpp, W., Wagner, F., Winiwarter, W.: Cost-effective control of air quality and greenhouse gases in Europe: modeling and policy applications. Environmental Modelling and Software 26, 1489–1501. doi:10.1016/j.envsoft.2011.07.012, 2011.

Klimont, Z., Kupiainen, K., Heyes, C., Purohit, P., Cofala, J., Rafaj, P., Borken-Kleefeld, J. and Schöpp, W.: Global anthropogenic emissions of particulate matter including black carbon. Atmos. Chem. Phys., 17, 8681-8723, https://doi.org/10.5194/acp-17-8681-2017, 2017.

**Line 235: So, the temporal profiles are not facility-specific. It's country and fuel-dependent.**

Yes. As mentioned in a previous comment, we have clarified this aspect in the abstract and conclusions sections.

**Line 321: this dataset doesn't include hydropower, right?**

No. The inventory described in the manuscript includes power plants that burn coal, natural gas, oil, solid biomass and municipal/industrial solid waste.

**Figure 5: please zoom in a little bit of the map (duplicate US and East Asia maps are shown). The circles are not clear. I suggest using colors to scale the annual emissions, but with various symbols (rectangles, circles, etc.) to denote the fuel type.**

We have reviewed the figure to remove duplicate US and East Asia maps and to increase its resolution. We decided to keep the original visualization, with circle sizes indicating annual emissions and colour indicating fuel type. However, we have refined the scale of the circle sizes used, and we have also added zooms over different areas of interest (e.g., Europe, Middle East, Asia) to facilitate the analysis of the results. In the revised version of the manuscript, we have divided the former Figure 5 into two new figures (Figure 7 for $CO_2$ and Figure 8 for $NO_x$).

**Line 433: please double check this, or give references.**

We double check it and in fact the MEICv2 electricity and heat production sector includes emissions from auto-producers (Li et al., 2017; Zheng et al., 2018). For all national inventories reported under the UNFCCC, the energy sector only includes public producers, as described in the IPCC guidelines (IPCC, 2019). We have added this reference in the text (line 895 of the revised manuscript)

*IPCC: 2019 Refinement to the 2006 IPCC Guidelines for National Greenhouse Gas Inventories, Calvo Buendia, E., Tanabe, K., Kranjc, A., Baasansuren, J., Fukuda, M., Ngarize, S., Osako, A., Pyrozhenko, Y., Shermanau, P. and Federici, S. (eds). Published: IPCC, Switzerland, available at: https://www.ipcc-nggip.iges.or.jp/public/2019rf/index.html (last access: October 2022), 2019.*

*Li, M., Liu, H., Geng, G., Hong, C., Liu, F., Song, Y., Tong, D., Zheng, B., Cui, H., Man, H., Zhang, Q., and He, K.: Anthropogenic emission inventories in China: a review, Natl. Sci. Rev., 4, 834-866, doi: 10.1093/nsr/nwx150, 2017.*

*Zheng, B., Tong, D., Li, M., Liu, F., Hong, C., Geng, G., Li, H., Li, X., Peng, L., Qi, J., Yan, L., Zhang, Y., Zhao, H., Zheng, Y., He, K., and Zhang, Q.: Trends in China's anthropogenic emissions since 2010 as the consequence of clean air actions, Atmos. Chem. Phys., 18, 14095-14111, doi: 10.5194/acp-18-14095-2018, 2018.*

**Figure 10: the figure is hard to understand.**

We have adjusted the dimensions of the figure as well as the thickness of the lines plotted on it to facilitate its interpretation and visualize better hourly variations in emissions from each plant (Figure 15 of the revised manuscript).

**Anonymous Referee #2**

This study constructed a global inventory of carbon and pollutant emissions from over 16,000 power plants in 2018, utilizing multiple data sources. Such work is foundational and significant. Despite the author's detailed methodological explanations and my agreement with his very good work on time allocation, I still have two methodological concerns:

**I have strong doubts about the accuracy of air pollutant emissions at the power plant level due to lack of data sources. The article attempts to fill the data gaps by using the proportional changes between pollutants and CO2. While this method may be reliable for CO and CH4, it may encounter problems when applied to SO2 and NOx due to variations in air pollution control levels. Need to explain more or switch to a better approach.**

We added additional explanations and a figure to the EU data gap filling description to make our assumptions more explicit (section 2.3.1 and Figure 1 of the revised manuscript). Moreover, we have included information on the shares of the final EU emissions that are directly derived from E-PRTR reporting and from the five gap filling steps (Tables 3 and 4 of the revised manuscript). The results indicate that the contribution of gap filled emissions is rather low for CO2, NOx and SO2 emissions (less than 10%), but more substantial for CO and CH4 (more than 60% of total emissions from gap filling). We agree with the reviewer that the approach considered may have limitations specially for SO2 and NOx due to variations in air pollution control levels among power plants. However, and considering that the gap filled emission values contribute less than 10% to these two pollutants, we believe the level of uncertainty introduced by this method is rather limited.

**The estimation of NOx and SO2 emissions in regions outside the United States and Europe based on the proportion of CO2 and pollutants in the eGRID database overlooks the differences in pollution control in different regions. Applying air pollution control levels from the United States to other regions may result in significant deviations.**

We completely agree with the reviewer. We have reviewed our approach to estimate emissions of NOx, SOx and CO by replacing the current fuel-dependent emissions ratios by a new set of fuel- and region/country-dependent emission ratios derived from the GAINS emission inventory (Amann et al., 2011; Klimont et al., 2017), which takes into account the heterogenous implementation of emission control restrictions in power plants across countries/regions. The new ratios were computed for the three fuel categories reported by GAINS for the power plants sector: coal, natural gas and oil. The emission ratios were estimated as an average of the emissions reported by GAINS for the years 2015 and 2020, since they are the closest to the reference year of our catalogue (2018). We included a figure (Figure 2 of the revised manuscript) showing the differences between coal-fired plants ratios computed for selected countries to illustrate the capability of GAINS in reflecting differences between national emission legislations associated to the power generation industry. For the other two fuel categories considered in the present catalogue (biomass and waste), we estimated a new set of fuel-dependent emission ratios considering the European power plant database based on E-PRTR. In the previous version of the manuscript, the emission ratios considered for these two power plant categories were derived from the US eGrid database. However, we decided to switch to the European database since the total number of plants considered to derive the emission ratios is much larger (i.e., 153 versus 319 for biomass power plants and 62 versus 210 for waste-to-energy plants). Regardless of this change, for these two fuels we still assume that all countries share the same ratios, which may introduce some uncertainty in the results. However, it is important to note that the contribution of these two fuels to the total combustion related electricity generation in non-European countries is rather residual (less than 5 %; IEA, 2021a). A description of the methodology and datasets considered to derive the new set of fuel- and country/region-dependent emission ratios has been included in the revised version of the manuscript (lines 275 to 315 of the revised manuscript).

The country-level NOx and SOx emissions estimated using the updated set of emission ratios were compared against independent national emission inventories, finding in general a better agreement than with our previous estimates. For instance, when using the previous emission ratios, our SOx emission estimates for India and South Africa were 5.6 and 14.4 times lower than the official inventories, while with the new emission ratios the level of disagreement has significantly decreased (i.e., our SOx estimates are now 1.1 times higher in India and 1.2 times lower in South Africa). A similar improvement is observed with regards to NOx emissions. The section of the manuscript describing the comparison between country-level emission estimates (Section 3.2.3 of the revised manuscript) and corresponding figures (Figure 12 of the revised manuscript) have been updated according to the new results.

Despite the improvements observed in the estimation of co-emitted species, we still find some discrepancies between our estimates and national independent inventories. For instance, our NOx emissions in South Africa and Turkey are 2 (4 with the previous version of the emission ratios) and 2.5 (4 with the previous version of the emission ratios) times lower than the corresponding national estimates, respectively. We attribute these discrepancies to the fact that pollution abatement controls do not only differ by country or regions, but also across power plants within the same country. The use of not only fuel- and country-dependent, but also technology-dependent emission ratios could potentially help in reducing these discrepancies. However, information on the technology implemented in each power plant is not currently included in our catalogue and should be added in future versions. We have introduced this point in the "limitations" and "future perspectives" sections of our revised manuscript.

*IEA: International Energy Agency. World Energy Balances 2021 Edition. Available at: https://www.iea.org/data-and-statistics/data-product/world-energy-balances (last access: November 2022), 2021a.*

*Amann, M., Bertok, I., Borken-Kleefeld, J., Cofala, J., Heyes, C., Höglund-Isaksson, L., Klimont, Z., Nguyen, B., Posch, M., Rafaj, P., Sander, R., Schöpp, W., Wagner, F., Winiwarter, W.: Cost-effective control of air quality and greenhouse gases in Europe: modeling and policy applications. Environmental Modelling and Software 26, 1489–1501. doi:10.1016/j.envsoft.2011.07.012, 2011.*

*Klimont, Z., Kupiainen, K., Heyes, C., Purohit, P., Cofala, J., Rafaj, P., Borken-Kleefeld, J. and Schöpp, W.: Global anthropogenic emissions of particulate matter including black carbon. Atmos. Chem. Phys., 17, 8681-8723, https://doi.org/10.5194/acp-17-8681-2017, 2017.*

**Other minor concerns include:**

**The resolution of Figure 5 is too low, making it difficult to interpret.**

We have reviewed the figure to increase its resolution. Moreover, we have refined the scale of the circle sizes used, and we have also added zooms over different areas of interest (e.g., Europe, Middle East, Asia) to facilitate the analysis of the results. In the revised version of the manuscript, we have divided the former Figure 5 into two new figures (Figure 7 for $CO_2$ and Figure 8 for $NO_x$).

**It would be beneficial to include comparisons of other pollutants in Figure 6, particularly NOx and SO2.**

Unfortunately, the CARMAv3 database only reports $CO_2$ emissions, and therefore comparisons with other pollutants cannot be done. Note that based on a comment from reviewer # 1 we have added an addition plant-level comparisons against the GID database (but again, only for $CO_2$ as no other pollutants are reported). Related to this limitation, we have added a new point in the "Future perspective" section mentioning that future works should also include comparisons against existing satellite-based point source catalogues, as they could provide additional insights to better represent the emissions from these co-emitted species in future versions of the dataset (lines 1390 to 1392 of the revised manuscript).

**Can the current data sources and methods support the extension of the analysis to multiple years?**

Yes. Using the current data sources and methods, the catalogue could be extended to multiple years, with a temporal coverage extending from 2007 until 2021. Covering years before 2007 would not be possible with the current approach, as the E-PRTR database does not report information on European point sources before that year. Similarly, extending the dataset to more recent years (i.e., 2022 and 2023) is still not possible, as most of the databases considered (i.e., E-PRTR, IEA, eGRID) are published with a two-year time lag. The GCPTv2021_01 and GGPTv2021_02 databases provide information on the start/retire years of each facility, which could be used to define which facilities are operating for each year.

**The handling of temporal allocation methods is commendable.**

We thank the reviewer for appreciating the work devoted to construct the temporal profiles.

**Tomohiro Oda**

Dear authors of the manuscript, and the handling editor,

This study developed a global high-resolution (spatial and temporal) emission dataset for $CO_2$ and co-emitted species emissions from power plants. The authors have collected and utilized a wide variety of data, including reported data, publicly available data, and commercial data to gather relevant underlying information, such as location, facility profiles, etc. The authors gap filled where needed in order to construct a global database.

**The dataset was compared to existing datasets that are somewhat old and outdated and thus the assessment did not give us validation.**

The only dataset that we use for comparison that is old and outdated is CARMAv3. As mentioned in the manuscript, we decided to perform this comparison because despite not being longer maintained, CARMAv3 is still used as a proxy for the spatial representation of power plant emissions in several state-of-the-art inventories and modelling systems. The other datasets considered for comparison at the plant (i.e., national point source inventories for India and Canada) and country level (i.e., EDGARv7 and national emission inventories) are updated.

Following the reviewer's advice, In the revised version of the manuscript we have added a comparison between our work and the Global Infrastructure emission Database (GID; Tong et al., 2018) at the facility-level for the top 50 emitters reported by each database (a total of 76 plants). The discussion of the results has been included in Sect 3.2.1 of the revised manuscript (lines 811 to 840 of the revised manuscript) together with a bar plot showing the plant-to-plant $CO_2$ annual emission comparison (Figure 10 of the revised manuscript). In the discussion of this comparison, we have also considered the results reported by Grant et al., (2021).

Grant, D., Zelinka, D., Mitova, S.: Reducing $CO_2$ emissions by targeting the world's hyper-polluting power plants, Environ. Res. Lett., 16, 094022, https://doi.org/10.1088/1748-9326/ac13f1, 2021.

Tong, D., Q. Zhang, S. J. Davis, F. Liu, B. Zheng, G. Geng, T. Xue, M. Li, C. Hong, Z. Lu, D. G. Streets, D. Guan, and K. He: Targeted emission reductions from global super-polluting power plant units, Nat. Sustain., 1, 59-68, https://doi.org/10.1038/s41893-017-0003-y, 2018.

**I have no doubt that this type of data would be useful for improving the representation of emissions in spatially explicit inventories as well as analyses of point source emissions. However, the usefulness is not objectively highlighted in a scientific/objective way, in my opinion. The dataset does, as the authors acknowledged in the manuscript, have a list of limitations. What are the subsequent impacts on the modeling? Given the limitations, users need to assess and decide if they should use this database (or not) for their own research objectives. I do feel this manuscript in the current form would not be helping users. That's the significance of ESSD papers and that distinguishes random reports from high standard ESSD papers. Some of the underlying data are shared by the outdated datasets that the authors attempted to use as a reference to measure their improvement/advancement they made. The emission calculation is reasonable. Thus, my comments would be just for presentation and uncertainty analysis. Ensuring the following of proper guidelines should be done prior to the review stage.**

The impact of the dataset (e.g., geolocation of emissions, temporal profiles, vertical profiles) on modelling results is out of the scope of this manuscript. The main objective of this paper is to describe the catalogue and methodologies considered to develop it, as well as to compare the emission results against independent emission datasets. In this sense, in the new version of the manuscript the database is compared against several independent datasets, including:

- gridded, country-level (i.e., EDGAR and various national inventories) and plant-level (i.e., CARMA, GID, Grant et al., 2021) datasets (more details given in the response to other comments below)
- The temporal profiles are now compared against the EDGAR temporal profiles and plant-level profiles (more details given in the response to other comments below)
- The vertical injection height profiles are now compared against the profile proposed by TNO for the public energy sector in the Copernicus CAMS-REG inventory (Kuenen et al., 2021), which is widely used among the air quality modelling community in Europe (more details given in the response to other comments below)

We believe including all these emission intercomparison exercises in the manuscript already provides the reader an idea of the level of usefulness of the catalogue. Adding study applications that show the impact of using this emission catalogue instead of other state-of-the-art inventories (e.g., EDGAR) or emission proxies (e.g., independent vertical profiles) on $CO_2$ modelling results is an independent task that we plan to tackle separately in a future publication. There is not a single answer given the fact that there are many study applications, and therefore a complete modelling framework should be set up to properly answer this point. We have included a reflection on this point in the "Future perspective" section of the manuscript (lines 1398 to 1399 of the revised manuscript)

As an author for ESSD, I assume you've familiarized yourself with the paper preparation guidelines including things we discussed in the ESSD guidelines published as Carlson and Oda (2018). What I discuss in this review aligns with the ESSD guidelines, not based on my own personal opinion. If you ever wonder and need clarification, the guidelines should be able to help you. Thus, I try to keep my review short by not repeating the guidelines. The purpose/hope for the guidelines was not to repeat this type of instruction at the review stage. What I've discussed so fare are something taken care of before the manuscript appeared on ESSD discussion. Ensuring the following of journal guidelines should be done prior to the review stage. As we all know, it is VERY important for ESSD.

We agree with the reviewer on the importance of following the ESSD manuscript preparation guidelines. In this sense, we have implemented all the suggestions made by the reviewer related to the manuscript format and contents (see answers to comments below).

Ultimately, I will let the handling editor decide, but should data journals like ESSD publish the datasets just because reviewers think they are useful? What is the real point/value of publishing a paper in ESSD? Why should we/experts in the community review a manuscript for ESSD? The utility and advancement needs to be shown. Otherwise, should we publish all the useful data like a free pass? I hope ESSD continues tackling and addressing this type of challenge and remain as more than a collection of "useful" data. I thank the handling editor for the opportunity to review this manuscript. Reviewing a manuscript for ESSD is always a challenge for us/reviewers because of ESSD's unique nature and the quality it needs to strive for. I would appreciate if you could provide a little more guidance and support to ESSD authors before sending their manuscript for review

We understand this comment is addressed to the editor and not the authors of the manuscript, and therefore we assume no answer from our side is expected.

Manuscript format/contents for ESSD

This should have been suggested in an earlier stage of the submission process before sending this to reviewers. Several things need to be fixed and/or improved, including:

• Data table - the data format table is often a key component of ESSD papers. You should be able to construct a useful informative table based on the description in Section 2.

We have included a summary table (Table 1 of the revised manuscript) describing the main characteristics of the power plant datasets considered to construct the inventory and specifying for

which countries and fuels we used each one of them. We have also improved the description of the dataset in the Data availability section, as mentioned in another comment in more detail.

**• Limitation sections should be there - This might be a preference, but the limitation should stand by itself. It is great to see the list of limitations that the authors have recognized. But some of them seem to deserve a little bit of investigation**

Following the reviewer's recommendation, we have moved the "Limitations of the dataset" section before the conclusions section (Section 5 of the revised manuscript). We have examined in more detail the limitations related to the temporal profiles, gap-filling procedure and use of emission ratios to estimated co-emitted emissions, as discussed in other comments.

**• Data section - Any update description of data? Unit? The title is not the same as presented. Future update plan, and how do you do version control, etc. I hesitate to pick previously published studies as a good example (which we did not do in the 2018 guidelines), but you should be able to find one.**

We modified the title to be consistent with the one presented. We have also updated the description of the different information provided in the catalogue, including units, among others. We also included the dataset in a Gitlab version control system (https://earth.bsc.es/gitlab/mguevara/global_catalogue_power_plant_emissions/, last access September 2023). To update and improve the "Data Availability" section we followed the example of this published study: https://essd.copernicus.org/articles/12/3039/2020/#section4

**Title**

**Database? High-resolution? Where is the spatial resolution? What is the resolution for spatial scale? This is a point source catalog as I understand. A bit strange to me.**

The concept "high-resolution" for the spatial scale is referring to the fact that the catalogue includes not only annual emissions at the exact geographical location, but also facility-level vertical profiles. We also included the concept "high-resolution" in the title as the catalogue includes country- and fuel-dependent temporal profiles that are linked to the individual plants and that the users can use to break down original annual emissions onto the hourly scale. We have re-defined the tittle to better communicate these two concepts and avoid confusions:

*"A global catalogue of CO2 emissions and co-emitted species from power plants including high resolution vertical and temporal profiles"*

**Evaluation of the dataset**

**This is often a difficult part for ESSD manuscripts, especially for a study like this. The authors essentially mentioned that there is no suitable perfect database to compare. I agreed. But it does not mean no evaluation is justified. As ESSD is a science journal, we have to give it a try. Especially, the dataset's skill/utility and the authors' claims need to be supported objectively. Here are several points to be discussed and improved with potential suggested approaches:**

**Location error (geolocation error) - Location uncertainty, Oda et al. (2019). Uncertainty is a function of spatial resolution. Given the potential use of this dataset, this is so important to highlight the unique value of this dataset. If this is incorporated into modeling, what would be the impact?**

**Location error (relevant to emission data and/or atmospheric modeling) - The modeling is not happening at point source level. Does the CAMS model run at a 1km spatial resolution globally by any chance? What are the impacts then in concentration? If the model spatial resolutions are > 0.1 degree, the comparison to any 0.1 degree resolution dataset does provide meaningful practical insights. The changes the users would expect can be shown. The paper needs to self-stand and help users to decide if they should use that paper for the analysis or not. This paper is currently not helping such decisions by providing guidance. Depending on the focus of evaluation, the seemingly not usable datasets can be used for meaningful evaluation.**

We agree with the reviewer that a comparison to an independent gridded emission dataset can provide meaningful practical insights to users wanting to use this dataset for modelling purposes, including the CAMS model as it currently runs at a maximum spatial resolution of 9kmx9km (Agustí-Panareda et al., 2022). We added the $CO_2$ annual emissions of our point source catalogue onto the same 0.1x0.1 degree grid as the EDGARv7 $CO_2$ inventory and evaluated the spatial correlations between the two gridded datasets. The discussion of the results, together with a figure showing the spatial correlation obtained per country were included in a new subsection (Section 3.2.2 Grid cell level) of the revised manuscript.

*Agustí-Panareda, A., McNorton, J., Balsamo, G., Baier, B.C., Bousserez, N., Boussetta, S., Brunner, D., Chevallier, F., Choulga, M., Diamantakis, M., Engelen, R., Flemming, J., Granier, C., Guevara, M., Denier van der Gon, H., Elguindi, N., Haussaire, J.-M., Jung, M., Janssens-Maenhout, G., Kivi, R., Massart, S., Papale, D., Parrington, M., Razinger, M., Sweeney, C., Vermeulen, A., and Walther, S.: Global nature run data with realistic high-resolution carbon weather for the year of the Paris Agreement. Sci Data, 9, 160, https://doi.org/10.1038/s41597-022-01228-2, 2022.*

**Further, some existing datasets could also be useful as the authors provided a list of reasons not to compare to them. Other power plant databases, such as this https://www.nature.com/articles/s41893-017-0003-y?**

In the revised version of the manuscript we have added a comparison between our work and the Global Infrastructure emission Database (GID; Tong et al., 2018) at the facility-level for the top 50 emitters reported by each database (a total of 76 plants). The discussion of the results has been included in Sect 3.2.1 of the revised manuscript (lines 811 to 845 of the revised manuscript) together with a bar plot showing the plant-to-plant CO2 annual emission comparison (Figure 10 of the revised manuscript). In the discussion of this comparison we have also considered the results reported Grant et al., (2021).

*Grant, D., Zelinka, D., Mitova, S.: Reducing CO2 emissions by targeting the world's hyper-polluting power plants, Environ. Res. Lett., 16, 094022, https://doi.org/10.1088/1748-9326/ac13f1, 2021.*

*Tong, D., Q. Zhang, S. J. Davis, F. Liu, B. Zheng, G. Geng, T. Xue, M. Li, C. Hong, Z. Lu, D. G. Streets, D. Guan, and K. He: Targeted emission reductions from global super-polluting power plant units, Nat. Sustain., 1, 59-68, https://doi.org/10.1038/s41893-017-0003-y, 2018.*

**Impact of the methodological/data differences - CARMA? Methodological differences. How is that? The CARMA did have text and we are aware of how they built the database. Even quantitative. Differences cause differences. If so, what are driving the differences then? At least characterize them. Users need to understand the difference to decide if they use this data w/ adequate understanding.**

In section 3.2.1 we perform a comparison between CARMAv3 and our catalogue that covers multiple aspects, including:

- comparison between national emissions
- comparison between plant-level emissions
- comparison between plant-level locations

When discussing these three aspects, we clarify the potential methodological reasons that could be driving the differences found between datasets, which include:

- CARMAv3 is mostly based on Platts WEPP, which contains many small size auto-producer units (e.g., boilers located in commercial and institutional buildings such as hospitals or airports) with very low emission levels associated to them that are not considered in the present work.
- CARMAv3's reference year is 2009, while in the present catalogue the reference year is 2018. Several power plants included in CARMAv3 were shut down during the last decade or suffered a fuel conversion from coal to natural gas.
- CARMAv3 exclude biogenic CO2 emissions from biofuels.

- In all countries except for USA, Canada, India, European Union and South Africa, CARMAv3 computed plant-level emissions considering estimated key variables (i.e., capacity factor, heat rate and CO2 emission factors) using statistical models fitted to a detailed dataset of USA facilities. This approach is completely different to the one considered in the present work, which consist on first estimating the $CO_2$ emissions at the national level per fuel type and then distributing them across facilities considering the installed capacity of each plant.
- For about 70% of the CARMAv3 power plants, the geocoding was performed using an algorithm that derive city-center latitude and longitude coordinates from the geopolitical data (i.e., country, state/province, and city names) provided by WEPPS. On the other hand, for the facilities located in Europe, USA and Canada (approximately 6000), exact geographical coordinates were obtained from high resolution disclosure databases and manual geocoding.

**Injection height - Especially, the vertical injection height section is something that could be more quantitative. How about the sensitivity of the height to the modeling results? What are the potential impacts? Regional difference? How can we determine if we should use your data? Would the inclusion of the injection height be a worth a try given the improvement we could potential get? Of course, there won't be a single answer given there are many study applications. However, even one single example of the potential use cases should help the potential users, and more importantly, should support the "usefulness (and degree/level of the usefulness)" that the authors claim objectively.**

The impact of the injection heights or other elements included in the present catalogue (e.g., geolocation of emissions, temporal profiles) on modelling results is out of the scope of this manuscript. The main objective of this paper is to describe the catalogue and methodologies considered to develop it, as well as to compare the emission results against independent emission datasets. In this sense, the total estimated emissions have been compared against independent gridded, country-level (i.e., EDGAR and various national inventories) and plant-level (i.e., CARMA, GID, Grant et al., 2021) datasets. The temporal profiles constructed in this work have now also been compared against independent databases (i.e., the EDGAR temporal profiles and plant-level profiles, see next comments for more details). Concerning the constructed vertical injection height profiles, we have now added a comparison against the profile proposed by TNO for the public energy sector in the Copernicus CAMS-REG inventory (Kuenen et al., 2021), which is based on the work by Bieser et al. (2011) and it is widely used among the air quality modelling community in Europe. The comparison is presented in Figure 20 and discussed in Section 3.4 of the revised manuscript.

We believe including all these emission intercomparison exercises in the manuscript already provides the reader an idea of the level of usefulness of the catalogue. Adding study applications that show the impact of using this emission catalogue instead of other state-of-the-art inventories (e.g., EDGAR) or emission proxies (e.g., independent vertical profiles) on $CO_2$ modelling results is an independent task that we plan to tackle separately in a future publication. As the reviewer points out, there is not a single answer given the fact that there are many study applications, and therefore a complete modelling framework should be set up to properly answer this point. We have included a reflection on this point in the "Future perspective" section of the manuscript.

**Temporal error – Is Crippa et al. (2022) high temporal? How are those two different (or similar) and what are the potential impacts we see in modelling applications? It is for 2015, but we could easily imagine that many users would have to use the temporal profiles for different years. In inventory research and applications, it is easy to imagine a situation to use the 2015 seasonality to other non-COVID times. Thus, regardless of the 2015 vs. 2018 base year difference, the comparison should provide practical information to guide the future users of this database**

Following the reviewer's suggestion, we performed a comparison between the country-level monthly profiles derived from this work and reported by Crippa et al. (2020). We considered the 2018 EDGARv7 $CO_2$ monthly emissions per country, which are calculated using the temporal distribution profiles

described in Crippa et al. (2020). We normalized these emissions to derive country-level monthly profiles and compared them against our results. We computed the correlation obtained between monthly profiles per country and plot the results in a map for visualization purposes (Figure 17 of the revised manuscript). The discussion of the results and corresponding figure were added in the revised version of the manuscript (lines 1116 to 1135 of the revised manuscript). As discussed in the previous comment, the inclusion of modelling application to see the impact of using different temporal profiles on $CO_2$ concentrations is out of the scope of the present paper.

Concerning the representativeness and stability of the temporal profiles over the years, we refer to the analysis performed by Crippa et al. (2020), in which monthly temporal profiles for the power generation sector for the 35 Organisation for Economic Co-operation and Development (OECD) countries over the time period 2000–2017 was analyzed. According to their results, large standard deviations mainly occur in countries where the use of fossil fuels to generate electricity is scarce such as Finland, Iceland, Norway and Sweden, where more than 90% of total electricity comes from renewable sources, or where the number of fossil fuel power plants that supply energy to the grid is very low (e.g., five in the case of Latvia). These situations can cause large relative year to year changes in the monthly profiles, as they are more sensible to changes in economic variables, meteorological conditions or the dynamics of specific facilities. On the other hand, and also according to Crippa et al. (2020), the year-to-year variations of the monthly profiles obtained for top emitting countries (e.g., China, Japan, USA, Australia) is in general much lower. We have included this point in the "Limitations of the dataset" section of the revised manuscript (lines 1310 to 1320 of the revised manuscript).

**Errors across different species – This is not just CO2, but other species. This study should evaluate location in space and time (as mentioned earlier), and also the different errors over different species. It is hard to imagine that we can use the same error/uncertainty estimates for non-CO2 other than location errors. Are they the same?**

Unfortunately, none of the other plant-level emission inventories used for comparison include information on co-emitted species. That's why for non-CO2 emissions we performed comparisons at the country level against EDGAR and other national emission inventories. The results of our comparisons indicate larger discrepancies between NOx and SOx estimates than between $CO_2$ estimates, which suggests that in fact the error and uncertainty estimates for these two species is larger than for $CO_2$. This is in line with the fact that the estimation of $NO_x$ and $SO_x$ emissions is typically much complex than for $CO_2$ as there are more elements that drive or influence the emission rates, such as combustion conditions, combustion technologies or air pollution control levels implemented in the facilities, among others.

Based on these comparisons and comments from the two other reviewers, in this new version of the manuscript we have improved the estimation of co-emitted species by considering new fuel- and country-dependent emission ratios based on the GAINS inventory, which considers the impact of heterogeneous implementation of technologies / air pollution control levels in power plants across world regions. The comparison of our updated $NO_x$ and $SO_x$ emissions against independent national estimates indicate a better agreement.

Additionally, we included a new point in the "Future perspective" section, mentioning the current lack of alternative bottom-up plant-level emission datasets for co-emitted species to compare with, and suggesting the possibility of performing comparisons against satellite-bases point source catalogues in future works, such as the ones reported by Beirle et al. (2023) or Fioletov et al. (2023) (lines 1390 to 1392 of the revised manuscript).

*Beirle, S., Borger, C., Jost, A., and Wagner, T.: Improved catalog of NOx point source emissions (version 2), Earth Syst. Sci. Data, 15, 3051–3073, https://doi.org/10.5194/essd-15-3051-2023, 2023*

*Fioletov, V. E., McLinden, C. A., Griffin, D., Abboud, I., Krotkov, N., Leonard, P. J. T., Li, C., Joiner, J., Theys, N., and Carn, S.: Version 2 of the global catalogue of large anthropogenic and volcanic SO2 sources and*

*emissions derived from satellite measurements, Earth Syst. Sci. Data, 15, 75–93, https://doi.org/10.5194/essd-15-75-2023, 2023.*

**Potential conflict of interest**

**The main author and the handling editor are working in the same working group under the Global Emission Initiative (GEIA). It is totally understandable that finding a right handling editor is a challenge for ESSD. However, the potential conflict of interest should've been clearly indicated, even if it is very minor, in order to protect the scientific integrity that the authors and the handling editor have maintained.**

The main author of the manuscript (M. Guevara) co-leads a GEIA working group on COVID-19 emissions, in which the handling editor participates as a key expert together with 31 other experts, as shown in this GEIA website: https://www.geiacenter.org/analysis/working-groups/covid-19-working-group. The interaction between WG co-leads and key experts has been up to now very limited – only a couple of online meetings and a participation in an online workshop, and it has not materialized in any type of collaboration with the handling editor in terms of e.g., funded projects or published articles.

**Line by line comments**

**P2, L45: inadequate – this depends on your purpose/application. Need elaboration. If the authors claim that this dataset is more "adequate," you need to objectively demonstrate the skill and utility. You also have limitations. Where are the impacts of the limitations in terms of potential errors and uncertainties? These could be discussed in the database space and also in an applications space (e.g. composition simulations)**

Authors agree with the reviewer. We have erased the "inadequate" adjective, and add the following statement to better clarify our point:

"Moreover, these inventories do not report the emissions from facilities at their exact geographical locations, but in the centroid of the respective inventory grid cells which typically have resolution of 0.1x0.1 degrees. Subsequently deviations from their exact locations can be up to a few kilometres. While this fact does not entail limitations for modelling applications working at the same or lower spatial resolutions (e.g., Agustí-Panadera et al., 2022), it may become critical for local and very high-resolution modelling applications (e.g., Brunner et al., 2023)." (lines 58 to 61 of the revised manuscript)

*Agustí-Panareda, A., McNorton, J., Balsamo, G., Baier, B.C., Bousserez, N., Boussetta, S., Brunner, D., Chevallier, F., Choulga, M., Diamantakis, M., Engelen, R., Flemming, J., Granier, C., Guevara, M., Denier van der Gon, H., Elguindi, N., Haussaire, J.-M., Jung, M., Janssens-Maenhout, G., Kivi, R., Massart, S., Papale, D., Parrington, M., Razinger, M., Sweeney, C., Vermeulen, A., and Walther, S.: Global nature run data with realistic high-resolution carbon weather for the year of the Paris Agreement. Sci Data, 9, 160, https://doi.org/10.1038/s41597-022-01228-2, 2022.*

*Brunner, D., Kuhlmann, G., Henne, S., Koene, E., Kern, B., Wolff, S., Voigt, C., Jöckel, P., Kiemle, C., Roiger, A., Fiehn, A., Krautwurst, S., Gerilowski, K., Bovensmann, H., Borchardt, J., Galkowski, M., Gerbig, C., Marshall, J., Klonecki, A., Prunet, P., Hanfland, R., Pattantyús-Ábrahám, M., Wyszogrodzki, A., and Fix, A.: Evaluation of simulated CO2 power plant plumes from six high-resolution atmospheric transport models, Atmos. Chem. Phys., 23, 2699–2728, https://doi.org/10.5194/acp-23-2699-2023, 2023.*

**P2, L49: 0.1 deg – this is a statement for EDGAR? Relating to the comment above. The uncertainty should depend on the spatial resolution.**

EDGAR emissions are reported in a 0.1x0.1deg, and all emissions are referenced to the centroid of each grid cell. We have clarified our point regarding the adequacy of using point source emissions at such spatial resolutions (see comment below).

**P2, L50: if so, you could use the 0.1 degree database to compare to the database presented in this manuscript**

We included this comparison in the new Sect. 3.2.2 of the revised version of the manuscript (discussed in more detail in a comment above)

**P2, L60: not perfectly – as far as I understood.**

Authors agree with the reviewer. The temporal profiles are country- and fuel-dependent. This point has been clarified here as well as in the abstract and conclusions sections of the revised manuscript.

**P3, L71: where are those datasets available?**

We have re-written the text to avoid confusions:

"The global point source database is a mosaic constructed using as a basis the European and global power plant databases described in Sect. 2.1." (lines 86 to 87 of the revised manuscript)

**2.1 Create a table? Base year 2018 reasonable? What are the uncertainties associated with that?**

We have included a summary table (Table 1 of the revised manuscript) describing the main characteristics of the power plant datasets considered to construct the inventory and specifying for which countries and fuels we used each one of them. The reference year of each database considered has been added to the summary table. For E-PRTR_v18, LCP_v5.2, IRD_v7 and eGRIDv2018 the information reported is for the year 2018 (other versions of the same databases will report the data for other years). For the GCPTv2021_01 and GGPTv2021_02 the base year is 2021, but we only selected those power plants that were operating in 2018 considering the information on status and start/retired year. For the other datasets, which base years are 2021/2022 for most of the cases, we assumed all power plants included were already operating in 2018, as no information on the status or start/retired year is reported. This information has been also added in a footnote in the new Table 1.

**L5, L126: Another assumption here. What is the potential impact of this?**

This limitation could have an impact on dual-fuel power plants that can use more than one fuel to operate (e.g., natural gas/diesel), as only emissions from its primary fuel will be allocated in them. This potential impact has been added into the revised version of the manuscript (lines 168 to 169 of the revised manuscript).

**2.2 how many of them were done by manual search?**

For 133 plants, the fuel type was determined by a manual search online.

**P13, Figure 4 , maybe by region/country? You should try to examine the utility of the gap filling of any approach by using the existent data,**

We added information regarding the share of $CO_2$ emission coverage at the world region level. Results indicate that the shares significantly vary across world regions. For the stack height parameter, in USA, South Africa and India the share is between 75% and 90%, while in Central Europe is around 50%. In many Asian, African and South American regions the share is below 5%. Regarding the utility of the gap filling that we apply, we refer to the comment introduced in the "Limitations of the dataset" section, where we refer to the sensitivity runs performed by Bieser et al. (2011), the results of which indicate that changes in estimated emission heights are almost linear with changes in stack height and exit velocity.

*Bieser, J., Aulinger, A., Matthias, V., Quante, M., and van der Gon, H. D.: Vertical emission profiles for Europe based on plume rise calculations, Environ. Pollut., 159, 2935–2946, https://doi.org/10.1016/j.envpol.2011.04.030, 2011.*

**3.2 spatial errors**

**P20, L405, not always. See Oda et al. 2019**

The work by Oda et al., (2019) includes an analysis of geolocation differences between Polish point sources reported by CARMA/ODIAC and a local bottom-up inventory (GESAPU) for a specific region of Poland (Lesser Poland Voivodeship). The work managed to find three perfectly paired plants between the two datasets, which were closely located (geolocation differences of between 1.3–3.2 km). These differences are in line with our findings, which focused on comparing the geographical location reported by the present catalogue and CARMAv3 for the top 20 emitters of Poland (average geolocation differences of 633 m, with a maximum difference of 5.2km). Our analysis only focusses on the top 20 emitters, and it is possible that for other small-to-medium plants the differences between locations are larger. However, performing such a detailed analysis is out of the scope of this work, as it would entail performing an extreme labor-intensive match up between CARMAv3 and present work facilities.

**P21, L414, R2 0.8 is concerning if at plant level. Is it adequate?**

Authors fail to understand why the reviewer thinks that a $R^2 = 0.8$ is a concerning result. This statistical score is the result of comparing the $CO_2$ emissions of the Indian and Canadian top 100 emitters as reported by our work and independent national power plant inventories at the plant level. Considering the assumptions made in our study (e.g., distribution of total national emissions among facilities as a function of their installed capacity), we believe the correlation obtained is quite satisfactory.

We have selected the Indian top 50 emitters and estimated the correlations obtained when comparing our results versus CEA and GIDv1.1 versus CEA. As observed in the figures below, the correlation obtained with our catalogue (0.75) is slightly higher than the one obtained with the GIDv1.1 (0.6), which is considered a state-of-the-art power plant catalogue. Therefore, we believe that the correlation value obtained in our comparison exercise is acceptable. We have also compared the RMSE values estimated in each case, the values obtained with this work (3291 kt $CO_2$ · year) being lower than with GIDv1.1 (4534 kt $CO_2$ · year).

[Figure]

**P21, L428, if CO2 is this bad, other compounds could be way worse.**

We disagree with the reviewer in qualifying the results of the $CO_2$ comparisons as "bad". As mentioned in the manuscript, for e.g., Russia we report -34% less $CO_2$ emissions than EDGAR mainly because we do not have information on the location of heat-only plants, which substantially contribute to total fossil fuel consumption of the power sector in this country (i.e., 20%). This limitation is also present in other state-of-the-art point source catalogues, include the GID database (Tong et al., 2018), which is mentioned by the reviewer in a previous comment. GID reports a total of 618.2 Mt $CO_2$ emissions for

Russia for 2019 for the power sector, which is close to the value we get in our database for the year 2018 (536.4 Mt $CO_2 \cdot$ year), and therefore also considerably underestimated when compared to the EDGAR value.

**P21, L442, CH4 needs to be evaluated.**

We are already performing an evaluation of $CH_4$ by comparing our results against the EDGAR inventory and national estimates. We could not find plant-level emission databases to compare with. Moreover, and as mentioned in the manuscript, the importance of $CH_4$ emissions from the power sector is very limited:

"the share of the power sector to total national $CH_4$ emissions is small, often around or below 1% (e.g., 0.24% for Italy, 0.19% for Poland, 1.1% for Sweden; UNFCCC, 2022). Hence, the deviations have a negligible influence on national total CH4 emissions and are not further investigated. Moreover, CH4 emissions in power plants are scarcely measured and that corresponding emission factors are associated to very large uncertainties (IPCC, 2019)."

*IPCC: 2019 Refinement to the 2006 IPCC Guidelines for National Greenhouse Gas Inventories, Calvo Buendia, E., Tanabe, K., Kranjc, A., Baasansuren, J., Fukuda, M., Ngarize, S., Osako, A., Pyrozhenko, Y., Shermanau, P. and Federici, S. (eds). Published: IPCC, Switzerland, available at: https://www.ipcc-nggip.iges.or.jp/public/2019rf/index.html (last access: October 2022), 2019.*

*UNFCCC: National Inventory Submissions 2022, available at: https://unfccc.int/ghg-inventories-annex-i-parties/2022 (last access: October 2022), 2022.*

**P22, L443, why?**

The discrepancies we see between our $NO_x$ results and EDGAR/national estimates are because for the estimation of co-emitted species we make use of fuel-dependent emission ratios that do not vary across countries and therefore do not take into account the impact of technologies / air pollution control levels implemented in each country. We have corrected this limitation by constructing a new set of emission ratios that not only vary per fuel but also per country making use of GAINS emission inventory, as explained in Section 2.3.2 of the revised manuscript. The comparison of our updated $NO_x$ (and $SO_x$) emission results against independent national estimates indicates a better agreement when considering the updated emission ratios.

**L26, L504. ESSD paper should stand by itself. Describe what the system does for the temporal modeling.**

We added a description of the expression used in HERMESv3 to perform the temporal distribution of the power plant emissions (lines 1059 to 1065 of the revised manuscript).

**Figure 11, What are the uncertainties around the temporal estimates?**

As mentioned in the "Limitations" section, temporal profiles assigned to the power plants are country and fuel-dependent, but not facility-dependent. Large differences between the emission temporal distribution of plants belonging to the same country may occur, e.g., if they are used for electricity only or electricity and heat. However, information to develop such detailed level of profiles is very scarce and limited only to certain regions (e.g., EU27 using the ENTSO-E database). This lack of information also limits our capacity for explicitly quantifying the uncertainty associated to the constructed profiles.

Nevertheless, and to address the reviewer's comment, we included an example to illustrate the differences between country and fuel-level versus plant-level temporal profiles. We compared the monthly, weekly and hourly profiles constructed in the present work for German coal-fired power plants against profiles estimated for individual facilities making use of plant-level electricity generation statistics provided by ENTSO-E (2021). The comparisons include the top 20 producing coal-fired German power plants, which together generated more than 75% of the national electricity using coal

in 2018. Results indicate a significant heterogeneity between profiles across individual plants. As expected, the more electricity a power plant produces, the more continuously it supplies electrical energy throughout the year and, subsequently, the flatter are its associated monthly, weekly and hourly profiles. On the contrary, power plants producing less energy tend to show large variations between e.g., weekdays and weekends and daytime and nighttime hours, as their behavior is more linked to demand response. The discrepancies between our profiles and the plant-level profiles tends to be lower when looking at the top generating facilities. A detailed discussion of the results was included in the revised manuscript (lines 1145 to 1165 and Figure 18 of the revised manuscript).

*ENTSO-E. European Network of Transmission System Operators. Transparency Platform. Available at: https://transparency.entsoe.eu/ (last access: November 2022), 2021.*

**4 data Name is different. What is the readme? That's what we need here, too.**

We have modified the dataset name to be consistent. The README file is a .docx file that provides a detailed description of the files contained in the global catalogue and associated fields of information. We have specified this point better to avoid confusions.

**5.1 limitation should be before the conclusion and discussed in more detail. Some of them could be examined in this study.**

Following the reviewer's recommendation, and as already mentioned in a previous comment, we have moved the Limitation section before the conclusions section (Section 6 of the revised manuscript). We have examined in more detail the limitations related to the temporal distribution profiles, as discussed in previous comments. We have also added more details into the discussion of the gap filling procedure and use of emission ratios to estimated emissions from co-emitted species.

**P34, L675 not just for inverse modeling. This is a huge limitation for the potential users of this product.**

We have erased the sentence "which is a key information for inverse modelling studies". As mentioned in the manuscript, we will address the inclusion of uncertainty estimates in future works. There are many emission datasets that have been published in ESSD without providing uncertainty information (e.g., many of the papers published under the "surface emissions for atmospheric chemistry and air quality modelling" special edition) and therefore we believe this limitation should not prevent this manuscript from being published.

**Supplemental information**

**Figure 2. The correlation should be high. The high correlation does not really tell us any practical information regarding the performance of the dataset. How about using RMSE for location and emission intensity as a metric for potential improvements brought by this dataset?**

As mentioned in a previous comment, the authors fail to understand why the reviewer thinks that a correlation of 0.8 is a concerning result. Note that the national plant-level databases considered do not report the location of the power plants, but just the total annual emissions per facility. More details are provided in the previous comment that focusses on this aspect.